# $\sigma$-PCA: A UNIFIED NEURAL MODEL FOR LINEAR AND NONLINEAR PRINCIPAL COMPONENT ANALYSIS

## ABSTRACT

Linear principal component analysis (PCA), nonlinear PCA, and linear independent component analysis (ICA) – those are three methods with single-layer autoencoder formulations for learning linear transformations with certain characteristics from data. Linear PCA learns rotations that orient axes to maximise variance, but it suffers from a subspace rotational indeterminacy: it fails to find a unique rotation for axes that share the same variance. Both nonlinear PCA and linear ICA reduce the subspace indeterminacy from rotational to permutational by maximising statistical independence under the assumption of unit variance. The main difference between them is that nonlinear PCA only learns rotations while linear ICA learns not just rotations but linear transformations. The relationship between all three can be understood by the singular value decomposition of the linear ICA transformation into a sequence of rotation, scale, rotation. Linear PCA learns the first rotation; nonlinear PCA learns the second. The scale is simply the inverse of the standard deviations. The problem is that, in contrast to linear PCA, conventional nonlinear PCA cannot be used directly on the data to learn the first rotation, the first being special as it reduces dimensionality and orders by variances. In this paper, we have identified the cause, and as a solution we propose $\sigma$-PCA: a unified neural model for linear and nonlinear PCA as single-layer autoencoders. One of its key ingredients: modelling not just the rotation but also the scale – the variances. This model bridges the disparity between linear and nonlinear PCA. With our formulation, nonlinear PCA can learn not just the second, but also the first rotation. And so, like linear PCA, it can learn a semi-orthogonal transformation (a rotation) that reduces dimensionality and orders by variances, but, unlike linear PCA, it does not suffer from rotational indeterminacy.

## 1 INTRODUCTION

Principal component analysis (PCA) (Pearson, 1901; Hotelling, 1933) needs no introduction. It is classical, ubiquitous, perennial. It is an unsupervised learning method that can be arrived at from three paradigms of representation learning (Bengio et al., 2013): neural, manifold, and probabilistic. This paper is about learning *linear* transformations from data using the first paradigm: the *neural*, in the form of a single-layer autoencoder – a model which encodes the data into a new representation and then decodes that representation back into a reconstruction of the original data.

From the data, PCA learns a linear *orthogonal* transformation $\mathbf{W}$ that transforms a given data point $\mathbf{x}$ into a new representation $\mathbf{y} = \mathbf{xW}$, a representation with reduced dimensionality and minimal loss of information (Diamantaras & Kung, 1996; Jolliffe & Cadima, 2016). This representation has components that are uncorrelated, i.e. have no linear associations, or, in other words, the covariance between two distinct components is zero. When the variances of the components are clearly distinct, PCA has a unique solution, up to sign indeterminacies, that consists in a set of principal eigenvectors (or axes) ordered by their variances. This solution can usually be obtained from the singular value decomposition (SVD) of the data matrix (or the eigendecomposition of the data covariance matrix). Such a solution not only maximises the amount of variance but also minimises the error of reconstruction. And it is the latter – in the form of the squared difference between the data and its reconstruction – that is also the loss function of a single-layer linear autoencoder. However, minimising such a loss will result in a solution that lies in the PCA *subspace* (Baldi & Hornik, 1989; Bourlard & Kamp, 1988; Oja, 1992b), i.e. a linear combination of the principal eigenvectors, rather

than the actual principal eigenvectors. And so, a linear autoencoder performs PCA – but not in its expected axis-aligned form.

What happens when we insert a nonlinearity in the single-layer autoencoder? By natural extension, it must be a form of nonlinear PCA (Xu, 1993; Karhunen & Joutsensalo, 1994; 1995; Karhunen et al., 1998). However, to yield an interesting output, it was observed that the input data needs to be *whitened*, i.e. transformed into uncorrelated components of *unit* variance. And after the input is whitened, applying nonlinear PCA results in a similar output (Karhunen & Joutsensalo, 1994; 1995; Karhunen et al., 1998; Hyvärinen & Oja, 2000) to linear independent component analysis (ICA) (Jutten & Herault, 1991; Comon, 1994; Bell & Sejnowski, 1997; Hyvarinen, 1999). [1]

Linear ICA, too with a long history in unsupervised learning (Hyvärinen & Oja, 2000), yields, as the name implies, not just uncorrelated but independent components. ICA seeks to find a transformation $\mathbf{B}$ – not necessarily orthogonal – such that $\mathbf{y} = \mathbf{x}\mathbf{B}$ has its components as statistically independent as possible. It is based on the idea that $\mathbf{x}$ is an observed linear mixing of the hidden sources $\mathbf{y}$, and it aims to recover these sources solely based on the assumptions that they are non-Gaussian and independent. Without any other assumptions, what is gained in independence is lost in precision: unlike PCA, we cannot determine the variances of the components nor their order (Hyvarinen, 1999; Hyvärinen & Oja, 2000; Hyvärinen, 2015); they are impossible to recover when we have no prior knowledge about how the data was generated. For if $\mathbf{B}$ is a transformation that recovers the sources, then so is $\mathbf{B}\mathbf{\Lambda}$, with $\mathbf{\Lambda}$ an arbitrary diagonal scaling matrix. And so, the best that can be done, without any priors, is to assume the sources have *unit* variance.

What is known of all three is the following: linear PCA can learn rotations that orient axes into directions that maximise variance, but it suffers from subspace rotational indeterminacy: it does not know what to do with axes that share the same variance, so those axes are arbitrarily rotated. Nonlinear PCA can learn rotations that orient axes into directions that maximise statistical independence, reducing the subspace indeterminacy from rotational to a trivial permutational, but it cannot be applied directly to data without preprocessing – whitening. Linear ICA learns a linear transformation – not just a rotation – that points the axes into directions that maximise statistical independence. The relationship between all three can be best understood by the SVD of the linear ICA transformation into a sequence of rotation, scale, rotation – stated formally, $\mathbf{W}\mathbf{\Sigma}^{-1}\mathbf{V}$, with $\mathbf{W}$ and $\mathbf{V}$ orthogonal, and $\mathbf{\Sigma}$ diagonal. In one formulation of linear ICA, linear PCA learns the first rotation, $\mathbf{W}$, and nonlinear PCA learns the second, $\mathbf{V}$. The scale, $\mathbf{\Sigma}^{-1}$, is simply the inverse of the standard deviations.

The problem is that, in contrast to linear PCA, conventional nonlinear PCA cannot be used directly on the data to learn the first rotation, the first being special as it can reduce dimensionality and order by variances. The process of first applying the linear PCA rotation and then dividing by the standard deviations is the whitening preprocessing step, the prerequisite for nonlinear PCA. And so, nonlinear PCA, rather than being on an equal footing, is dependent on linear PCA. Conventional nonlinear PCA, by the mere introduction of a nonlinear function, loses the ability for dimensionality reduction and ordering by variances – a jarring disparity between both linear and nonlinear PCA.

In this paper, we have identified the reason why conventional nonlinear PCA has been unable to recover the first rotation, $\mathbf{W}$: it has been missing, in the reconstruction loss, $\mathbf{\Sigma}$. This means that the nonlinear PCA model we want to learn is not $\mathbf{W}$ but $\mathbf{W}\mathbf{\Sigma}^{-1}$. The reason why $\mathbf{\Sigma}$ is needed is that it puts the components on the same scale before applying the nonlinearity. Another key observation for nonlinear PCA to work for dimensionality reduction is that it should put an emphasis not on the decoder contribution, but on the on the encoder contribution – in contrast to conventional nonlinear PCA. We call this unified model $\sigma$-PCA to distinguish it from the conventional PCA model, which merely is a special case with $\mathbf{\Sigma} = \mathbf{I}$. [2] With our formulation, nonlinear PCA can now learn not just the second, but also the first rotation, i.e. it can also be applied directly to the data without whitening. And so, like linear PCA, it can learn an orthogonal transformation (a rotation) that reduces dimensionality and orders by variances, but, unlike linear PCA, it does not suffer from rotational indeterminacy. Our main contribution is thus carving out a place for nonlinear PCA as a method in its own right.

---

[1] For ICA, the terms linear and nonlinear refer to the transformation, while, for PCA, they refer to the function. Both linear ICA and nonlinear PCA use nonlinear functions, but their resulting transformations are still linear.

[2] Arguably, $\sigma$-PCA should be the general PCA model, for it is common to divide by the standard deviations when performing the PCA transformation. To avoid scattering $\sigma$- every time we mention PCA, we shall simply refer to it as PCA and to the model with $\mathbf{\Sigma} = \mathbf{I}$ as the conventional PCA model.

## 2 BACKGROUND

### 2.1 CONVENTIONAL LINEAR PCA

Let $\mathbf{X} \in \mathbb{R}^{n \times p}$ be the *centred* data matrix, with $\mathbf{x}$ a row vector of $\mathbf{X}$ representing a data sample. The canonical solution of PCA can be obtained as the eigendecomposition of the data covariance matrix (see Appendix A.1). If all the variances are clearly distinct, there is a unique solution, up to sign indeterminacies, where we obtain an ordered set of orthonormal principal eigenvectors $\{\mathbf{e}_1, ..., \mathbf{e}_p\}$ ordered by a corresponding set of eigenvalues $\lambda_1 > ... > \lambda_p > 0$ representing the variances.

Here, we are interested in the neural formulation of PCA as a tied linear autoencoder. In this form, PCA finds a matrix $\mathbf{W} \in \mathbb{R}^{p \times k}$, with $k \leq p$, that minimises the reconstruction loss

$$\mathcal{L}(\mathbf{W}) = \mathbb{E}(||\mathbf{x} - \mathbf{x}\mathbf{W}\mathbf{W}^T||_2^2) \tag{1}$$

under the orthonormality constraint $\mathbf{W}^T\mathbf{W} = \mathbf{I}$. From this we see that PCA transforms the data with $\mathbf{W}$ (the encoder) into a set of components $\mathbf{y} = \mathbf{x}\mathbf{W}$, and then reconstructs the original data with $\mathbf{W}^T$ (the decoder) to obtain the reconstruction $\hat{\mathbf{x}} = \mathbf{y}\mathbf{W}^T$. If $k < p$ then PCA performs dimensionality reduction. The minimum of Eq. 1 is obtained by *any* orthonormal basis of the PCA subspace span$(\mathbf{e}_1, ..., \mathbf{e}_k)$. That is, the optimal $\mathbf{W}$ has the form

$$\mathbf{ER} = [\pm\mathbf{e}_1^T, ..., \pm\mathbf{e}_k^T]\mathbf{R}, \tag{2}$$

where $\mathbf{R} \in \mathbb{R}^{k \times k}$ is any square orthogonal matrix, representing an arbitrary rotation; this is because we have $\mathbf{ERR}^T\mathbf{E}^T = \mathbf{EE}^T$ – a rotational indeterminacy over the entire subspace. If $\mathbf{R} = \mathbf{I}$, then the components are axis aligned. Minimising Eq. 1 on its own will not lead to axis-aligned components, because the loss is symmetric: no component is favoured over the other. To obtain axis-aligned components, we simply need to break the symmetry (see Section 3.6 and Appendix A.3).

### 2.2 CONVENTIONAL NONLINEAR PCA

A straightforward extension (Xu, 1993; Karhunen & Joutsensalo, 1994; 1995; Karhunen et al., 1998) from linear to nonlinear PCA is to simply introduce in Eq. 1 a nonlinearity $h$, such as $\tanh$. The reconstruction loss, under the constraint $\mathbf{W}^T\mathbf{W} = \mathbf{I}$, becomes

$$\mathcal{L}(\mathbf{W}) = \frac{1}{2}\mathbb{E}(||\mathbf{x} - h(\mathbf{x}\mathbf{W})\mathbf{W}^T||_2^2). \tag{3}$$

We might expect for this to work well, it, in fact, does not; in particular, it does not work when the variances of the components are far from unit variance. However, there is currently a remedy: whitening. Whitening is a common pre-processing step (Olshausen & Field, 1996; Cardoso & Laheld, 1996; Karhunen et al., 1997; Hyvärinen & Oja, 2000; Vincent et al., 2010; Coates et al., 2011; Coates & Ng, 2012), and it involves finding a matrix $\mathbf{A}$ such that $\mathbb{E}((\mathbf{x}\mathbf{A})^T\mathbf{x}\mathbf{A}) = \mathbf{I}$. Whitening decorrelates the input such that the components have unit variance, and, in particular, applying any other rotation $\mathbf{W}$ would preserve the unit variance: $\mathbb{E}(\mathbf{W}^T\mathbf{A}^T\mathbf{x}^T\mathbf{x}\mathbf{A}\mathbf{W}) = \mathbf{W}^T\mathbb{E}((\mathbf{x}\mathbf{A})^T\mathbf{x}\mathbf{A})\mathbf{W} = \mathbf{W}^T\mathbf{W} = \mathbf{I}$.

When the input is whitened, nonlinear PCA becomes related to linear ICA, having an exact relationship with kurtosis maximisation or maximum likelihood (Karhunen et al., 1998; Hyvärinen & Oja, 2000). A special case of whitening is linear PCA – that is, if $\mathbf{U}\mathbf{S}\mathbf{V}^T$ is the SVD of the covariance matrix of $\mathbf{X}$, then $\mathbf{A} = \mathbf{V}\mathbf{S}^{-\frac{1}{2}}$ is a whitening matrix with $\mathbf{V}$ orthonormal and $\mathbf{S}$ diagonal. FastICA (Hyvarinen, 1999) uses linear PCA as a pre-processing step to whiten the input data. Other ICA methods (Cardoso & Laheld, 1996; Karhunen et al., 1998) derive a combined update rule for $\mathbf{A}\mathbf{W}$. With nonlinear PCA, whitening makes it impossible to determine the variances and results in an overall transformation $\mathbf{A}\mathbf{W}$ that is no longer orthogonal – in stark contrast to linear PCA.

## 3 UNIFIED NEURAL MODEL FOR LINEAR AND NONLINEAR PCA

Instead of whitening, we propose to introduce a diagonal matrix $\mathbf{\Sigma} = \text{diag}(\boldsymbol{\sigma})$, representing the standard deviations, so that the reconstruction loss becomes

$$\mathcal{L}(\mathbf{W}, \boldsymbol{\Sigma}) = \mathbb{E}(||\mathbf{x} - h(\mathbf{x}\mathbf{W}\boldsymbol{\Sigma}^{-1})\boldsymbol{\Sigma}\mathbf{W}^T||_2^2), \tag{4}$$

with $\mathbf{W}^T\mathbf{W} = \mathbf{I}$. Formulating it this way allows us to unify linear and nonlinear PCA. Indeed, if $h$ is linear, then we recover the linear PCA loss: $\mathbb{E}(||\mathbf{x} - \mathbf{x}\mathbf{W}\boldsymbol{\Sigma}^{-1}\boldsymbol{\Sigma}\mathbf{W}^T||_2^2) = \mathbb{E}(||\mathbf{x} - \mathbf{x}\mathbf{W}\mathbf{W}^T||_2^2)$.

Intuitively, the reason why $\boldsymbol{\Sigma}$ is necessary is because it standardises the components of $\mathbf{y} = \mathbf{x}\mathbf{W}$ to be on the same scale, namely to have unit variance, before applying $h$.

## 3.1 Weight update analysis

If we try to optimise the loss in Eq. 4 with gradient descent, nothing interesting happens. To understand why, we need to have a closer look at the individual contributions of both the encoder and the decoder to the update of $\mathbf{W}$. Given that gradient contributions are additive, we will be computing each contribution separately then adding them up. This is akin to untying the weights of the encoder $(\mathbf{W}_e, \boldsymbol{\Sigma}_e)$ and the decoder $(\mathbf{W}_d, \boldsymbol{\Sigma}_d)$, computing their respective gradients, and then tying them back up. Without loss of generality, we will look at the stochastic gradient descent update (i.e. omit the expectation), and multiply by $\frac{1}{2}$ for convenience, so that we have

$$\mathcal{L}(\mathbf{W}_e, \mathbf{W}_d, \boldsymbol{\Sigma}_e, \boldsymbol{\Sigma}_d) = \frac{1}{2}||\mathbf{x} - h(\mathbf{x}\mathbf{W}_e\boldsymbol{\Sigma}_e^{-1})\boldsymbol{\Sigma}_d\mathbf{W}_d^T||_2^2. \tag{5}$$

Let $\mathbf{y} = \mathbf{x}\mathbf{W}_e$, $\mathbf{z} = \mathbf{y}\boldsymbol{\Sigma}_e^{-1}$, $\hat{\mathbf{x}} = h(\mathbf{z})\boldsymbol{\Sigma}_d\mathbf{W}_d^T$, and $\hat{\mathbf{y}} = \hat{\mathbf{x}}\mathbf{W}_e$. By computing the gradients, we get

$$\frac{\partial \mathcal{L}}{\partial \mathbf{W}_e} = \mathbf{x}^T(\hat{\mathbf{x}} - \mathbf{x})\mathbf{W}_d\boldsymbol{\Sigma}_d \odot h'(\mathbf{z})\boldsymbol{\Sigma}_e^{-1} \tag{6}$$

$$\frac{\partial \mathcal{L}}{\partial \mathbf{W}_d} = (\hat{\mathbf{x}} - \mathbf{x})^T h(\mathbf{z})\boldsymbol{\Sigma}_d \tag{7}$$

Now that we have traced each contribution, we can easily compute the gradient when the weights are tied, i.e. $\mathbf{W} = \mathbf{W}_e = \mathbf{W}_d$ and $\boldsymbol{\Sigma} = \boldsymbol{\Sigma}_e = \boldsymbol{\Sigma}_d$, but we can still inspect each contribution separately:

$$\frac{\partial \mathcal{L}}{\partial \mathbf{W}_e} = \mathbf{x}^T(\hat{\mathbf{y}} - \mathbf{y}) \odot h'(\mathbf{z}) \tag{8}$$

$$\frac{\partial \mathcal{L}}{\partial \mathbf{W}_d} = (\hat{\mathbf{x}} - \mathbf{x})^T h(\mathbf{z})\boldsymbol{\Sigma} \tag{9}$$

$$\frac{\partial \mathcal{L}}{\partial \mathbf{W}} = \frac{\partial \mathcal{L}}{\partial \mathbf{W}_e} + \frac{\partial \mathcal{L}}{\partial \mathbf{W}_d}. \tag{10}$$

From this, we can see the following: the contribution of the decoder, $\frac{\partial \mathcal{L}}{\partial \mathbf{W}_d}$, will be zero when $\hat{\mathbf{x}} \approx \mathbf{x}$, and that of the encoder, $\frac{\partial \mathcal{L}}{\partial \mathbf{W}_e}$, when $\hat{\mathbf{y}} \approx \mathbf{y}$. Thus, unsurprisingly, the decoder puts the emphasis on the input reconstruction, while the encoder puts the emphasis on the latent reconstruction.

In the *linear* case, if $\mathbf{W}$ is orthogonal, the contribution of the encoder is zero. This is because we have $\frac{\partial \mathcal{L}}{\partial \mathbf{W}_e} = \mathbf{x}^T\mathbf{y}(\mathbf{W}^T\mathbf{W} - \mathbf{I}) = \mathbf{0}$, and so it has a negligible effect and can be omitted. We can omit it from the loss by using the stop gradient operator $[\quad]_{sg}$:

$$\mathcal{L}(\mathbf{W}) = ||\mathbf{x} - \mathbf{x}[\mathbf{W}]_{sg}\mathbf{W}^T||_2^2. \tag{11}$$

This results in $\frac{\partial \mathcal{L}}{\partial \mathbf{W}} = -\mathbf{x}^T\mathbf{y} + \mathbf{W}\mathbf{y}^T\mathbf{y}$, which is simply the subspace learning algorithm (Oja, 1989).

In the conventional extension from linear to nonlinear PCA, the contribution of the encoder was also omitted (Karhunen & Joutsensalo, 1994), as it seemed appropriate to simply generalise the subspace learning algorithm (Oja, 1989; Karhunen & Joutsensalo, 1994). But we find that it should be the

other way around for the *nonlinear* case: what matters is not the contribution of the decoder, but the contribution of the encoder.

Indeed, in the nonlinear case, $\frac{\partial \mathcal{L}}{\partial \mathbf{W}_e}$ is no longer zero. The problem is that it is overpowered by the decoder contribution, in particular when performing dimensionality reduction (see Appendix E.6). Arguably it would be desirable that the latent of the reconstruction is as close as possible to the latent of the input. This means that we want to eliminate $\frac{\partial \mathcal{L}}{\partial \mathbf{W}_d}$ from the update and only keep $\frac{\partial \mathcal{L}}{\partial \mathbf{W}_e}$. We thus arrive at what the loss should be in the nonlinear case:

$$\mathcal{L}(\mathbf{W}, \mathbf{\Sigma}) = \boxed{\mathbb{E}(||\mathbf{x} - h(\mathbf{x}\mathbf{W}\mathbf{\Sigma}^{-1})\mathbf{\Sigma}[\mathbf{W}^T]_{sg}||_2^2).} \tag{12}$$

This loss does not maintain unit norm columns of $\mathbf{W}$, so, for that, we need to use a constraint or a regulariser. The former, for instance, can be a projective constraint, i.e. normalising each column of $\mathbf{W}$ to unit norm after each update. (See Appendix C for other methods.).

Without any constraints on a trainable $\mathbf{\Sigma}$, it could increase without limit. For instance, if $h = \tanh$, we have $\lim_{\sigma \to \infty} \sigma \tanh(z/\sigma) = z$. One option is to add an $L_2$ regulariser, which should prevent it from growing, but $\mathbf{\Sigma}$ need not be trainable, as we already have a natural choice for it: the estimated standard deviation of $\mathbf{y}$. Indeed we can set $\mathbf{\Sigma} = \mathrm{diag}(\sqrt{\mathrm{var}(\mathbf{y})})$ (see Appendix E.2). We discuss how $h$ needs to be adjusted for this to work in Section 3.3.

The emphasis on latent reconstruction in Eq. 8 can be put in relation to both blind deconvolution and the $L_1$ sparsity constraints. Both connections are discussed in Appendix E.3 and E.4.

## 3.2 RELATIONSHIP BETWEEN LINEAR PCA, NONLINEAR PCA, AND LINEAR ICA

Linear PCA, nonlinear PCA, and linear ICA seek to find a matrix $\mathbf{B} \in \mathbb{R}^{p \times k}$ such that $\mathbf{y} = \mathbf{x}\mathbf{B}$. Linear PCA puts an emphasis on maximising variance, linear ICA on maximising statistical independence, and nonlinear PCA on both. This $\mathbf{B}$ is found by being decomposed into $\mathbf{W}\mathbf{\Sigma}^{-1}\mathbf{V}$, with $\mathbf{W} \in \mathbb{R}^{p \times k}$ semi-orthogonal, $\mathbf{\Sigma} \in \mathbb{R}^{k \times k}$ diagonal, and $\mathbf{V} \in \mathbb{R}^{k \times k}$ orthogonal.

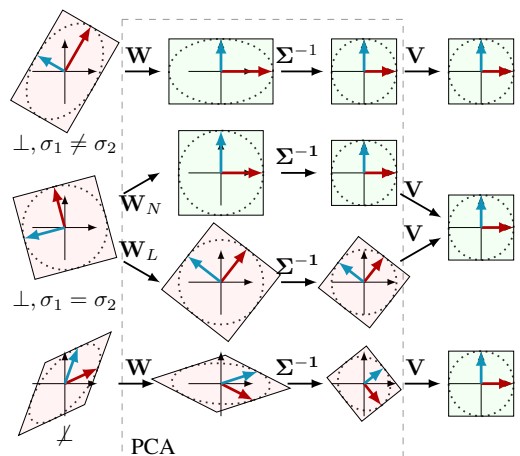

Orthogonality is a defining aspect of PCA, so in the case of both linear and nonlinear PCA, we have $\mathbf{V} = \mathbf{I}$, i.e. the restriction is for $\mathbf{B}$ to have orthogonal columns; there is no such restriction with ICA. Simply stated,

$$\text{PCA} \quad \mathbf{W}\mathbf{\Sigma}^{-1} \tag{13}$$
$$\text{ICA} \quad \mathbf{W}\mathbf{\Sigma}^{-1}\mathbf{V}. \tag{14}$$

Figure 1: Linear PCA, nonlinear PCA, and ICA. Green indicates alignment of axes.

The reconstruction losses follow the same pattern of $\mathbb{E}(||\mathbf{x} - h(\mathbf{x}\mathbf{B})\mathbf{B}^{-1}||_2^2)$, but with a few subtle differences, which we can write as

$$\mathcal{L}_{\text{PCA}}(\mathbf{W}) = \mathbb{E}(||\mathbf{x} - \mathbf{x}\mathbf{W}\mathbf{\Sigma}^{-1}\mathbf{\Sigma}\mathbf{W}^T||_2^2), \tag{15}$$
$$\mathcal{L}_{\text{NLPCA}}(\mathbf{W}) = \mathbb{E}(||\mathbf{x} - h(\mathbf{x}\mathbf{W}\mathbf{\Sigma}^{-1})\mathbf{\Sigma}[\mathbf{W}^T]_{sg}||_2^2), \tag{16}$$
$$\mathcal{L}_{\text{ICA}}(\mathbf{W}, \mathbf{V}) = \mathbb{E}(||\mathbf{x}[\mathbf{W}\mathbf{\Sigma}^{-1}]_{sg} - h(\mathbf{x}[\mathbf{W}\mathbf{\Sigma}^{-1}]_{sg}\mathbf{V})\mathbf{V}^T||_2^2) + \mathcal{L}_{\text{NLPCA}}(\mathbf{W}). \tag{17}$$

Unlike the nonlinear PCA and ICA losses, the linear PCA loss in Eq. 15 does not allow us to obtain an axis-aligned solution (see Section 2.1), but there are a few ways to obtain one (see Section 3.6 and Appendix A.3). Given the unit variance, the ICA loss works with and without a stop gradient on $\mathbf{V}^T$.

Linear PCA emphasises input reconstruction, while nonlinear PCA emphasises latent reconstruction. We can see this clearly from the their respective weight updates (assuming $\mathbf{W}^T\mathbf{W} = \mathbf{I}$):

$$\text{Linear PCA} \quad \Delta\mathbf{W} \propto \mathbf{x}^T\mathbf{y} - \mathbf{W}\mathbf{y}^T\mathbf{y} \qquad = \boxed{(\mathbf{x} - \hat{\mathbf{x}})^T\mathbf{y}}, \tag{18}$$

$$\text{Nonlinear PCA} \quad \Delta\mathbf{W} \propto (\mathbf{x}^T\mathbf{y} - \mathbf{x}^T\hat{\mathbf{x}}\mathbf{W}) \odot h'(\mathbf{y}) = \boxed{\mathbf{x}^T(\mathbf{y} - \hat{\mathbf{y}})} \odot h'(\mathbf{y}). \tag{19}$$

Figure 1 summarises the differences between linear PCA, nonlinear PCA, and ICA based on orthogonality and variances. When the true $\mathbf{B}$ is *orthogonal*, we can note the following: (1) if all the variances are clearly distinct, then all three methods would coincide. (2) If some of the variances are the same or close to each other, then only nonlinear PCA and ICA would coincide; linear PCA would fail to separate the subset of components that have the same variance, because that subset can be multiplied with an arbitrary orthogonal matrix without changing their variance (Comon, 1994; Hyvärinen et al., 2009). In contrast, nonlinear PCA can still separate those components. (3) We can determine the variances of the independent components with nonlinear PCA and ICA (given the assumption that the transformation has unit norm columns).

Suppose that $\boldsymbol{\Sigma}$ is ordered, let $\mathbf{W}_L$ be the axis-aligned linear PCA solution, $\mathbf{W}_N$ the nonlinear PCA solution, and $\mathbf{R}$ a block orthogonal matrix where each block affects the subset of components that share the same variance, then nonlinear PCA is related to linear PCA as follows

$$\mathbf{W}_N\boldsymbol{\Sigma}^{-1} = \mathbf{W}_L\mathbf{R}\boldsymbol{\Sigma}^{-1}. \tag{20}$$

Note that we can commute $\mathbf{R}$ and $\boldsymbol{\Sigma}^{-1}$, i.e. $\mathbf{R}\boldsymbol{\Sigma}^{-1} = \boldsymbol{\Sigma}^{-1}\mathbf{R}$ (see Appendix H).

When the true $\mathbf{B}$ is a *non-orthogonal*, only ICA can recover it. However, ICA can be seen as two isolated layers, where the first layer is linear PCA (if axis-aligned) or nonlinear PCA, and the second layer is nonlinear PCA, resulting in an overall transformation of $\mathbf{B} = \mathbf{W}\boldsymbol{\Sigma}_1^{-1}\mathbf{V}\boldsymbol{\Sigma}_2^{-1}$ with $\boldsymbol{\Sigma}_2 = \mathbf{I}$. We can use either linear or nonlinear PCA as the first layer because

$$\mathbf{B} = \mathbf{W}_N\boldsymbol{\Sigma}^{-1}\mathbf{V} = \mathbf{W}_L\mathbf{R}\boldsymbol{\Sigma}^{-1}\mathbf{V} = \mathbf{W}_L\boldsymbol{\Sigma}^{-1}\mathbf{R}\mathbf{V} = \mathbf{W}_L\boldsymbol{\Sigma}^{-1}\mathbf{V}'. \tag{21}$$

### 3.3 CHOICE OF NONLINEARITY

The choice of nonlinearity generally depends on whether the distribution is sub- or super-Gaussian (Hyvärinen & Oja, 2000; Bingham et al., 2015), but a typical one is $\tanh$. When $\boldsymbol{\Sigma}$ is estimated from the data, it is beneficial to use $a\tanh(z/a)$ with $a \geq 0$ instead of $\tanh$. The reason for this is that we want $\mathbb{E}(a\tanh(z/a)) \approx \mathbb{E}(z)$, but this would be difficult to achieve because, given $z$ is standardised, $\tanh$ would squash close to 1 any value greater than $2\sigma$, affecting the stationary point (see Appendix E.1). There is also the option of setting $a$ to be trainable, or using a crude approximation $\max(-a, \min(z, a))$ (see Appendix J). We can also consider an asymmetric function such as $\tanh(z)/\sqrt{\text{var}(tanh(z))}$.

### 3.4 MEAN AND VARIANCE OF DATA

Mean centring is an important part of PCA (Diamantaras & Kung, 1996; Hyvärinen & Oja, 2000; Jolliffe & Cadima, 2016). So far, we have assumed that the data $\mathbf{x}$ is centred; if it is not, then it can easily be centred as a pre-processing step or during training. If the latter, we can explicitly write the loss function in Eq. 4 as

$$\mathcal{L}(\mathbf{W}) = \mathbb{E}(||\mathbf{x} - \boldsymbol{\mu}_x - (h(\frac{(\mathbf{x} - \boldsymbol{\mu}_x)\mathbf{W}}{\boldsymbol{\sigma}})\boldsymbol{\sigma})[\mathbf{W}^T]_{sg}||_2^2), \tag{22}$$

where $\boldsymbol{\mu}_x = \mathbb{E}(\mathbf{x})$ and $\boldsymbol{\sigma}^2 = \text{var}(\mathbf{x}\mathbf{W})$. The mean and variance can be estimated on a batch of data, or, as is done with batch normalisation (Ioffe & Szegedy, 2015), estimated using exponential moving averages (Appendix I). See Appendix E.8 for a non-centred variant.

### 3.5 ORDERING OF THE COMPONENTS

After minimising the nonlinear PCA loss, we can order the components based on $\mathbf{\Sigma}$. It is also possible to automatically induce the order based on index position. One such method is to introduce the projective deflation-like term $P(\mathbf{W}) = I - \unrhd(\mathbf{W}^T[\mathbf{W}]_{sg})$ into Eq. 12 as follows

$$\mathcal{L}(\mathbf{W}) = \mathbb{E}(||\mathbf{x} - h(\mathbf{x}\mathbf{W}P(\mathbf{W})\mathbf{\Sigma}^{-1})\mathbf{\Sigma}[\mathbf{W}^T]_{sg}||_2^2), \tag{23}$$

where $\unrhd$ is lower triangular without the diagonal. See Appendix E.7 for details and other methods.

### 3.6 AN ASYMMETRIC LINEAR PCA LOSS

In Eq. 18 and 19, we see that $\mathbf{y}^T\mathbf{y}$ is symmetric, while $\mathbf{x}^T\hat{\mathbf{x}}$ is not. This is why we obtain axis-aligned components with the latter, but not the former. So to obtain axis-aligned components with linear PCA, we simply need to break the symmetry. The generalised Hebbian algorithm (Sanger, 1989) breaks the symmetry by replacing $\mathbf{y}^T\mathbf{y}$ with its lower triangular $\triangle(\mathbf{y}^T\mathbf{y})$. We can also break the symmetry by introducing $\mathbf{\Lambda} = \mathrm{diag}(\lambda_1, ..., \lambda_k)$, a linearly-spaced diagonal matrix such that $1 \geq \lambda_1 > ... > \lambda_k > 0$ to obtain

$$\Delta\mathbf{W} \propto \mathbf{x}^T\mathbf{y} - \mathbf{W}(\mathbf{y}\mathbf{\Lambda}^{\frac{1}{2}})^T\mathbf{y}\mathbf{\Lambda}^{-\frac{1}{2}}. \tag{24}$$

This is simply a unit-norm-preserving variant of the weighted subspace algorithm (Oja, 1992a; Oja et al., 1992). Another unit-norm-preserving variant that we can propose as a loss function is

$$\mathcal{L}(\mathbf{W}) = \mathbb{E}(||\mathbf{x}[\mathbf{W}]_{sg}\mathbf{W}^T - \mathbf{x}||_2^2 + ||\mathbf{W}[\hat{\mathbf{\Sigma}}]_{sg}\mathbf{\Lambda}^{\frac{1}{2}}||_2^2 - ||\mathbf{W}[\hat{\mathbf{\Sigma}}]_{sg}||_2^2 - ||\mathbf{x}\mathbf{W}\mathbf{\Lambda}^{\frac{1}{2}}||_2^2 + ||\mathbf{x}\mathbf{W}||_2^2), \tag{25}$$

where $\hat{\mathbf{\Sigma}}$ is a diagonal matrix of the estimated batch standard deviations. We see that this loss combines reconstruction, weighted regularisation, and weighted variance maximisation. See Appendix A.3.2 for derivation. We list a few other linear PCA symmetry-breaking methods in Appendix A.3.

## 4 EXPERIMENTS

We applied our nonlinear PCA model on image patches and time signals (see Appendix F.1 for additional experiments). We optimised using gradient descent (see Appendix G for training details).

**Image patches**  We evaluated the proposed nonlinear PCA model on patches extracted from CIFAR-10 (Krizhevsky et al., 2009). We extracted $11 \times 11$px overlapping patches, with zero padding, from a random subset of 500 images, resulting in a total of 512K patches. We compared either leaving $\mathbf{\Sigma}$ trainable or estimating it directly from the data. For the latter, we used $a\tanh(z/a)$ with $a = 4$, though $a \in [3, 14]$ also worked well (see Appendix F.1.3). We applied linear PCA, nonlinear PCA, and FastICA (Hyvarinen, 1999), and summarised the results in Fig. 2. We see that the PCA filters appear like blurred combinations of the filters found by nonlinear PCA. That is because a lot of the principal components from natural images have similar variances (Hyvärinen et al., 2009), which means that linear PCA would be unable to separate them. We obtained unrecognisable filters both when using the conventional nonlinear PCA loss (Fig. 2g) and when using nonlinear PCA with the contribution of the decoder included (Fig. 2f).

**Time signals**  Similar to a scikit-learn (Pedregosa et al., 2011) linear ICA example, we generated three noisy signals: sinusoidal, square, and sawtooth. The last two had the same variance, and we mixed all three using a random *orthogonal* mixing matrix. We estimated the variance from data and used $a\tanh(z/a)$ with $a = 0.8$. Figure 3 shows the result of applying linear and nonlinear PCA. As expected, linear PCA separated the sinusoidal signal, but not the square and sawtooth signals because they had the same variance. Nonlinear PCA recovered the three signals and their variances, up to sign indeterminacies. Next we considered mixing the three signals, this time with distinct variances, using

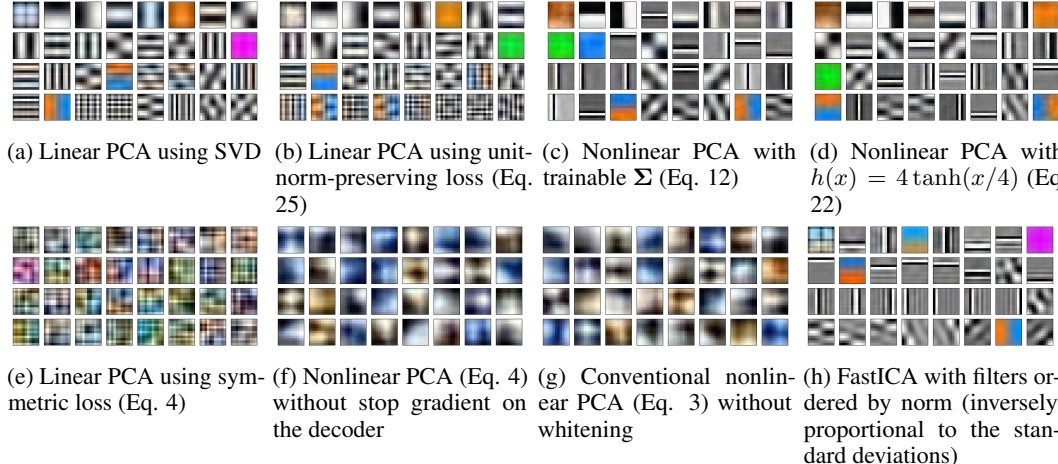

(a) Linear PCA using SVD

(b) Linear PCA using unit-norm-preserving loss (Eq. 25)

(c) Nonlinear PCA with trainable $\Sigma$ (Eq. 12)

(d) Nonlinear PCA with $h(x) = 4\tanh(x/4)$ (Eq. 22)

(e) Linear PCA using symmetric loss (Eq. 4)

(f) Nonlinear PCA (Eq. 4) without stop gradient on the decoder

(g) Conventional nonlinear PCA (Eq. 3) without whitening

(h) FastICA with filters ordered by norm (inversely proportional to the standard deviations)

Figure 2: A set of 32 11x11px filters obtained on patches from the CIFAR-10 dataset. We obtain similar filters with the proposed unit-norm-preserving linear PCA loss (b) as the ones obtained via SVD (a). The filters obtained by nonlinear PCA (c,d) seem to have further separated the mixed filters from linear PCA. In particular, this is obvious by looking at the vertical and horizontal line filters at different positions that have been unmixed with nonlinear PCA. We obtain similar filters with either having $\Sigma$ trainable (c) or estimated from data (d). We obtain indistinct filters in the PCA subspace with the symmetric linear PCA loss (e). We see in (f) that including the contribution of the decoder results in indistinct filters. This is similarly the case with conventional nonlinear PCA without whitening (g). FastICA (h) relaxes the orthogonality assumption of the overall transformation; we see that nonetheless there is an overlap with some nonlinear PCA filters, indicating that the mixing is mostly orthogonal.

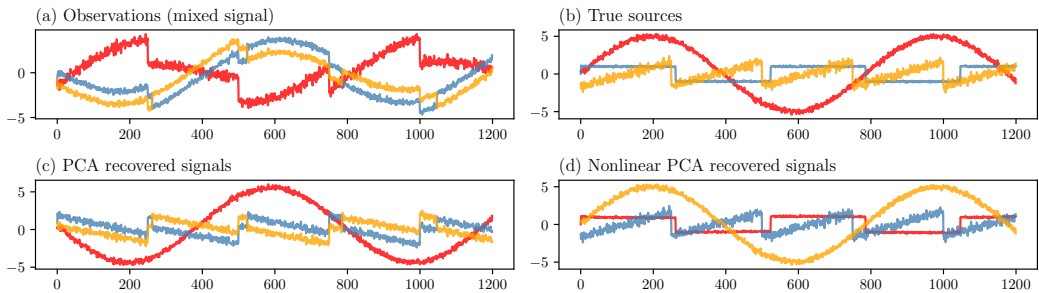

Figure 3: Three signals (sinusoidal, square, and sawtooth) that were mixed with an *orthogonal* mixing matrix. Linear PCA separated the sinusoidal signal as it had a distinct variance, but was not able to separate the square and the sawtooth signals as they have the same variance. Nonlinear PCA separated the signals and recovered their variances.

a non-orthogonal mixing matrix. We summarise the results in Fig. 4. As expected, both single-layer linear and nonlinear PCA were not able to recover the signals. Only FastICA and the two-layer nonlinear PCA model recovered the signals, but without their variances.

## 5 RELATED WORK

Hyvärinen (2015) previously proposed a unified model of linear PCA and linear ICA as a theoretical framework from the probabilistic paradigm, where the variances of the components are modelled as separate parameters but are eventually integrated out. Our key idea is the same: we model the variances as separate parameters, except that we do not integrate them out, and we flesh out the key role they play, from the neural paradigm, for nonlinear PCA.

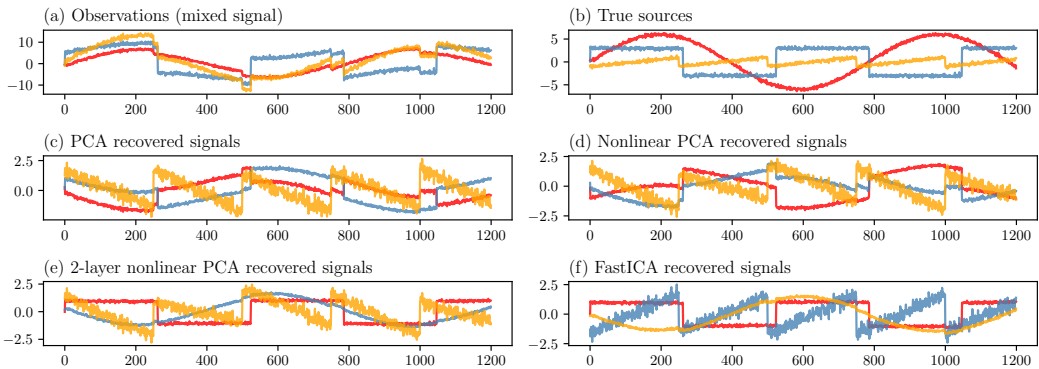

Figure 4: Three signals (sinusoidal, square, and sawtooth) that were mixed with a *non-orthogonal* mixing matrix. As expected, both single-layer linear (c) and nonlinear PCA (d) are not able to recover the signals. Only FastICA (f) and the two-layer nonlinear PCA model (e) have recovered the signals, but without their variances.

Neural PCA was started with the seminal work of Oja (1982) which showed that the first principal component can be extracted with a Hebbian (Hebb, 1949) learning rule. This spurred a lot of interest in PCA neural networks in the 80s and 90s, resulting in different variants and extensions for the extraction of multiple components (Foldiak, 1989; Oja, 1989; Rubner & Schulten, 1990; Kung & Diamantaras, 1990; Oja, 1992b; Bourlard & Kamp, 1988; Baldi & Hornik, 1989; Oja, 1992a; 1995) (see Diamantaras & Kung (1996) for an overview), some of which extracted the axis-aligned solution while others the subspace solution. A linear autoencoder with a mean squared error reconstruction loss was shown to extract the subspace solution (Bourlard & Kamp, 1988; Baldi & Hornik, 1989).

Unlike in the linear case, the term nonlinear PCA has been applied broadly (Hyvärinen & Oja, 2000) to single-layer (Xu, 1993; Karhunen & Joutsensalo, 1994; Oja, 1995), multi-layer (Kramer, 1991; Oja, 1991; Scholz & Vigário, 2002), and kernel-based (Schölkopf et al., 1998) variants. As a single-layer autoencoder, the main emphasis has been its close connection to linear ICA, especially with whitened data (Karhunen et al., 1997; 1998; Hyvärinen & Oja, 2000). Linear ICA is a more powerful extension to PCA in that it seeks to find components that are non-Gaussian and statistically independent, with the overall transformation not necessarily orthogonal. Many algorithms have been proposed (see Hyvärinen & Oja (2000); Bingham et al. (2015) for an exhaustive overview), with the most popular being FastICA (Hyvarinen, 1999). Reconstruction ICA (RICA) (Le et al., 2011) is an autoencoder formulation which combines a linear reconstruction loss with $L_1$ sparsity; although it was proposed as a method for learning overcomplete ICA features, we show that RICA, in fact, is simply performing nonlinear PCA, except that it does not preserve unit norm (see Appendix E.4).

## 6    DISCUSSION AND CONCLUSION

We have proposed $\sigma$-PCA, a unified neural model for linear and nonlinear PCA. This model allows nonlinear PCA to retain PCA characteristics: orthogonal transformation and estimable variances. Without whitening, this is made possible by modelling the variances as separate parameters and by putting an emphasis not on input reconstruction but on latent reconstruction. And so, in a tied nonlinear single-layer autoencoder, what matters is not the decoder contribution but the encoder contribution. This is in contrast to the conventional nonlinear PCA formulation (Karhunen & Joutsensalo, 1994) where the encoder contribution was omitted and the variance parameters were missing. Although the nonlinear PCA solution implicitly exists in the ICA solution space, it is not necessarily explicitly recovered as ICA seeks to find a transformation that maximises independence by fitting an over-parametrised arbitrary transformation of the form $\mathbf{W}\mathbf{\Sigma}^{-1}\mathbf{V}$ while nonlinear PCA explicitly restricts it to an orthogonal transformation of the form $\mathbf{W}\mathbf{\Sigma}^{-1}$. The benefit of nonlinear PCA in our $\sigma$-PCA formulation is that it can be applied wherever linear PCA would be applied, but, unlike linear PCA, it can further separate components that have the same variance.

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

## APPENDIX

## A    LINEAR PCA

In this section we go through different methods related to PCA and highlight their loss functions and weight updates where relevant. There are a variety of other neural methods for PCA; see (Diamantaras & Kung, 1996) for an in-depth review.

## A.1 PCA AS EIGENDECOMPOSITION OF COVARIANCE MATRIX OR SVD OF DATA MATRIX

Let $\mathbf{X} \in \mathbb{R}^{n \times p}$ be the *centred* data matrix and let $\mathbf{C} = \frac{1}{n}\mathbf{X}^T\mathbf{X}$ be the symmetric covariance matrix, then we can write the SVD of $\mathbf{X}$ as

$$\mathbf{X} = \mathbf{U}\mathbf{S}\mathbf{V}^T, \tag{26}$$

where $\mathbf{U} \in \mathbb{R}^{n \times p}$ is a semi-orthogonal matrix, $\mathbf{S} \in \mathbb{R}^{p \times p}$ is a diagonal matrix of singular values, and $\mathbf{V} \in \mathbb{R}^{p \times p}$ is an orthogonal matrix; and we can write the eigendecomposition of $\mathbf{C}$ as

$$\mathbf{C} = \mathbf{E}\mathbf{\Lambda}\mathbf{E}^T, \tag{27}$$

where $\mathbf{E} \in \mathbb{R}^{p \times p}$ is an orthogonal matrix of eigenvectors corresponding to the principal axes, and $\mathbf{\Lambda} \in \mathbb{R}^{p \times p}$ is a diagonal matrix corresponding to the variances.

We can relate both by noting that we have

$$\mathbf{C} = \frac{1}{n}\mathbf{X}^T\mathbf{X} = \frac{1}{n}\mathbf{V}\mathbf{S}^T\mathbf{U}^T\mathbf{U}\mathbf{S}\mathbf{V}^T \tag{28}$$

$$\mathbf{E}\mathbf{\Lambda}\mathbf{E}^T = \mathbf{V}\frac{1}{n}\mathbf{S}^2\mathbf{V}^T. \tag{29}$$

From this we can see that $\mathbf{E} = \mathbf{V}$ and $\mathbf{\Lambda} = \frac{1}{n}\mathbf{S}^2 = \mathrm{diag}(\boldsymbol{\sigma}^2)$.

## A.2 PCA SUBSPACE SOLUTIONS

### A.2.1 LINEAR AUTOENCODER

Let $\mathbf{X}$ be the centred data matrix and $\mathbf{x}$ a row of $\mathbf{X}$, then we can write the reconstruction loss of a linear autoencoder as

$$\mathcal{L}(\mathbf{W}_e, \mathbf{W}_d) = \frac{1}{2}\mathbb{E}(||\mathbf{x}\mathbf{W}_e\mathbf{W}_d^T - \mathbf{x}||_2^2), \tag{30}$$

where $\mathbf{W}_e \in \mathbb{R}^{p \times k}$ is the encoder and $\mathbf{W}_d^T \mathbb{R}^{p \times k}$ is the decoder. Without loss of generality, we omit the expectation, the loss yields the following gradients:

$$\frac{\partial \mathcal{L}}{\partial \mathbf{W}_e} = \mathbf{x}^T(\mathbf{x}\mathbf{W}_e\mathbf{W}_d^T - \mathbf{x})\mathbf{W}_d \tag{31}$$

$$\frac{\partial \mathcal{L}}{\partial \mathbf{W}_d} = (\mathbf{x}\mathbf{W}_e\mathbf{W}_d^T - \mathbf{x})^T\mathbf{x}\mathbf{W}_e. \tag{32}$$

Let us consider the tied weights case, i.e. $\mathbf{W}_e = \mathbf{W}_d = \mathbf{W}$, but keep the contributions separate. Let $\mathbf{y} = \mathbf{x}\mathbf{W}$, we can write

$$\frac{\partial \mathcal{L}}{\partial \mathbf{W}_e} = \mathbf{x}^T(\mathbf{x}\mathbf{W}\mathbf{W}^T - \mathbf{x})\mathbf{W} \tag{33}$$

$$= \mathbf{x}^T\mathbf{x}\mathbf{W}(\mathbf{W}^T\mathbf{W} - \mathbf{I}) \tag{34}$$

$$= \mathbf{x}^T\mathbf{y}(\mathbf{W}^T\mathbf{W} - \mathbf{I}) \tag{35}$$

$$\frac{\partial \mathcal{L}}{\partial \mathbf{W}_d} = (\mathbf{x}\mathbf{W}\mathbf{W}^T - \mathbf{x})^T\mathbf{x}\mathbf{W} \tag{36}$$

$$= (\mathbf{x}\mathbf{W}\mathbf{W}^T)^T\mathbf{x}\mathbf{W} - \mathbf{x}^T\mathbf{x}\mathbf{W} \tag{37}$$

$$= \mathbf{W}\mathbf{W}^T\mathbf{x}^T\mathbf{x}\mathbf{W} - \mathbf{x}^T\mathbf{x}\mathbf{W} \tag{38}$$

$$= (\mathbf{W}\mathbf{W}^T - \mathbf{I})\mathbf{x}^T\mathbf{x}\mathbf{W} \tag{39}$$

$$= (\mathbf{W}\mathbf{W}^T - \mathbf{I})\mathbf{x}^T\mathbf{y} \tag{40}$$

We can note that the term $-\mathbf{x}^T\mathbf{y}$ appears in both the contributions of the encoder and the decoder, and this term is a variance maximisation term. Indeed, given that the data is centred, we can write the variance maximisation loss as

$$\mathcal{J}(\mathbf{W}) = -\frac{1}{2}\mathbb{E}(\|\mathbf{x}\mathbf{W}\|_2^2) = -\frac{1}{2}\mathbb{E}(\|\mathbf{y}\|_2^2), \tag{41}$$

which has the following gradient (omitting the expectation)

$$\frac{\partial \mathcal{J}}{\partial \mathbf{W}} = -\mathbf{x}^T\mathbf{x}\mathbf{W} = -\mathbf{x}^T\mathbf{y}. \tag{42}$$

Thus, the encoder multiplies the variance maximisation term with $(\mathbf{W}^T\mathbf{W} - \mathbf{I})$ and the decoder with $(\mathbf{W}\mathbf{W}^T - \mathbf{I})$. If $\mathbf{W}$ is constrained to be orthogonal, i.e. $\mathbf{W}^T\mathbf{W} = \mathbf{I}$, then the contribution from the encoder is zero, and the contribution from the decoder is dominate. We can also note that the variance maximisation of the latent results in a Hebbian update rule $\Delta\mathbf{W} \propto \mathbf{x}^T\mathbf{y}$ (Oja, 1982).

Let $\hat{\mathbf{x}} = \mathbf{y}\mathbf{W}^T$ and $\hat{\mathbf{y}} = \mathbf{x}\mathbf{W}$, then we can rewrite the contributions as

$$\frac{\partial \mathcal{L}}{\partial \mathbf{W}_e} = \mathbf{x}^T(\hat{\mathbf{y}} - \mathbf{y}) \tag{43}$$

$$\frac{\partial \mathcal{L}}{\partial \mathbf{W}_d} = (\hat{\mathbf{x}} - \mathbf{x})^T\mathbf{y}, \tag{44}$$

highlighting that the encoder puts an emphasis on latent reconstruction while the decoder puts an emphasis on input reconstruction. Rewritten as weight updates, we have

$$\Delta\mathbf{W}_e \propto \mathbf{x}^T\mathbf{y} - \mathbf{x}^T\mathbf{x}\mathbf{W} \tag{45}$$

$$\Delta\mathbf{W}_d \propto \mathbf{x}^T\mathbf{y} - \mathbf{W}\mathbf{y}^T\mathbf{y}. \tag{46}$$

### A.2.2 SUBSPACE LEARNING ALGORITHM

The subspace learning algorithm (Oja, 1983; Williams & University of California, 1985; Oja, 1989; Hyvärinen & Oja, 2000) was proposed as a generalisation of Oja's single component neural learning algorithm (Oja, 1982). The weight update has the following form:

$$\Delta\mathbf{W} = \mathbf{x}^T\mathbf{x}\mathbf{W} - \mathbf{W}(\mathbf{W}^T\mathbf{x}^T\mathbf{x}\mathbf{W}) \tag{47}$$

$$= \mathbf{x}^T\mathbf{y} - \mathbf{W}\mathbf{y}^T\mathbf{y}. \tag{48}$$

We see that this update is exactly the same as Eq. 46: it is simply the decoder contribution from a linear reconstruction loss. As we have seen from Eq. 34, the contribution of the encoder is zero if $\mathbf{W}$ is semi-orthogonal, so in the subspace learning algorithm the encoder contribution is simply omitted. The subspace learning algorithm can be obtained as a reconstruction loss with the stop gradient operator placed on the encoder:

$$\mathcal{L}(\mathbf{W}) = \frac{1}{2}||\mathbf{x}[\mathbf{W}]_{sg}\mathbf{W}^T - \mathbf{x}||_2^2. \tag{49}$$

### A.3 FINDING AXIS-ALIGNED PRINCIPAL VECTORS

Here we look at the main idea for obtaining axis-aligned solutions: symmetry breaking. We saw in the previous section that the weight updates in the linear case are symmetric – no particular component is favoured over the other. Therefore, any method that breaks the symmetry will tend to lead to the axis-aligned PCA solution.

#### A.3.1 WEIGHTED SUBSPACE ALGORITHM

One straightforward way to break the symmetry is to simply weigh each component differently. This has been shown to converge (Oja, 1992a; Oja et al., 1992; Xu, 1993; Hyvärinen & Oja, 2000) to the PCA eigenvectors.

The weighted subspace algorithm does exactly that, and it modifies the update rule of the subspace learning algorithm into

$$\Delta\mathbf{W} = \mathbf{x}^T\mathbf{y} - \mathbf{W}\mathbf{y}^T\mathbf{y}\mathbf{\Lambda}^{-1}, \tag{50}$$

$$\text{or } \Delta\mathbf{W} = \mathbf{x}^T\mathbf{y}\mathbf{\Lambda} - \mathbf{W}\mathbf{y}^T\mathbf{y}. \tag{51}$$

where $\mathbf{\Lambda} = \mathrm{diag}(\lambda_1, ..., \lambda_k)$ such that $\lambda_1 > ... > \lambda_k > 0$.

The second form can be written as a loss function:

$$\mathcal{L}(\mathbf{W}) = \frac{1}{2}\mathbb{E}(||\mathbf{x}[\mathbf{W}]_{sg}\mathbf{W}^T - \mathbf{x}||_2^2 - ||\mathbf{x}\mathbf{W}\mathbf{\Lambda}^{\frac{1}{2}}||_2^2 + ||\mathbf{x}\mathbf{W}||_2^2). \tag{52}$$

#### A.3.2 WEIGHTED SUBSPACE ALGORITHM WITH UNIT NORM

Although the above updates (Eq. 50 and 51) do converge to the eigenvectors, they no longer maintain unit norm columns. To see why, we can look at the stationary point, where we know that $\Delta\mathbf{W}$ should be $\mathbf{0}$ and that the components of $\mathbf{y}$ should be uncorrelated, i.e. $\mathbb{E}(\mathbf{y}^T\mathbf{y}) = \hat{\mathbf{\Sigma}}$ is diagonal. We can write:

$$\Delta\mathbf{W} = \mathbf{0} \tag{53}$$

$$\mathbb{E}(\mathbf{x}^T\mathbf{y} - \mathbf{W}\mathbf{y}^T\mathbf{y}\mathbf{\Lambda}^{-1}) = \mathbf{0} \tag{54}$$

$$\mathbb{E}(\mathbf{W}^T\mathbf{x}^T\mathbf{y} - \mathbf{W}^T\mathbf{W}\mathbf{y}^T\mathbf{y}\mathbf{\Lambda}^{-1}) = \mathbf{0} \tag{55}$$

$$\mathbb{E}(\mathbf{y}^T\mathbf{y}) - \mathbf{W}^T\mathbf{W}\mathbb{E}(\mathbf{y}^T\mathbf{y})\mathbf{\Lambda}^{-1} = \mathbf{0} \tag{56}$$

$$\hat{\mathbf{\Sigma}}^2 - \mathbf{W}^T\mathbf{W}\hat{\mathbf{\Sigma}}^2\mathbf{\Lambda}^{-1} = \mathbf{0} \tag{57}$$

$$\mathbf{W}^T\mathbf{W}\mathbf{\Lambda}^{-1}\hat{\mathbf{\Sigma}}^2 = \hat{\mathbf{\Sigma}}^2 \tag{58}$$

$$\mathbf{W}^T\mathbf{W} = \mathbf{\Lambda}. \tag{59}$$

As we know $\mathbf{W}$ has orthogonal columns, $\mathbf{W}^T\mathbf{W}$ must be diagonal, and so the norm of each $i$th column becomes equal to $\sqrt{\lambda_i}$. If we want unit norm columns, a straightforward remedy is to normalise at the end of training. A closer look allows us to see that the main contributor to the norm is the diagonal part of $\mathbf{y}^T\mathbf{y}\mathbf{\Lambda}^{-1}$ – this suggests to us two options that could maintain unit norm columns.

As a first option, we can simply counteract the effect of $\mathbf{\Lambda}^{-1}$ on the diagonal by multiplying by its inverse on the other side of $\mathbf{y}^T\mathbf{y}$ to obtain

$$\Delta\mathbf{W} = \mathbf{x}^T\mathbf{y} - \mathbf{W}(\mathbf{y}\mathbf{\Lambda})^T\mathbf{y}\mathbf{\Lambda}^{-1}, \tag{60}$$

$$\text{or } \Delta\mathbf{W} = \mathbf{x}^T\mathbf{y}\mathbf{\Lambda} - \mathbf{W}(\mathbf{y}\mathbf{\Lambda})^T\mathbf{y}, \tag{61}$$

for which we now have

$$\Delta\mathbf{W} = \mathbf{0} \tag{62}$$

$$\hat{\mathbf{\Sigma}}^2 - \mathbf{W}^T\mathbf{W}\mathbf{\Lambda}\hat{\mathbf{\Sigma}}^2\mathbf{\Lambda}^{-1} = \mathbf{0} \tag{63}$$

$$\mathbf{W}^T\mathbf{W}\mathbf{\Lambda}\mathbf{\Lambda}^{-1}\hat{\mathbf{\Sigma}}^2 = \hat{\mathbf{\Sigma}}^2 \tag{64}$$

$$\mathbf{W}^T\mathbf{W} = \mathbf{I}. \tag{65}$$

As a second option, we can remove the diagonal part of $\mathbf{y}^T\mathbf{y}\mathbf{\Lambda}^{-1}$. This consists in adding

$$-\mathbf{W}(\text{diag}(\text{diag}^{-1}(\mathbf{y}^T\mathbf{y})) - \text{diag}(\text{diag}^{-1}(\mathbf{y}^T\mathbf{y}\mathbf{\Lambda}^{-1}))) \tag{66}$$

to Eq. 50, or

$$-\mathbf{W}(\text{diag}(\text{diag}^{-1}(\mathbf{y}^T\mathbf{y}\mathbf{\Lambda})) - \text{diag}(\text{diag}^{-1}(\mathbf{y}^T\mathbf{y}))) \tag{67}$$

to Eq. 51. The latter can be derived from the gradient of

$$\frac{1}{2}(||\mathbf{W}[\hat{\mathbf{\Sigma}}]_{sg}\mathbf{\Lambda}^{\frac{1}{2}}||_2^2 - ||\mathbf{W}[\hat{\mathbf{\Sigma}}]_{sg}||_2^2). \tag{68}$$

We thus arrive at a total loss that maintains unit norm columns:

$$\mathcal{L}(\mathbf{W}) = \frac{1}{2}\mathbb{E}(||\mathbf{x}[\mathbf{W}]_{sg}\mathbf{W}^T - \mathbf{x}||_2^2 + ||\mathbf{W}[\hat{\mathbf{\Sigma}}]_{sg}\mathbf{\Lambda}^{\frac{1}{2}}||_2^2 - ||\mathbf{W}[\hat{\mathbf{\Sigma}}]_{sg}||_2^2 - ||\mathbf{x}\mathbf{W}\mathbf{\Lambda}^{\frac{1}{2}}||_2^2 + ||\mathbf{x}\mathbf{W}||_2^2) \tag{69}$$

We see that this combines reconstruction, weighted regularisation, and weighted variance maximisation. We can also note from this that $\lambda_i$ should be $\leq 1$ to avoid the variance maximisation term overpowering the other terms.

### A.3.3 GENERALISED HEBBIAN ALGORITHM (GHA)

The generalised Hebbian algorithm (GHA) (Sanger, 1989) learns multiple PCA components by combining Oja's update rule (Oja, 1982) with a Gram-Schmidt-like orthogonalisation term. It breaks the symmetry in the subspace learning algorithm by taking the lower triangular part of $\mathbf{y}^T\mathbf{y}$, allowing the weights to converge to the true PCA eigenvectors.

$$\mathbf{y} = \mathbf{x}\mathbf{W} \tag{70}$$

$$\Delta\mathbf{W} \propto \mathbf{x}^T\mathbf{y} - \mathbf{W}_{\triangle}(\mathbf{y}^T\mathbf{y}) \tag{71}$$

One way to write this as a loss function is to use variance maximisation and the stop gradient operator to get the following:

$$\mathcal{L}(\mathbf{W}) = -\frac{1}{2}||\mathbf{x}\mathbf{W}||_2^2 + \mathbf{1}\mathbf{W} \odot [\mathbf{W}_{\triangle}(\mathbf{y}^T\mathbf{y})]_{sg}\mathbf{1}^T. \tag{72}$$

### A.3.4 AUTOENCODER WITH GHA UPDATE

We can also consider combining the orthogonalisation term of GHA with the autoencoder reconstruction to get

$$\mathcal{L}(\mathbf{W}) = ||\mathbf{x} - \mathbf{x}\mathbf{W}\mathbf{W}^T||_2^2 + \mathbf{1}\mathbf{W} \odot [\mathbf{W}_{\unlhd}(\mathbf{y}^T\mathbf{y})]_{sg}\mathbf{1}^T, \tag{73}$$

where $\unlhd$ is the operation that takes the lower triangular without the diagonal; otherwise, the term $-\mathbf{W}\mathrm{diag}(\mathrm{diag}^{-1}(\mathbf{y}^T\mathbf{y}))$ would be doubled unless we also include another $\mathbf{x}^T\mathbf{y}$ to counteract it. The double term might still work in practice, but it becomes detrimental when combined with a projective unit norm constraint. If $\mathbf{W}$ is orthogonal, then this is similar to combining the subspace update rule (Eq. 48) with the orthogonalisation term of GHA to result in

$$\Delta\mathbf{W} = \mathbf{x}^T\mathbf{y} - \mathbf{W}\mathbf{y}^T\mathbf{y} - \mathbf{W}_{\unlhd}(\mathbf{y}^T\mathbf{y}), \tag{74}$$

which can be obtained from the following loss

$$\mathcal{L}(\mathbf{W}) = ||\mathbf{x} - \mathbf{x}[\mathbf{W}]_{sg}\mathbf{W}^T||_2^2 + \mathbf{1}\mathbf{W} \odot [\mathbf{W}_{\unlhd}(\mathbf{y}^T\mathbf{y})]_{sg}\mathbf{1}^T. \tag{75}$$

Nonetheless, keeping the encoder contribution can still help maintain the orthogonality of $\mathbf{W}$, as the term $\mathbf{W}^T\mathbf{W} - \mathbf{I}$ is similar to the one derived from a symmetric orthogonalisation regulariser (see Appendix B).

Another variant we can consider is to make triangular the full gradient of the linear autoencoder update, which includes not just the contribution of the decoder but also that of the encoder, i.e. by taking

$$\frac{\partial\mathcal{L}}{\partial\mathbf{W}} = \mathbf{x}^T\mathbf{y}(\mathbf{W}^T\mathbf{W} - \mathbf{I}) - \mathbf{x}^T\mathbf{y} + \mathbf{W}\mathbf{y}^T\mathbf{y} \tag{76}$$

and changing it into

$$\frac{\partial\mathcal{L}}{\partial\mathbf{W}} = \mathbf{x}^T\mathbf{y}(\unlhd(\mathbf{W}^T\mathbf{W}) - \mathbf{I}) - \mathbf{x}^T\mathbf{y} + \mathbf{W}_{\unlhd}(\mathbf{y}^T\mathbf{y}). \tag{77}$$

Though the contribution from the encoder is negligible when $\mathbf{W}$ is close to orthogonal, the term $\unlhd\mathbf{W}^T\mathbf{W} - \mathbf{I}$ also acts as an approximation to Gram-Schmidt orthogonalisation (see Appendix B), and so when $\mathbf{W}$ deviates from being orthogonal, it can help bring it back more quickly.

### A.3.5 NESTED DROPOUT

Another way to break the symmetry and enforce an ordering is nested dropout (Rippel et al., 2014), a procedure for randomly removing nested sets of components. It can be shown (Rippel et al., 2014) that it leads to an exact equivalence with PCA in the case of a single-layer linear autoencoder. An idea similar in spirit was the previously proposed hierarchical PCA method (Scholz & Vigário, 2002) for ordering of the principal components; however, it was limited in number of dimensions and was not stochastic. Nested dropout works by assigning a prior distribution $p(.)$ over the component indices $1...k$, and then using it to sample an index $\mathcal{L}$ and drop the units $\mathcal{L} + 1, ..., k$. A typical choice of distribution is a geometric distribution with $p(j) = \rho^{j-1}(1 - \rho), 0 < \rho < 1$.

Let $\mathbf{m}_{1|j} \in \{0, 1\}^k$ be a vector such that $m_i = \begin{cases} 1 & \text{if } i < j \\ 0 & \text{otherwise} \end{cases}$, then the reconstruction loss becomes

$$\mathcal{L}(\mathbf{W}) = \frac{1}{2}\mathbb{E}(||(\mathbf{x}\mathbf{W} \odot \mathbf{m}_{1|j})\mathbf{W}^T - \mathbf{x}||_2^2), \tag{78}$$

where the mask $\mathbf{m}_{1|j}$ is randomly sampled during each update step. One issue with nested dropout is that the higher the index the smaller the gradient update. This means that training has to run much longer for them to converge. One suggested remedy for this (Rippel et al., 2014) is to perform gradient sweeping, where once an earlier index has converged, it can be frozen and the nested dropout index can be incremented.

A non-stochastic version, which is an extension of the hierarchical PCA method (Scholz & Vigário, 2002), sums all the reconstruction loss to result in

$$\mathcal{L}(\mathbf{W}) = \frac{1}{2} \sum_j \mathbb{E}(||\mathbf{x}\mathbf{W}_{1|j}\mathbf{W}_{1|j}^T - \mathbf{x}||_2^2), \tag{79}$$

where $\mathbf{W}_{1|j}$ is the truncated matrix that contains the first $j$ columns of $\mathbf{W}$. However, this can get more computationally demanding when there is a large number of components compared to the stochastic version.

### A.3.6 WEIGHTED VARIANCE MAXIMISATION

We can formulate a variant of the weighted subspace algorithm more explicitly as a reconstruction loss combined with a weighted variance maximisation term:

$$\mathcal{L}(\mathbf{W}) = \frac{1}{2}\mathbb{E}(||\mathbf{x} - \mathbf{x}\mathbf{W}\mathbf{W}^T||_2^2 - \alpha||\mathbf{x}\mathbf{W}\mathbf{\Lambda}^{\frac{1}{2}}||_2^2), \tag{80}$$

where $\mathbf{\Lambda} = \mathrm{diag}(\lambda_1, ..., \lambda_k)$ consists of linearly spaced values between 0 and 1, and $\alpha \in \{-1, 1\}$. We can also pick $\mathbf{\Lambda}$ to be proportional to the variances, but the term should not be trainable – we can do that with the stop gradient operator: $\mathbf{\Lambda} = [\frac{\mathbb{E}(\mathbf{y}^2)}{\max_i(\mathbb{E}(y_i^2))}]_{sg}$.

The associated gradient of the loss is

$$\frac{\partial \mathcal{L}}{\partial \mathbf{W}} = -\mathbf{x}^T\mathbf{y} + \mathbf{W}\mathbf{y}^T\mathbf{y} + \mathbf{x}^T\mathbf{y}(\mathbf{W}^T\mathbf{W} - \mathbf{I}) - \alpha\mathbf{x}^T\mathbf{y}\mathbf{\Lambda} \tag{81}$$

$$= -\mathbf{x}^T\mathbf{y}(\mathbf{I} + \alpha\mathbf{\Lambda}) + \mathbf{W}\mathbf{y}^T\mathbf{y} + \mathbf{x}^T\mathbf{y}(\mathbf{W}^T\mathbf{W} - \mathbf{I}). \tag{82}$$

Following a similar analysis as in Appendix A.3.2, it can be shown that at the stationary point we have $\mathbf{W}^T\mathbf{W} = \mathbf{I} + \frac{1}{2}\alpha\mathbf{\Lambda}$.

Alternative, we can use a stochastic weighting by applying nested dropout on the variance regulariser:

$$\mathcal{L}(\mathbf{W}) = \mathbb{E}(||\mathbf{x} - \mathbf{x}\mathbf{W}\mathbf{W}^T||_2^2 - \alpha||\mathbf{x}\mathbf{W} \odot \mathbf{m}_{1|j}||_2^2), \tag{83}$$

with a randomly sampled mask per data sample.

## B ENFORCING ORTHOGONALITY

There are a few ways to enforce orthogonality either via a regulariser or a projective constraint, and this can be symmetric or asymmetric.

### B.1 SYMMETRIC

A symmetric orthogonalisation scheme can be used when there are no favoured components.

**Regulariser** A straightforward regulariser, which also enforces unit norm, is

$$J(\mathbf{W}) = \alpha||\mathbf{I} - \mathbf{W}^T\mathbf{W}||_F^2, \tag{84}$$

and it has the gradient

$$\frac{\partial J}{\partial \mathbf{W}} = 4\alpha \mathbf{W}(\mathbf{W}^T \mathbf{W} - \mathbf{I}). \tag{85}$$

Combining the above with PCA or ICA updates has been previously referred to as the bigradient rule (Wang et al., 1995; Wang & Karhunen, 1996; Karhunen et al., 1997), where $\alpha$ is at most $\frac{1}{8}$, for which the justification can be derived from its corresponding projective constraint.

**Projective constraint**   An iterative algorithm, as used in FastICA (Hyvarinen, 1999), is the following:

1. $\mathbf{w}_i \leftarrow \dfrac{\mathbf{w}_i}{||\mathbf{w}_i||}$

2. Repeat until convergence:

$$\mathbf{W} \leftarrow \frac{3}{2}\mathbf{W} - \frac{1}{2}\mathbf{W}\mathbf{W}^T\mathbf{W}.$$

We can note that

$$\frac{3}{2}\mathbf{W} - \frac{1}{2}\mathbf{W}\mathbf{W}^T\mathbf{W} = \mathbf{W} - \frac{1}{2}\mathbf{W}(\mathbf{W}^T\mathbf{W} - \mathbf{I}) \tag{86}$$

$$= \mathbf{W} - \frac{1}{8\alpha}\frac{\partial J}{\partial \mathbf{W}}, \tag{87}$$

which allows to to see that we can set $\alpha = \frac{1}{8}$ in Eq. 85 to derive the projective constraint from the stochastic gradient descent update.

The iterative algorithm can be shown to converge (Hyvarinen, 1999; Hyvärinen & Oja, 2000) by analysing the evolution of the eigenvalues of $\mathbf{W}^T\mathbf{W}$. The initial normalisation guarantees that, prior to any iterations, the eigenvalues of $\mathbf{W}^T\mathbf{W}$ are $\leq 1$. Let $\mathbf{E}^T\boldsymbol{\Lambda}\mathbf{E}$ be the eigendecomposition of $\mathbf{W}^T\mathbf{W}$, and let us consider the general case

$$\hat{\mathbf{W}} = \mathbf{W} - \beta\mathbf{W}(\mathbf{W}^T\mathbf{W} - \mathbf{I}) \tag{88}$$

$$= (1 + \beta)\mathbf{W} - \beta\mathbf{W}\mathbf{W}^T\mathbf{W}. \tag{89}$$

After one iteration we have

$$\hat{\mathbf{W}}^T\hat{\mathbf{W}} = (1 + \beta)^2\mathbf{W}^T\mathbf{W} - 2(1 + \beta)\beta(\mathbf{W}^T\mathbf{W})^2 + \beta^2(\mathbf{W}^T\mathbf{W})^3 \tag{90}$$

$$= (1 + \beta)^2\mathbf{E}^T\boldsymbol{\Lambda}\mathbf{E} - 2(1 + \beta)\beta(\mathbf{E}^T\boldsymbol{\Lambda}\mathbf{E})^2 + \beta^2(\mathbf{E}^T\boldsymbol{\Lambda}\mathbf{E})^3 \tag{91}$$

$$= (1 + \beta)^2\mathbf{E}^T\boldsymbol{\Lambda}\mathbf{E} - 2(1 + \beta)\beta\mathbf{E}^T\boldsymbol{\Lambda}^2\mathbf{E} + \beta^2\mathbf{E}^T\boldsymbol{\Lambda}^3\mathbf{E} \tag{92}$$

$$= \mathbf{E}^T((1 + \beta)^2\boldsymbol{\Lambda} - 2(1 + \beta)\beta\boldsymbol{\Lambda}^2 + \beta^2\boldsymbol{\Lambda}^3)\mathbf{E}. \tag{93}$$

We now have the eigenvalues after one iteration as a function of the eigenvalues of the previous iteration:

$$f_\beta(\lambda) = (1 + \beta)^2\lambda - 2(1 + \beta)\beta\lambda^2 + \beta^2\lambda^3 \tag{94}$$

$$f'_\beta(\lambda) = (1 + \beta)^2 - 4(1 + \beta)\beta\lambda + 3\beta^2\lambda^2. \tag{95}$$

We know that the eigenvalues are in the interval $]0, 1]$. For $\lambda = 1$ we have

$$f_\beta(1) = 1 \tag{96}$$

$$f'_\beta(1) = (1 + \beta)^2 - 4(1 + \beta)\beta + 3\beta^2 = 1 - 2\beta. \tag{97}$$

We see that there is a stationary point at $\beta = \frac{1}{2}$. Noting that $f_0(\lambda) = \lambda$, we can plot $f_\beta$:

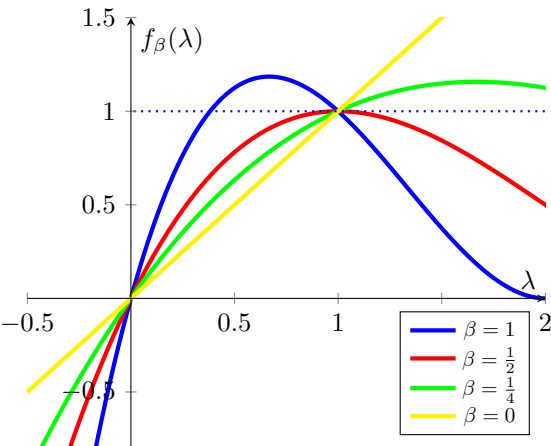

We see that if $\lambda \in ]0, 1]$ and $\beta \in ]0, \frac{1}{2}]$, then the eigenvalues will always be in $]0, 1]$; and given that $f(\lambda) > \lambda$, all the eigenvalues will eventually converge to 1.

Thus, for $\beta = \frac{1}{2}$ we get the fastest convergence where $\lambda$ does not exceed $]0, 1]$. This means that when using stochastic gradient descent with a learning rate of 1 and with a symmetric regulariser (Eq. 84), an optimal value for $\alpha$ is $\frac{1}{8}$. This analysis does not consider potential other interactions from the use of additional losses that could affect $\mathbf{W}$ or the use of an adaptive optimiser like Adam Kingma & Ba (2014).

### B.2 Asymmetric

**Regulariser** From the symmetric regulariser we can derive an asymmetric version that performs a Gram-Schmidt-like orthogonalisation. We simply need to use the lower (or upper) triangular part of $\mathbf{W}^T\mathbf{W}$ while using a stop gradient on either $\mathbf{W}$ or $\mathbf{W}^T$. This means that the loss, which also maintains unit norm, is

$$J(\mathbf{W}) = \frac{1}{2}||\unlhd(\mathbf{W}^T[\mathbf{W}]_{sg}) - \mathbf{I}||_F^2 \tag{98}$$

or without the diagonal (to remove the unit norm regulariser):

$$J(\mathbf{W}) = \frac{1}{2}||\unlhd(\mathbf{W}^T[\mathbf{W}]_{sg})||_F^2, \tag{99}$$

where $\unlhd$ refers to the lower triangular matrix without the diagonal. From this, we have

$$\frac{\partial J}{\partial \mathbf{W}} = \mathbf{W}\unlhd(\mathbf{W}^T\mathbf{W} - \mathbf{I}), \tag{100}$$

or

$$\frac{\partial J}{\partial \mathbf{W}} = \mathbf{W}\unlhd(\mathbf{W}^T\mathbf{W}). \tag{101}$$

The use of such an update has been previously referred to as the hierarchic version of the bigradient algorithm (Wang et al., 1995).

Recall that in the weight update of GHA (Appendix A.3.3) we have the term $\mathbf{W}\unlhd(\mathbf{y}^T\mathbf{y})$; the asymmetric update can benefit from being at a similar scale in order to be effective, especially when the variances are large. To do this, we can weight the loss by the non-trainable standard deviations to obtain

$$J(\mathbf{W}) = \frac{1}{2}||\unlhd(\mathbf{W}^T[\mathbf{W}\hat{\mathbf{\Sigma}}]_{sg})||_F^2. \tag{102}$$

**Gram-Schmidt projective constraint**   After each gradient update step, we can apply the Gram-Schmidt orthogonalisation procedure (Schmidt, 1907).

### B.3   ENCODER CONTRIBUTION OF LINEAR RECONSTRUCTION

As noted in Appendix A.2.1, in a linear autoencoder the contribution of the encoder to the gradient is zero when $\mathbf{W}$ is orthogonal, and so if used in isolation it can serve as an implicit orthogonal regulariser:

$$J(\mathbf{W}) = \mathbb{E}(||\mathbf{x} - \mathbf{x}\mathbf{W}[\mathbf{W}^T]_{sg}||_2^2), \tag{103}$$

having

$$\frac{\partial J}{\partial \mathbf{W}} = \mathbf{x}^T\mathbf{y}(\mathbf{W}^T\mathbf{W} - \mathbf{I}) \tag{104}$$

as gradient. We see that it simply replaces in the symmetric regulariser the first $\mathbf{W}$ with $\mathbf{x^T y}$.

## C   ENFORCING UNIT NORM

If not already taken care of by the orthogonality constraint, there are a few ways to enforce unit norm.

### C.1   PROJECTIVE CONSTRAINT

The columns $\mathbf{w}_i^T$ of $\mathbf{W}$ can be normalised to unit norm after each update step:

$$\mathbf{w}_i^T \leftarrow \frac{\mathbf{w}_i^T}{||\mathbf{w}_i^T||}. \tag{105}$$

### C.2   REGULARISATION

We can enforce unit norm by adding a regulariser on the norm of the column vectors $\mathbf{w}_i^T$. This results in the loss

$$J(\mathbf{w}_i^T) = \frac{1}{2}(\mathbf{1} - ||\mathbf{w}_i^T||_2)^2 \tag{106}$$

which has the following gradient

$$\frac{\partial J}{\partial \mathbf{w}_i^T} = (1 - \frac{1}{||\mathbf{w}_i^T||})\mathbf{w}_i^T \tag{107}$$

### C.3   DIFFERENTIABLE WEIGHT NORMALISATION

A third way to do this is to use differentiable weight normalisation (Salimans & Kingma, 2016), except without the scale parameter. This consists in replacing the column vectors with

$$\hat{\mathbf{w}}_i^T = \frac{\mathbf{w}_i^T}{||\mathbf{w}_i^T||}. \tag{108}$$

This has the following Jacobian

$$\frac{\partial \hat{\mathbf{w}}_i^T}{\partial \mathbf{w}_i^T} = \frac{1}{||\mathbf{w}_i^T||}(\mathbf{I} - \frac{\mathbf{w}_i^T\mathbf{w}_i}{||\mathbf{w}_i^T||^2}), \tag{109}$$

which results for a given loss $\mathcal{L}$ in

$$\frac{\partial \mathcal{L}}{\partial \mathbf{w}_i^T} = \frac{\partial \hat{\mathbf{w}}_i^T}{\partial \mathbf{w}_i^T} \frac{\partial \mathcal{L}}{\partial \hat{\mathbf{w}}_i^T}. \tag{110}$$

# D    LINEAR ICA

## D.1    OVERVIEW

The goal of ICA (Jutten & Herault, 1991; Comon, 1994; Bell & Sejnowski, 1997; Hyvarinen, 1999) is to linearly transform the data into a set of components that are as statistically independent as possible. That is, if $\mathbf{x} \in \mathbb{R}^k$ is a row vector, the goal is to find an unmixing matrix $\mathbf{B} \in \mathbb{R}^{k \times k}$ such that

$$\mathbf{y} = \mathbf{x}\mathbf{B} \tag{111}$$

has its components $y_1, ..., y_k$ as independent as possible.

In an alternative equivalent formulation, we assume that the observed $x_1, ..., x_k$ were generated by mixing $k$ independent sources using a mixing matrix $\mathbf{A}$, i.e. we have $\mathbf{x} = \mathbf{s}\mathbf{A}$ with $\mathbf{A} = \mathbf{B}^{-1}$ and $\mathbf{s} = \mathbf{y}$, and the goal is to recover the mixing matrix.

To be able to estimate $\mathbf{B}$, at least $k - 1$ of the independent components *must* have non-Gaussian distributions. Otherwise, if the independent components have Gaussian distributions, then the model is *not* identifiable.

Without any other assumptions about how the data was mixed, ICA has two ambiguities: it is not possible to determine the variances nor the order of the independent sources.

There are a few different algorithms for performing ICA (Hyvärinen & Oja, 2000; Bingham et al., 2015), the most popular being FastICA (Hyvarinen, 1999). In the ICA model, the mixing matrix can be arbitrary, meaning that it is not necessarily orthogonal; however, to go about finding it, ICA algorithms generally decompose it into a sequence of transformations that include orthogonal matrices, for there is a guarantee that any matrix has an SVD that encodes the sequence: rotation (orthogonal), scaling (diagonal), rotation (orthogonal). FastICA uses PCA as a preprocessing step to whiten the data and reduce dimensionality. The whitening operation performs the initial rotation and scaling. What remains after that is to find the last rotation that maximises non-Gaussianity. If $\mathbf{U}\mathbf{S}\mathbf{V}^{\mathbf{T}}$ is the SVD of the covariance matrix, then the whitening transformation is $\mathbf{V}\mathbf{S}^{-1}$, and the resulting overall ICA transformation has the following form

$$\mathbf{B} = \mathbf{V}\mathbf{S}^{-1}\mathbf{W}, \tag{112}$$

with $\mathbf{V}$ and $\mathbf{W}$ orthogonal matrices and $\mathbf{S}$ diagonal.

**Ordering of components**    As ICA makes no assumption about the transformation, all the ICs are assumed to have unit variance. And so there is no order implied. Without prior knowledge about how the mixing of the sources occurred, it is impossible to resolve the ICA ambiguity. However, if we have reason to assume that the unmixing matrix is close to orthogonal, or simply has unit norm columns (akin to relaxing the orthogonality constraint in PCA while maintaining unit norm directions) then we can in fact order the components by their variances. In this case, the norm of the columns of the unmixing matrix obtained by FastICA is inversely proportional to the standard deviations of the components. On the other hand, it is also possible to assume that it is the mixing matrix, rather than the unmixing matrix, that has unit norm columns (Hyvärinen, 1999). It is also possible to base the ordering on the measure of non-Gaussianity (Hyvärinen, 1999).

## D.2    EQUIVARIANT ADAPTIVE SEPARATION VIA INDEPENDENCE

Equivariant adaptive separation via independence (EASI) (Cardoso & Laheld, 1996) is a serial updating algorithm for source separation. It combines both a whitening term

$$\mathbf{W}(\mathbf{y}^T\mathbf{y} - \mathbf{I}) \tag{113}$$

and a skew-symmetric term

$$\mathbf{W}(\mathbf{y}^T h(\mathbf{y}) - h(\mathbf{y})^T\mathbf{y}) \tag{114}$$

to obtain the following global relative gradient update rule

$$\Delta\mathbf{W} = -\eta\mathbf{W}(\mathbf{y}^T\mathbf{y} - \mathbf{I} + \mathbf{y}^T h(\mathbf{y}) - h(\mathbf{y})^T\mathbf{y}), \tag{115}$$

with $\eta$ the learning rate. The skew-symmetric term originates from skew-symmetrising $\mathbf{W}h(\mathbf{y})^T\mathbf{y}$ in order to roughly preserve orthogonality with each update. To see why, suppose we have $\mathbf{W}^T\mathbf{W} = \mathbf{I}$, and we modify it into $\mathbf{W} + \mathbf{W}\boldsymbol{\mathcal{E}}$, then we can expand it as

$$(\mathbf{W} + \mathbf{W}\boldsymbol{\mathcal{E}})^T(\mathbf{W} + \mathbf{W}\boldsymbol{\mathcal{E}}) = \mathbf{I} + \boldsymbol{\mathcal{E}}^T + \boldsymbol{\mathcal{E}} + \boldsymbol{\mathcal{E}}^T\boldsymbol{\mathcal{E}}. \tag{116}$$

If we want $\mathbf{W} + \mathbf{W}\boldsymbol{\mathcal{E}}$ to remain orthogonal up to first-order, we must also have $(\mathbf{W} + \mathbf{W}\boldsymbol{\mathcal{E}})^T(\mathbf{W} + \mathbf{W}\boldsymbol{\mathcal{E}}) = \mathbf{I} + o(\boldsymbol{\mathcal{E}})$. This implies that $\boldsymbol{\mathcal{E}}$ must be skew-symmetric with $\boldsymbol{\mathcal{E}}^T = -\boldsymbol{\mathcal{E}}$.

# E  NONLINEAR PCA

## E.1  STATIONARY POINT

### E.1.1  MODIFIED DERIVATIVE

Lets consider the case where $h'(x) = 1$. This approximation does not seem to affect the learned filters when $\boldsymbol{\Sigma}$ is estimated from the data (see Appendix F.4). We can write our gradient as

$$\frac{\partial\mathcal{L}}{\partial\mathbf{W}} \approx \mathbf{x}^T(h(\mathbf{y}\boldsymbol{\Sigma}^{-1})\boldsymbol{\Sigma}\mathbf{W}^T\mathbf{W} - \mathbf{y}) = \mathbf{x}^T(\hat{\mathbf{y}} - \mathbf{y}). \tag{117}$$

We can take a look at what happens at the stationary point. Taking Eq. 117, and assuming that $\mathbf{W}^T\mathbf{W} = \mathbf{I}$, we have

$$\Delta\mathbf{W} = \mathbf{x}^T\mathbf{y} - \mathbf{x}^T h(\mathbf{y}\boldsymbol{\Sigma}^{-1})\boldsymbol{\Sigma}\mathbf{W}^T\mathbf{W} \tag{118}$$

$$\mathbf{0} = \mathbb{E}(\mathbf{W}^T\mathbf{x}^T\mathbf{y} - \mathbf{W}^T\mathbf{x}^T h(\mathbf{y}\boldsymbol{\Sigma}^{-1})\boldsymbol{\Sigma}\mathbf{W}^T\mathbf{W}) \tag{119}$$

$$\mathbf{0} = \boldsymbol{\Sigma}^2 - \mathbb{E}(\mathbf{y}^T h(\mathbf{y}\boldsymbol{\Sigma}^{-1}))\boldsymbol{\Sigma}\mathbf{W}^T\mathbf{W} \tag{120}$$

$$\boldsymbol{\Sigma}^2 = \mathbb{E}(\mathbf{y}^T h(\mathbf{y}\boldsymbol{\Sigma}^{-1}))\boldsymbol{\Sigma}\mathbf{I} \tag{121}$$

$$\boldsymbol{\Sigma} = \mathbb{E}(\mathbf{y}^T h(\mathbf{y}\boldsymbol{\Sigma}^{-1})). \tag{122}$$

Given that $\mathbb{E}(\mathbf{y}^T h(\mathbf{y}\boldsymbol{\Sigma}^{-1}))$ is expected to be diagonal, we end up with $\sigma_i = \mathbb{E}(y_i h(y_i/\sigma_i))$. To be able to satisfy the above, we would need $h$ to adapt such that $\sqrt{\operatorname{var}(h(\mathbf{y}\boldsymbol{\Sigma}^{-1}))} \approx \mathbf{1}$. We can do this by using a function that adjusts the scale, such as $h_a(x) = a\tanh(x/a)$. For instance, if $x \sim \mathcal{N}(0,1)$, we have $\sqrt{\operatorname{var}(tanh(x))} \approx 0.62$ and $\sqrt{\operatorname{var}(3\tanh(\frac{x}{3}))} \approx 0.90$. So by setting a value of at least $a = 3$, we get closer back to unit variance.

Alternatively, we can consider an asymmetric nonlinear function such as $a\tanh(x)$, where $a = 1/\sqrt{\operatorname{var}(tanh(x))}$. If the standardised $x$ is roughly standard normal, then $a \approx 1.6$.

We also have another option instead of scaling: we can compensate by adding an additional loss term that adds to the encoder contribution as follows:

$$\mathcal{L}(\mathbf{W}) = \mathbb{E}(||\mathbf{x} - h(\mathbf{x}\mathbf{W}\boldsymbol{\Sigma}^{-1})\boldsymbol{\Sigma}[\mathbf{W}^T]_{sg}||_2^2) \tag{123}$$

$$+ \alpha\mathbb{E}(||\mathbf{x} - \mathbf{x}\mathbf{W}[\mathbf{W}^T]_{sg}||_2^2) \tag{124}$$

$$h(x) = \tanh(x) \tag{125}$$

$$h'(x) = 1 \tag{126}$$

This results in

$$\Delta\mathbf{W} = \alpha(\mathbf{x}^T\mathbf{y} - \mathbf{x}^T\mathbf{y}\mathbf{W}^T\mathbf{W}) + \mathbf{x}^T\mathbf{y} - \mathbf{x}^T h(\mathbf{y}\boldsymbol{\Sigma}^{-1})\boldsymbol{\Sigma}\mathbf{W}^T\mathbf{W} \tag{127}$$

$$\mathbf{W}^T\mathbf{W} = (\alpha\boldsymbol{\Sigma}^2 + \boldsymbol{\Sigma}^2)(\alpha\boldsymbol{\Sigma}^2 + \mathbb{E}(\mathbf{y}^T h(\mathbf{y}\boldsymbol{\Sigma}^{-1}))\boldsymbol{\Sigma})^{-1} \tag{128}$$

$$= (\mathbf{I} + \alpha^{-1}\mathbf{I})(\mathbf{I} + \alpha^{-1}\mathbb{E}(\mathbf{y}^T h(\mathbf{y}\boldsymbol{\Sigma}^{-1}))\boldsymbol{\Sigma}^{-1})^{-1} \tag{129}$$

$$\xrightarrow[\alpha\to\infty]{} \mathbf{I} \tag{130}$$

And so if we set $\alpha$ large enough, we can compensate for the scale. In practice $\alpha \geq 2$ is enough to induce an effect.

### E.1.2 Unmodified derivative

If the derivative of $h$ is not modified, and assuming $h_a = a\tanh(x/a)$, $h'_a(x) = 1 - h_a^2(x)/a^2$, the stationary point yields

$$\boldsymbol{\Sigma}^2 = \mathbb{E}(\mathbf{y}^T h_a(\mathbf{y}\boldsymbol{\Sigma}^{-1})\boldsymbol{\Sigma}) - \frac{1}{a}\mathbb{E}((\mathbf{y}^T\mathbf{y} - \mathbf{y}^T h_a(\mathbf{y}\boldsymbol{\Sigma}^{-1})\boldsymbol{\Sigma})h_a^2(\mathbf{y}\boldsymbol{\Sigma}^{-1})). \tag{131}$$

We can see that the larger $a$ is, the more the equality will be satisfied. The choice of $a$ will generally depend on the type of distribution, which can be seen more clearly in Fig. 5.

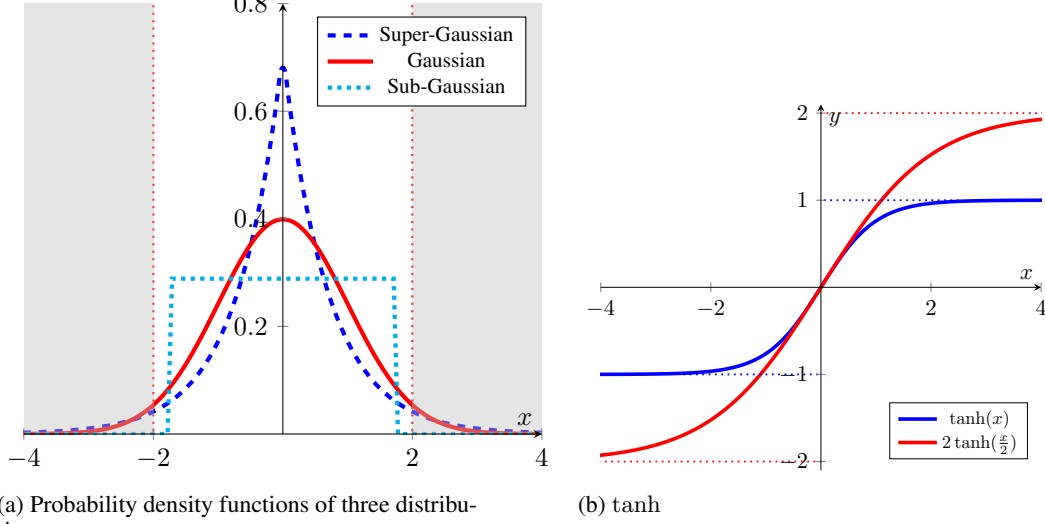

(a) Probability density functions of three distributions

(b) $\tanh$

Figure 5: Three distributions with unit variance (a): uniform distribution (sub-Gaussian), Gaussian distribution, and Laplace distribution (super-Gaussian). Shaded in grey is any $|x| \geq 2\sigma$. When using $\tanh$ without any scale adjustment, we can see that for this super-Gaussian distribution (or for any heavy-tailed distribution), values beyond $2\sigma$ might have their reconstruction impaired because of the squashing by $\tanh$ (b). A remedy, in this case, would be to use $a\tanh(x/a)$ with $a \geq 1$. For a sub-Gaussian distribution, it is more suited to use $a = 1$ (or even slightly less than 1 in some cases).

### E.2 GRADIENT OF TRAINABLE $\boldsymbol{\Sigma}$

From the loss in Eq. 5, we have

$$\frac{\partial \mathcal{L}}{\partial \boldsymbol{\Sigma}_e} = -\mathrm{diag}(\mathrm{diag}^{-1}(\mathbf{W}_e^T \mathbf{x}^T(\hat{\mathbf{x}} - \mathbf{x})\mathbf{W}_d \boldsymbol{\Sigma}_d \odot h'(\mathbf{z})\boldsymbol{\Sigma}_e^{-2})) \tag{132}$$

$$\frac{\partial \mathcal{L}}{\partial \boldsymbol{\Sigma}_d} = \mathrm{diag}((\hat{\mathbf{x}} - \mathbf{x})\mathbf{W}_d \odot h(\mathbf{z})) \tag{133}$$

where diag creates a diagonal matrix from a vector and $\mathrm{diag}^{-1}$ takes the diagonal part of the matrix as vector. If we tie the weights but keep the contributions separate, we obtain

$$\frac{\partial \mathcal{L}}{\partial \boldsymbol{\Sigma}_e} = -\mathrm{diag}(\mathrm{diag}^{-1}(\mathbf{y}^T(\hat{\mathbf{y}} - \mathbf{y}) \odot h'(\mathbf{y}\boldsymbol{\Sigma}^{-1})\boldsymbol{\Sigma}^{-1})) \tag{134}$$

$$\frac{\partial \mathcal{L}}{\partial \boldsymbol{\Sigma}_d} = \mathrm{diag}((\hat{\mathbf{y}} - \mathbf{y}) \odot h(\mathbf{y}\boldsymbol{\Sigma}^{-1})) \tag{135}$$

Given $\boldsymbol{\Sigma} = \mathrm{diag}(\boldsymbol{\sigma})$, we can write

$$\frac{\partial \mathcal{L}}{\partial \boldsymbol{\sigma}_e} = -\mathbf{y}(\hat{\mathbf{y}} - \mathbf{y})h'(\mathbf{y}\boldsymbol{\sigma}^{-1})\boldsymbol{\sigma}^{-1} \tag{136}$$

$$\frac{\partial \mathcal{L}}{\partial \boldsymbol{\sigma}_d} = (\hat{\mathbf{y}} - \mathbf{y})h(\mathbf{y}\boldsymbol{\sigma}^{-1}) \tag{137}$$

$$\frac{\partial \mathcal{L}}{\partial \boldsymbol{\sigma}} = -\mathbf{y}(\hat{\mathbf{y}} - \mathbf{y})h'(\mathbf{y}\boldsymbol{\sigma}^{-1})\boldsymbol{\sigma}^{-1} + (\hat{\mathbf{y}} - \mathbf{y})h(\mathbf{y}\boldsymbol{\sigma}^{-1}) \tag{138}$$

Let us consider $h = \tanh$, which results in

$$\frac{\partial \mathcal{L}}{\partial \boldsymbol{\sigma}} = -\mathbf{y}(\hat{\mathbf{y}} - \mathbf{y})(1 - h^2(\mathbf{y}\boldsymbol{\sigma}^{-1}))\boldsymbol{\sigma}^{-1} + (\hat{\mathbf{y}} - \mathbf{y})h(\mathbf{y}\boldsymbol{\sigma}^{-1}) \tag{139}$$

$$= (\hat{\mathbf{y}} - \mathbf{y})(-\mathbf{y}\boldsymbol{\sigma}^{-1} + \mathbf{y}\boldsymbol{\sigma}^{-1}h^2(\mathbf{y}\boldsymbol{\sigma}^{-1}) + h(\mathbf{y}\boldsymbol{\sigma}^{-1})) \tag{140}$$

$$= (\hat{\mathbf{y}} - \mathbf{y})(-\mathbf{y}\boldsymbol{\sigma}^{-1} + f(\mathbf{y}\boldsymbol{\sigma}^{-1})) \tag{141}$$

$$= \boldsymbol{\sigma}(\mathbf{y}^2\boldsymbol{\sigma}^{-2} - \mathbf{y}\boldsymbol{\sigma}^{-1}f(\mathbf{y}\boldsymbol{\sigma}^{-1}) - \hat{\mathbf{y}}\mathbf{y}\boldsymbol{\sigma}^{-2} + \hat{\mathbf{y}}\boldsymbol{\sigma}^{-1}f(\mathbf{y}\boldsymbol{\sigma}^{-1})) \tag{142}$$

$$= \boldsymbol{\sigma}(\mathbf{z}^2 - \mathbf{z}f(\mathbf{z}) - \hat{\mathbf{z}}\mathbf{z} + \hat{\mathbf{z}}f(\mathbf{z})), \tag{143}$$

where $f(z) = zh^2(z) + h(z)$, $\mathbf{z} = \mathbf{y}\boldsymbol{\sigma}^{-1}$, and $\hat{\mathbf{z}} = \hat{\mathbf{y}}\boldsymbol{\sigma}^{-1}$. We can plot $f(z)z$, $z^2$ and $f(z)z - z^2$

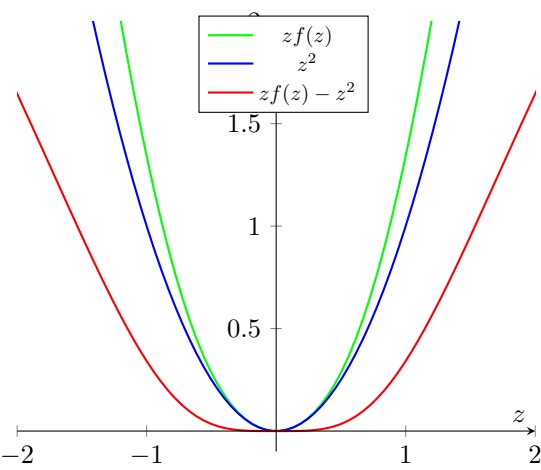

Given that $\mathbf{y}$ is centred, we have $\mathbb{E}(\mathbf{y}^2) = \text{var}(\mathbf{y})$, and so if we take the expectation for the gradient, we will obtain

$$\mathbb{E}(\mathbf{z}^2\boldsymbol{\sigma}) = \mathbb{E}(\mathbf{y}^2\boldsymbol{\sigma}^{-1}) = \text{var}(\mathbf{y})\boldsymbol{\sigma}^{-1}. \tag{144}$$

For $|z| < 1$, we also have $zf(z) \approx z^2$. And so we can see that $\frac{\partial \mathcal{L}}{\partial \boldsymbol{\sigma}}$ takes the form of terms that are roughly $\propto \text{var}(\mathbf{y})\boldsymbol{\sigma}^{-1}$.

### E.3 RECONSTRUCTION LOSS IN LATENT SPACE AND RELATIONSHIP TO BLIND DECONVOLUTION

When performing the reconstruction in latent space, we need to add something that maintains the orthogonality of $\mathbf{W}$. If we multiply both sides of the difference in Eq. 12 by $[\mathbf{W}]_{sg}$ and include a symmetric orthogonal regulariser, we obtain

$$\mathcal{L}(\mathbf{W}, \boldsymbol{\Sigma}) = \mathbb{E}(||\mathbf{x}[\mathbf{W}]_{sg} - h(\frac{\mathbf{xW}}{\boldsymbol{\sigma}})\boldsymbol{\sigma}[\mathbf{W}^T\mathbf{W}]_{sg}||_2^2) + \beta||\mathbf{I} - \mathbf{W}^T\mathbf{W}||_F^2. \tag{145}$$

Although we can assume that $[\mathbf{W}^T\mathbf{W}]_{sg} = \mathbf{I}$ is maintained by an orthogonal regulariser, and might be tempted to simply remove it, we find that keeping it helps make it more stable, especially during the initial period of training where $\mathbf{W}$ might be transitioning via a non-orthogonal matrix. Given that $[\mathbf{W}^T\mathbf{W}]_{sg}$ is square, we can also use its lower triangular form $\triangle[\mathbf{W}^T\mathbf{W}]_{sg}$ to obtain an automatic ordering:

$$\mathcal{L}(\mathbf{W}, \boldsymbol{\Sigma}) = \mathbb{E}(||\mathbf{x}[\mathbf{W}]_{sg} - h(\frac{\mathbf{xW}}{\boldsymbol{\sigma}})\boldsymbol{\sigma}\triangle[\mathbf{W}^T\mathbf{W}]_{sg}||_2^2) + \beta||\mathbf{I} - \mathbf{W}^T\mathbf{W}||_F^2. \tag{146}$$

If indeed we do assume that $[\mathbf{W}^T\mathbf{W}]_{sg} = \mathbf{I}$, we find that it generally requires a much stronger orthogonal regulariser $\beta > 1$, with an asymmetric regulariser being better than the symmetric, and it works better when $\boldsymbol{\Sigma}$ is trainable rather than estimated from data.

$$\mathcal{L}(\mathbf{W}, \boldsymbol{\Sigma}) = \mathbb{E}(||\mathbf{x}[\mathbf{W}]_{sg} - h(\frac{\mathbf{xW}}{\boldsymbol{\sigma}})\boldsymbol{\sigma}||_2^2) + \beta||\mathbf{I} - \mathbf{W}^T\mathbf{W}||_F^2. \tag{147}$$

Instead of $\beta||\mathbf{I} - \mathbf{W}^T\mathbf{W}||_F^2$ we can also use the asymmetric version $\beta||\triangle\mathbf{W}^T[\mathbf{W}]_{sg}||_F^2$ (see Appendix B.2).

Alternatively, we can use the linear reconstruction loss $\mathbb{E}(||\mathbf{xW}[\mathbf{W}^T]_{sg} - \mathbf{x}||_2^2)$ for maintaining the orthogonality:

$$\mathcal{L}(\mathbf{W}, \boldsymbol{\Sigma}) = \mathbb{E}(||\mathbf{x}[\mathbf{W}]_{sg} - h(\mathbf{xW\Sigma})\boldsymbol{\Sigma}||_2^2) + \mathbb{E}(||\mathbf{xW}[\mathbf{W}^T]_{sg} - \mathbf{x}||_2^2). \tag{148}$$

Putting this in context of blind deconvolution (Lambert, 1996; Haykin, 1996), we can note the first term in Eq. 145,

$$\mathbb{E}(||h(\frac{\mathbf{y}}{\boldsymbol{\sigma}})\boldsymbol{\sigma} - [\mathbf{y}]_{sg}||_2^2), \tag{149}$$

can be seen as a modified form of the Bussgang criterion

$$\mathbb{E}(||h(\mathbf{y}) - \mathbf{y}||_2^2. \tag{150}$$

### E.4 CONNECTION WITH $L_1$ SPARSITY

ICA is closely related to sparse coding (Olshausen & Field, 1996; Bell & Sejnowski, 1997). This is because maximising sparsity can be seen as a method for maximising non-Gaussianity (Hyvärinen & Oja, 2000)– which is a particular ICA method. Reconstruction ICA (RICA) (Le et al., 2011) is a method that combines $L_1$ sparsity with a linear reconstruction loss, and it was initially proposed for learning overcomplete sparse features, in contrast to conventional ICA which does not model over-completeness. RICA is, in fact, simply performing nonlinear PCA: it induces a latent reconstruction term in the weight update, and, by tying the weights of the encoder and decoder, it enforces $\mathbf{W}$ to have orthogonal columns. RICA is thus a special case of the more general ICA model which can learn an arbitrary transformation rather than just an orthogonal transformation. The RICA loss is

$$\mathcal{L}(\mathbf{W}) = ||\mathbf{x} - \mathbf{x}\mathbf{W}\mathbf{W}^T||_2^2 + \beta \sum |\mathbf{x}\mathbf{w_j^T}|, \tag{151}$$

where $\mathbf{w}_j^T$ is a column of $\mathbf{W}$. This works equally well if we use the subspace variant, i.e.:

$$\mathcal{L}(\mathbf{W}) = ||\mathbf{x} - \mathbf{x}[\mathbf{W}]_{sg}\mathbf{W}^T||_2^2 + \beta \sum |\mathbf{x}\mathbf{w_j^T}|, \tag{152}$$

If we compute the gradient contributions from Eq. 152, we obtain

$$\frac{\partial \mathcal{L}}{\partial \mathbf{W}_e} = \beta \mathbf{x}^T \text{sign}(\mathbf{y}) \tag{153}$$

$$\frac{\partial \mathcal{L}}{\partial \mathbf{W}_d} = (\hat{\mathbf{x}} - \mathbf{x})^T \mathbf{y}, \tag{154}$$

which results in the combined gradient

$$\frac{\partial \mathcal{L}}{\partial \mathbf{W}} = (\hat{\mathbf{x}} - \mathbf{x})^T \mathbf{y} + \beta \mathbf{x}^T \text{sign}(\mathbf{y}) \tag{155}$$

$$= \hat{\mathbf{x}}^T \mathbf{y} + \mathbf{x}^T (\beta \text{sign}(\mathbf{y}) - \mathbf{y}). \tag{156}$$

The main relevant part in Eq. 156 is

$$\mathbf{x}^T (\beta \text{sign}(\mathbf{y}) - \mathbf{y}), \tag{157}$$

and this is similar to the latent reconstruction term in the nonlinear PCA gradient, except we now have a dependency on the value of $\beta$. No suggestion for the best value of $\beta$ is given by the authors (Le et al., 2011); however, previous works on sparse coding (Olshausen & Field, 1996) have set $\beta$ to be proportional to the standard deviation of the input. We can perhaps gain some insight why by looking more closely at the influence on Eq. 156 of the norm of the input. For that we extract $||\mathbf{x}||_2$ to get

$$\beta \text{sign}(\mathbf{y}) - \mathbf{y} = ||\mathbf{x}||_2 (\beta \frac{\text{sign}(\mathbf{y})}{||\mathbf{x}||_2} - \frac{\mathbf{x}\mathbf{W}}{||\mathbf{x}||_2}). \tag{158}$$

We now see that if $||\mathbf{x}||_2$ increases, then the norm of $\frac{\text{sign}(\mathbf{y})}{||\mathbf{x}||_2}$ will decrease, while that of $\frac{\mathbf{x}\mathbf{W}}{||\mathbf{x}||_2}$ will remain constant. Indeed we have

$$\frac{||\text{sign}(\mathbf{y})||_2}{||\mathbf{x}||_2} \leq \frac{\sqrt{k}}{||\mathbf{x}||_2} \xrightarrow[||\mathbf{x}||_2 \to \infty]{} 0 \tag{159}$$

and

$$\frac{||\mathbf{xW}||_2}{||\mathbf{x}||_2} \approx 1, \tag{160}$$

because, from the reconstruction term, we have

$$||\mathbf{xW}||_2^2 = \mathbf{xWW}^T\mathbf{x}^T = \hat{\mathbf{x}}\mathbf{x}^T \approx \mathbf{xx}^T = ||\mathbf{x}||_2^2. \tag{161}$$

And so to make the relative difference $\beta\mathrm{sign}(\mathbf{y}) - \mathbf{y}$ invariant to the norm of the input, we can set $\beta$ to be proportional to it, or more simply proportional to the standard deviation of the input as it is already centred, i.e. $\beta = \beta_0\mathbb{E}(||\mathbf{x}||_2)$. As to what the value of $\beta_0$ should be, we can plot the functions $f_{\beta_0}(y) = \beta_0\mathrm{sign}(y) - y$ and $f_{\beta_0}(y) = \beta_0\tanh(y) - y$:

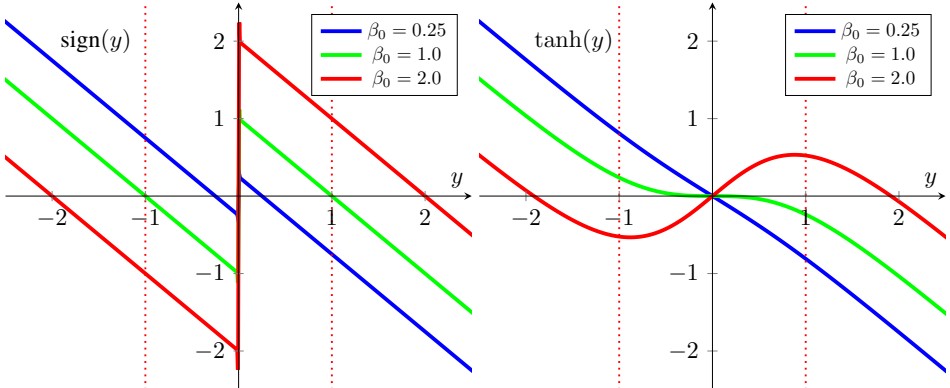

For $\tanh$, if $\beta_0 \leq 1$, then it has a single inflection point, while if $\beta_0 > 1$, then it has two inflection points. A reasonable range for $\beta_0$ appears to be $]0.0, 1.0]$. This range is in line with (Olshausen & Field, 1996) where $\beta_0$ was set to $0.14$.

The additional term in the gradient does not maintain unit norm exactly; this is because, if we follow a similar analysis as in Appendix A.3.2, we have for $\Delta\mathbf{W} = \mathbf{0}$

$$\mathbf{W}^T\mathbf{W} = \mathbf{I} - \beta\mathbb{E}(\mathbf{y}^T\mathrm{sign}(\mathbf{y}))\mathbf{\Sigma}^{-2}, \tag{162}$$

if using the subspace RICA loss, or

$$\mathbf{W}^T\mathbf{W} = \mathbf{I} - \frac{\beta}{2}\mathbb{E}(\mathbf{y}^T\mathrm{sign}(\mathbf{y}))\mathbf{\Sigma}^{-2}, \tag{163}$$

if using the original RICA loss. Therefore, the norm of the columns is less than one.

Another thing we can note is that $\lim_{a\to\infty}\tanh(ax) = \mathrm{sign}(x)$, so in Eq. 156 we could potentially replace sign with $\tanh$ to get

$$\frac{\partial\mathcal{L}}{\partial\mathbf{W}} = \hat{\mathbf{x}}^T\mathbf{y} + \mathbf{x}^T(\beta\tanh(\mathbf{y}) - \mathbf{y}). \tag{164}$$

We know that $\tanh$ is the derivative of $\log\cosh$, which is none other than the function used by FastICA for the negentropy approximation (Hyvarinen, 1999). This can be expressed as the following loss function:

$$\mathcal{L}(\mathbf{W}) = ||\mathbf{x} - \mathbf{x}[\mathbf{W}]_{sg}\mathbf{W}^T||_2^2 + \beta\sum\log\cosh(\mathbf{xw_j^T}). \tag{165}$$

### E.5 SKEW-SYMMETRIC UPDATE

Similar to the derivation of the EASI algorithm (see Appendix D.2), we can combine the gradient obtained from a linear reconstruction loss with the term $\mathbf{Wy}^Th(\mathbf{y})$. However, to avoid a similar

problem as in the previous section of the gradient update not maintaining unit norm columns, we can simply remove the diagonal part of $\mathbf{y}^T h(\mathbf{y})$, resulting in the weight update

$$\Delta \mathbf{W} \propto \mathbf{x}^T \mathbf{y} - \mathbf{W} \mathbf{y}^T \mathbf{y} - \beta \mathbf{W}(\mathbf{y}^T h(\mathbf{y}) - \text{diag}(h(\mathbf{y}) \odot \mathbf{y})). \tag{166}$$

Or, similar to EASI, we can use it in its skew-symmetric form:

$$\Delta \mathbf{W} \propto \mathbf{x}^T \mathbf{y} - \mathbf{W} \mathbf{y}^T \mathbf{y} - \beta \mathbf{W}(\mathbf{y}^T h(\mathbf{y}) - h(\mathbf{y})^T \mathbf{y})), \tag{167}$$

which we can also write as the loss function

$$\mathcal{L}(\mathbf{W}) = \frac{1}{2}||\mathbf{x} - \mathbf{x}[\mathbf{W}]_{sg}\mathbf{W}^T||_2^2 + \beta \mathbf{1} \mathbf{W} \odot [\mathbf{W}(\mathbf{y}^T h(\mathbf{y}) - h(\mathbf{y})^T \mathbf{y})))]_{sg}\mathbf{1}^T. \tag{168}$$

As justified by the previous section, we can set $\beta = \mathbb{E}(||\mathbf{x}||_2)$ or we can simply use $\mathbf{\Sigma}$ to compensate:

$$\Delta \mathbf{W} \propto \mathbf{x}^T \mathbf{y} - \mathbf{W} \mathbf{y}^T \mathbf{y} - \mathbf{W}(\mathbf{y}^T h(\mathbf{y}\mathbf{\Sigma}^{-1}) - \text{diag}(\mathbf{y} \odot h(\mathbf{y}\mathbf{\Sigma}^{-1})))\mathbf{\Sigma}. \tag{169}$$

In all of these variants, we can have $h = \text{sign}$ or $h = \tanh$.

### E.6 MODIFIED DECODER CONTRIBUTION

The gradient updates of the encoder and decoder (as derived in 3.1) are

$$\frac{\partial \mathcal{L}}{\partial \mathbf{W}_e} = \mathbf{x}^T(\hat{\mathbf{y}} - \mathbf{y}) \odot h'(\mathbf{z}) \tag{170}$$

$$\frac{\partial \mathcal{L}}{\partial \mathbf{W}_d} = (\hat{\mathbf{x}} - \mathbf{x})^T h(\mathbf{z})\mathbf{\Sigma}. \tag{171}$$

The main issue is that, when both are used in the tied case, the decoder contribution overpowers the encoder contribution. This is evident in the linear case, where, when $\mathbf{W}$ is semi-orthogonal, the encoder contribution is zero. Indeed, let $\mathbf{W} \in \mathbb{R}^{p \times k}$ with $p > k$, recall that in the linear case (Appendix A.2.1) we have

$$\frac{\partial \mathcal{L}}{\partial \mathbf{W}_e} = \mathbf{x}^T \mathbf{y}(\mathbf{W}^T \mathbf{W} - \mathbf{I}) \tag{172}$$

$$\frac{\partial \mathcal{L}}{\partial \mathbf{W}_d} = (\mathbf{W}\mathbf{W}^T - \mathbf{I})\mathbf{x}^T \mathbf{y} \tag{173}$$

If we consider their Frobineus norms, we can write

$$||\frac{\partial \mathcal{L}}{\partial \mathbf{W}_e}||_F = 0 \tag{174}$$

$$\lambda_{\min}(\mathbf{x}^T \mathbf{y})\sqrt{p - k} \le ||\frac{\partial \mathcal{L}}{\partial \mathbf{W}_d}||_F, \tag{175}$$

where the lower bound on the decoder contribution can be derived from the fact that for $\mathbf{A} \in \mathbb{R}^{p \times k}$ and $\mathbf{B} \in \mathbb{R}^{k \times n}$, we have (Fang et al., 1994)

$$\lambda_{\min}(\mathbf{A})||\mathbf{B}||_F \le ||\mathbf{A}\mathbf{B}||_F \le \lambda_{\max}(\mathbf{A})||\mathbf{B}||_F, \tag{176}$$

where $\lambda_{\min}(\mathbf{A})$ and $\lambda_{\max}(\mathbf{A})$ refer to the smallest and largest eigenvalues of $\mathbf{A}$, respectively, and

$$||\mathbf{W}\mathbf{W}^T - \mathbf{I}||_F^2 = tr((\mathbf{W}\mathbf{W}^T - \mathbf{I})^T(\mathbf{W}\mathbf{W}^T - \mathbf{I})) \tag{177}$$

$$= tr(\mathbf{W}\mathbf{W}^T\mathbf{W}\mathbf{W}^T - 2\mathbf{W}\mathbf{W}^T + \mathbf{I}) \tag{178}$$

$$= tr(\mathbf{I} - \mathbf{W}\mathbf{W}^T) = p - k. \tag{179}$$

In the nonlinear case, the gradient updates are approximately close to the linear update (especially if $h(z) = a\tanh(z/a)$ with $a > 3$), except that the encoder contribution is no longer zero, and it is still overpowered by the decoder contribution. And so if omit the latter, we can better see the effect of the former.

An alternative would be to modify the relative scale so that the decoder contribution does not overpower the encoder contribution. The variance of the components plays a role in this, given that this is not a problem in the case of conventional nonlinear PCA with whitened input, where all the components have unit variance. As a This suggests three options: (1) scale the encoder contribution by $\mathbf{\Sigma}$; (2) drop $\mathbf{\Sigma}$ from the decoder contribution and, optionally, $h'(\mathbf{z})$ from the encoder contribution; (3) scale down the decoder contribution by a constant, which can simply be the inverse of the spectral, Frobineus, or nuclear norm of $\mathbf{\Sigma}$.

For the third option we can write it as a loss:

$$\mathcal{L}(\mathbf{W}) = \mathbb{E}(||\mathbf{x} - h(\mathbf{x}\mathbf{W}\mathbf{\Sigma}^{-1})\mathbf{\Sigma}[\mathbf{W}^T]_{sg}||_2^2 + \frac{1}{||\mathbf{\Sigma}||_2}||\mathbf{x} - [h(\mathbf{x}\mathbf{W}\mathbf{\Sigma}^{-1})\mathbf{\Sigma}]_{sg}\mathbf{W}^T||_2^2). \tag{180}$$

### E.7 Ordering the components based on index position

Here we look at a few ways for ordering the components automatically based on index position.

#### E.7.1 Regulariser

One way to do this is via a Gram-Schmidt-like regulariser (Wang et al., 1995) by adding

$$J(\mathbf{W}) = ||\triangle(\mathbf{W}^T[\mathbf{W}]_{sg})||_F^2, \tag{181}$$

where $\triangle$ refers to the lower triangular matrix without the diagonal element.

#### E.7.2 Triangular weight update

In the GHA (see Appendix A.3.3), we can order the components by index position by taking the lower (or upper) triangular part of the term $\mathbf{W}\mathbf{y}^T\mathbf{y}$, a term which originates from the linear decoder contribution. Similarly to the GHA, we can also include a triangular term taken from the nonlinear decoder contribution, which is

$$\mathbf{W}(h(\mathbf{z})\mathbf{\Sigma})^T h(\mathbf{z})\mathbf{\Sigma}, \tag{182}$$

where $\mathbf{z} = \mathbf{y}\mathbf{\Sigma}^{-1}$. However, as we have seen in Section E.6, we need to scale this term appropriately lest it overpowers the encoder contribution. We can consider variations where we drop $\mathbf{\Sigma}$ from either left or right, or make the approximation of $y \approx h(\mathbf{z})\mathbf{\Sigma}$:

$$\mathbf{W}\triangledown((h(\mathbf{z})\mathbf{\Sigma})^T h(\mathbf{z})) \tag{183}$$

$$\text{or } \mathbf{W}\triangledown(h(\mathbf{z})^T h(\mathbf{z})\mathbf{\Sigma}) \tag{184}$$

$$\text{or } \mathbf{W}\triangledown(h(\mathbf{z})^T\mathbf{y}) \tag{185}$$

$$\text{or } \mathbf{W}\triangledown(\mathbf{y}^T h(\mathbf{z})) \tag{186}$$

$$\text{or } \mathbf{W}\triangledown(\mathbf{z}^T\mathbf{y}) \tag{187}$$

$$\text{or } \mathbf{W}\triangledown(\mathbf{y}^T\mathbf{z}) \tag{188}$$

If $\mathbf{\Sigma}$ is non-trainable and $h = \tanh$, then $\mathbf{\Sigma} = a\hat{\mathbf{\Sigma}}$ where $a \geq 3$ and $\hat{\mathbf{\Sigma}}$ is the estimated standard deviation of $\mathbf{y}$.

### E.7.3 WEIGHTED LATENT RECONSTRUCTION

Another option is similar to the weighted subspace algorithm (see A.3.2) where we insert a linearly-spaced $\mathbf{\Lambda}$ into the weight update derived in Eq. 117:

$$\Delta\mathbf{W} \propto \mathbf{x}^T\mathbf{y}\mathbf{\Lambda} - \mathbf{x}^T h(\mathbf{y}\mathbf{\Sigma}^{-1})\mathbf{\Sigma}\mathbf{\Lambda}\mathbf{W}^T\mathbf{W}, \tag{189}$$

where $\mathbf{\Lambda} = \mathrm{diag}(\lambda_1, ..., \lambda_k)$ such that $1 \geq \lambda_1 > ... > \lambda_k > 0$.

As loss function we can have:

$$\mathcal{L}(\mathbf{W}) = \mathbb{E}(||[\mathbf{y}]_{sg}\mathbf{\Lambda} - h(\mathbf{y}\mathbf{\Sigma}^{-1})\mathbf{\Sigma}\mathbf{\Lambda}[\mathbf{W}^T\mathbf{W}]_{sg}||_2^2). \tag{190}$$

### E.7.4 EMBEDDED PROJECTIVE DEFLATION

Let us consider the loss

$$\mathcal{L}(\mathbf{W}_e, \mathbf{W}_d, \mathbf{\Sigma}_e, \mathbf{\Sigma}_d) = \frac{1}{2}||\mathbf{x} - h(\mathbf{x}\mathbf{W}_{e1}P(\mathbf{W}_e)\mathbf{\Sigma}_e^{-1})\mathbf{\Sigma}_d\mathbf{W}_d^T||_2^2, \tag{191}$$

where

$$P(\mathbf{W}_e) = \alpha\mathbf{I} + \beta\mathbf{W}_{e2}^T\mathbf{W}_{e3}, \tag{192}$$

and $\mathbf{W}_e = \mathbf{W}_{e1} = \mathbf{W}_{e2} = \mathbf{W}_{e3}$ for elucidating the contribution of each part to the gradient. Taking the gradient of the loss with respect to the weights, we have

$$\frac{\partial\mathcal{L}}{\partial\mathbf{W}_{e1}} = \mathbf{x}^T(((\hat{\mathbf{x}} - \mathbf{x})\mathbf{W}_d) \odot h'(\mathbf{y}\mathbf{\Sigma}^{-1}))P(\mathbf{W}_e) \tag{193}$$

$$\frac{\partial\mathcal{L}}{\partial\mathbf{W}_{e2}} = \beta\mathbf{W}_e((\hat{\mathbf{x}} - \mathbf{x})\mathbf{W}_d \odot h'(\mathbf{y}\mathbf{\Sigma}^{-1}))^T\mathbf{y} \tag{194}$$

$$\frac{\partial\mathcal{L}}{\partial\mathbf{W}_{e3}} = \beta\mathbf{W}_e\mathbf{y}^T((\hat{\mathbf{x}} - \mathbf{x})\mathbf{W}_d \odot h'(\mathbf{y}\mathbf{\Sigma}^{-1})), \tag{195}$$

Setting $\mathbf{W}_e = \mathbf{W}_d = \mathbf{W}$ and $\mathbf{\Sigma}_e = \mathbf{\Sigma}_d = \mathbf{\Sigma}$, we obtain

$$\frac{\partial\mathcal{L}}{\partial\mathbf{W}_{e1}} = \mathbf{x}^T((\hat{\mathbf{y}} - \mathbf{y}) \odot h'(\mathbf{y}\mathbf{\Sigma}^{-1}))P(\mathbf{W}) \tag{196}$$

$$\frac{\partial\mathcal{L}}{\partial\mathbf{W}_{e2}} = \beta\mathbf{W}((\hat{\mathbf{y}} - \mathbf{y}) \odot h'(\mathbf{y}\mathbf{\Sigma}^{-1}))^T\mathbf{y} \tag{197}$$

$$\frac{\partial\mathcal{L}}{\partial\mathbf{W}_{e3}} = \beta\mathbf{W}\mathbf{y}^T((\hat{\mathbf{y}} - \mathbf{y}) \odot h'(\mathbf{y}\mathbf{\Sigma}^{-1})) \tag{198}$$

Let $\mathbf{y}_\delta = (\hat{\mathbf{y}} - \mathbf{y}) \odot h'(\mathbf{y}\mathbf{\Sigma}^{-1})$, then we can write

$$\frac{\partial\mathcal{L}}{\partial\mathbf{W}_{e1}} = \mathbf{x}^T\mathbf{y}_\delta P(\mathbf{W}) \tag{199}$$

$$\frac{\partial\mathcal{L}}{\partial\mathbf{W}_{e2}} = \beta\mathbf{W}\mathbf{y}_\delta^T\mathbf{y} \tag{200}$$

$$\frac{\partial\mathcal{L}}{\partial\mathbf{W}_{e3}} = \beta\mathbf{W}\mathbf{y}^T\mathbf{y}_\delta, \tag{201}$$

which results in the overall gradient

$$\frac{\partial \mathcal{L}}{\partial \mathbf{W}_e} = \mathbf{x}^T \mathbf{y}_\delta P(\mathbf{W}) + \beta \mathbf{W}(\mathbf{y}_\delta^T \mathbf{y} + \mathbf{y}^T \mathbf{y}_\delta). \tag{202}$$

If we set $P(\mathbf{W}) = \mathbf{I} - \searrow\mathbf{W}^T[\mathbf{W}]_{sg}$, we obtain

$$\frac{\partial \mathcal{L}}{\partial \mathbf{W}_e} = \mathbf{x}^T \mathbf{y}_\delta P(\mathbf{W}) - \mathbf{W}\overline{\searrow}\,(\mathbf{y}_\delta^T \mathbf{y}). \tag{203}$$

This introduces an asymmetric term $\mathbf{W}\overline{\searrow}\,(\mathbf{y}_\delta^T \mathbf{y})$, similar to GHA, resulting in an ordering of components by index position.

### E.7.5 NESTED DROPOUT

Similarly to linear case (Appendix A.3.5), let $\mathbf{m}_{1|j} \in \{0,1\}^k$ be a vector such that $m_i = \begin{cases} 1 & \text{if } i < j \\ 0 & \text{otherwise} \end{cases}$, then we can order components using the loss

$$\mathcal{L}(\mathbf{W}) = \mathbb{E}(||\mathbf{x} - h(\mathbf{x}\mathbf{W}\boldsymbol{\Sigma}^{-1}) \odot \mathbf{m}_{1|j}\boldsymbol{\Sigma}[\mathbf{W}^T]_{sg}||_2^2 + ||\mathbf{I} - \mathbf{W}^T\mathbf{W}||_F^2. \tag{204}$$

One thing to note about nested dropout is that the larger the index, the less frequently the component receives a gradient update, so training has to run for longer. It can also benefit from the addition of an orthogonal regulariser to provide a gradient signal to the components receiving less frequent updates from the reconstruction.

### E.8 NON-CENTRED NONLINEAR PCA

When using a zero-centred function like $\tanh$, its input should also be zero-centred, so, to obtain a non-centred version of nonlinear PCA, we can simply subtract the mean, and then add it back after passing through the nonlinear function:

$$\mathcal{L}(\mathbf{W}, \boldsymbol{\sigma}) = \mathbb{E}(||\mathbf{x} - (h(\frac{\mathbf{x}\mathbf{W} - \bar{\boldsymbol{\mu}}_y}{\boldsymbol{\sigma}})\boldsymbol{\sigma} + \bar{\boldsymbol{\mu}}_y)[\mathbf{W}^T]_{sg}||_2^2). \tag{205}$$

We are simply standardising (or normalising to zero mean and unit variance) the components before the nonlinearity and then undoing the standardisation after the nonlinearity. This could also be seen as applying a batch normalisation layer (Ioffe & Szegedy, 2015) and then undoing it after the nonlinearity.

We can find an upper bound of the loss by writing

$$\mathcal{L}(\mathbf{W}, \boldsymbol{\sigma}) = \mathbb{E}(||\mathbf{x} - (h(\frac{\mathbf{x}\mathbf{W} - \bar{\boldsymbol{\mu}}_\mathbf{y}}{\boldsymbol{\sigma}})\boldsymbol{\sigma} + \bar{\boldsymbol{\mu}}_\mathbf{y})[\mathbf{W}^T]_{sg}||_2^2) \tag{206}$$

$$= \mathbb{E}(||\mathbf{x} - \bar{\boldsymbol{\mu}}_\mathbf{x} + \bar{\boldsymbol{\mu}}_\mathbf{x} - (h(\frac{\mathbf{x}\mathbf{W} - \bar{\boldsymbol{\mu}}_\mathbf{y}}{\boldsymbol{\sigma}})\boldsymbol{\sigma} + \bar{\boldsymbol{\mu}}_\mathbf{y})[\mathbf{W}^T]_{sg}||_2^2) \tag{207}$$

$$\leq \mathbb{E}(||\mathbf{x} - \bar{\boldsymbol{\mu}}_\mathbf{x} - (h(\frac{\mathbf{x}\mathbf{W} - \bar{\boldsymbol{\mu}}_\mathbf{y}}{\boldsymbol{\sigma}})\boldsymbol{\sigma})[\mathbf{W}^T]_{sg}||_2^2) + \mathbb{E}(||\bar{\boldsymbol{\mu}}_\mathbf{x} - \bar{\boldsymbol{\mu}}_\mathbf{y}[\mathbf{W}^T]_{sg}||_2^2). \tag{208}$$

The upper bound consists of the centred nonlinear PCA loss and a linear mean reconstruction loss. Due to the stop gradient operator, the latter has no effect when $\mathbf{W}$ is orthogonal (see Appendix A.2.1). We can instead change it to

$$\mathbb{E}(||\bar{\boldsymbol{\mu}}_\mathbf{x} - \bar{\boldsymbol{\mu}}_\mathbf{x}\mathbf{W}\mathbf{W}^T||_2^2), \tag{209}$$

resulting into the following non-centred loss:

$$\mathcal{L}(\mathbf{W}, \boldsymbol{\sigma}) = \mathbb{E}(||\mathbf{x} - \bar{\boldsymbol{\mu}}_{\mathbf{x}} - (h(\frac{\mathbf{xW} - \bar{\boldsymbol{\mu}}_{\mathbf{y}}}{\boldsymbol{\sigma}})\boldsymbol{\sigma})[\mathbf{W}^T]_{sg}||_2^2) + \mathbb{E}(||\bar{\boldsymbol{\mu}}_{\mathbf{x}} - \bar{\boldsymbol{\mu}}_{\mathbf{x}}\mathbf{WW}^T||_2^2). \quad (210)$$

# F    ADDITIONAL EXPERIMENTS

## F.1    NONLINEAR PCA

### F.1.1    TRAINABLE $\boldsymbol{\Sigma}$

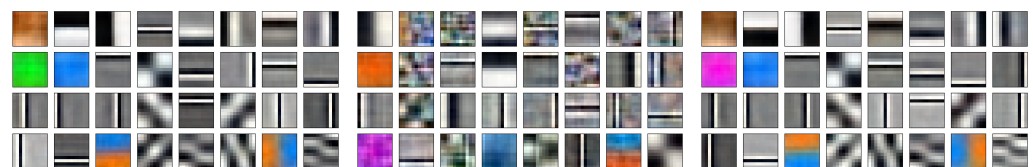

(a) Epoch 3 - without regularisation    (b) Epoch 10 - without regularisation    (c) Epoch 10 - with $1e^{-3}$ $L_2$ regularisation

Figure 6: Without any regularisation, $\sigma$ will tend to keep increasing and result in the degeneration of the filters. This does no occur when using the estimated standard deviations for $\sigma$.

### F.1.2    SCALING OF THE INPUT

Here we look at the effect of scaling the standardised input by multiplying it with a scalar.

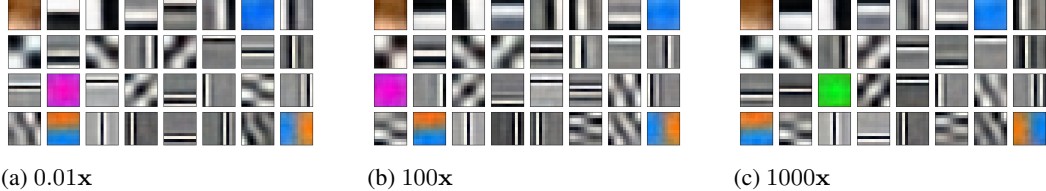

(a) $0.01\mathbf{x}$        (b) $100\mathbf{x}$        (c) $1000\mathbf{x}$

Figure 7: When using nonlinear PCA with estimated standard deviations, it automatically adapts to the scale of the input.

### F.1.3    SCALE OF TANH

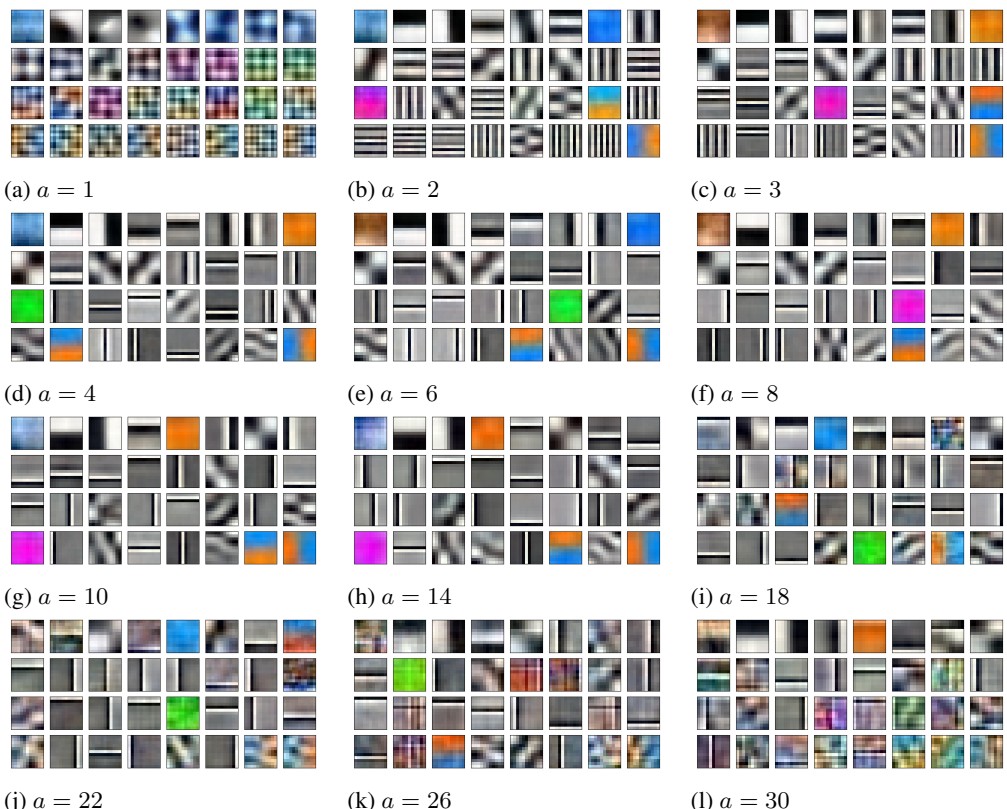

(a) $a = 1$

(b) $a = 2$

(c) $a = 3$

(d) $a = 4$

(e) $a = 6$

(f) $a = 8$

(g) $a = 10$

(h) $a = 14$

(i) $a = 18$

(j) $a = 22$

(k) $a = 26$

(l) $a = 30$

Figure 8: Showing the effect of varying the scalar $a$ in $a \tanh(x/a)$ on the obtained filters. We see that we start getting better defined filters from $a = 2$, but quite a few are still entangled/superimposed. We get better separation from at least $a = 3$. The filters start degenerating $a > 14$.

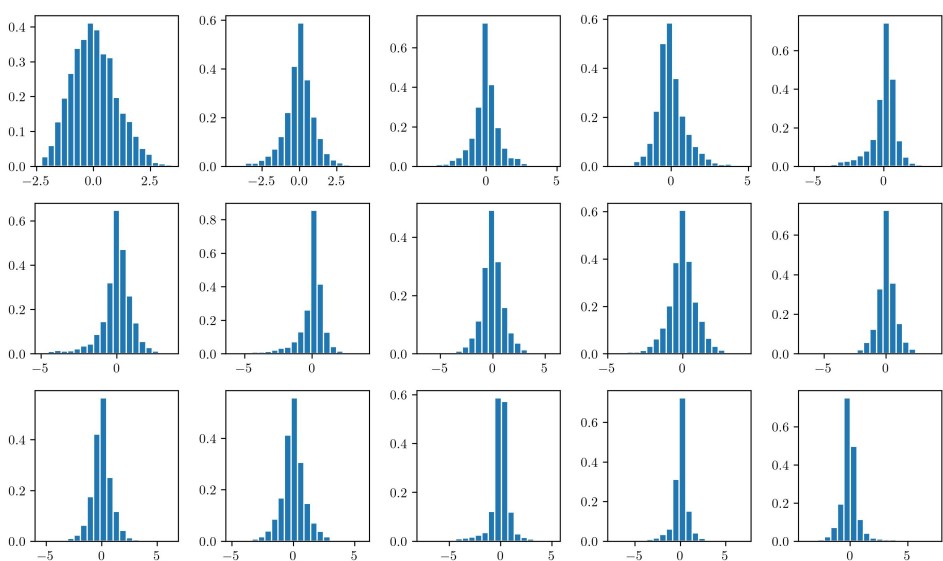

Figure 9: The distributions of the first 15 filters ordered by their variances. We see that they mostly tend to be super-Gaussian distributions, which is why a larger value of $a$ in $a \tanh(x/a)$ was needed.

### F.1.4 LATENT SPACE RECONSTRUCTION

Here we look at the reconstruction loss in latent space and look the filters obtained using the losses described in Section E.3, which we repeat here for convenience:

$$\mathcal{L}(\mathbf{W}, \boldsymbol{\Sigma}) = \mathbb{E}(||\mathbf{x}[\mathbf{W}]_{sg} - h(\frac{\mathbf{xW}}{\boldsymbol{\sigma}})\boldsymbol{\sigma}[\mathbf{W}^T\mathbf{W}]_{sg}||_2^2) + \beta||\mathbf{I} - \mathbf{W}^T\mathbf{W}||_F^2 \tag{145}$$

$$\mathcal{L}(\mathbf{W}, \boldsymbol{\Sigma}) = \mathbb{E}(||\mathbf{x}[\mathbf{W}]_{sg} - h(\frac{\mathbf{xW}}{\boldsymbol{\sigma}})\boldsymbol{\sigma}\triangle[\mathbf{W}^T\mathbf{W}]_{sg}||_2^2) + \beta||\mathbf{I} - \mathbf{W}^T\mathbf{W}||_F^2 \tag{146}$$

$$\mathcal{L}(\mathbf{W}, \boldsymbol{\Sigma}) = \mathbb{E}(||\mathbf{x}[\mathbf{W}]_{sg} - h(\frac{\mathbf{xW}}{\boldsymbol{\sigma}})\boldsymbol{\sigma}||_2^2) + \beta||\mathbf{I} - \mathbf{W}^T\mathbf{W}||_F^2. \tag{147}$$

$$\mathcal{L}(\mathbf{W}, \boldsymbol{\Sigma}) = \mathbb{E}(||\mathbf{x}[\mathbf{W}]_{sg} - h(\mathbf{xW\Sigma})\boldsymbol{\Sigma}||_2^2) + \mathbb{E}(||\mathbf{xW}[\mathbf{W}^T]_{sg} - \mathbf{x}||_2^2). \tag{148}$$

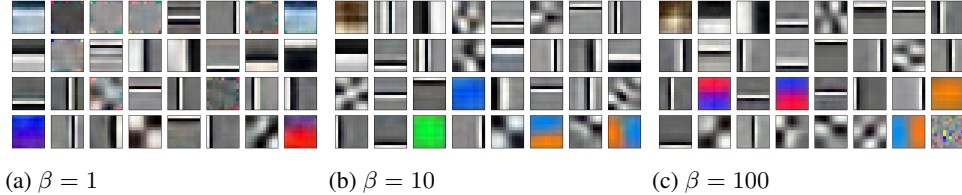

(a) $\beta = 1$        (b) $\beta = 0$        (c) $\beta = 1, \triangle[\mathbf{W}^T\mathbf{W}]_{sg}$

Figure 10: Using Eq. 145, we see that we do not require a strong gain on the orthogonal regulariser.

(a) $\beta = 1$        (b) $\beta = 10$        (c) $\beta = 100$

Figure 11: Using Eq. 147, we see that it benefits from having a larger gain on the orthogonal regulariser.

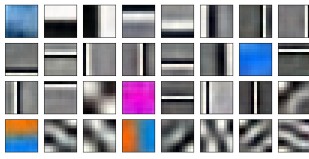

Figure 12: Using Eq. 148, we see that adding the encoder contribution from the linear reconstruction maintains the orthogonality of the components.

### F.1.5 ASYMMETRIC NONLINEAR FUNCTION WITH MODIFIED DERIVATIVE

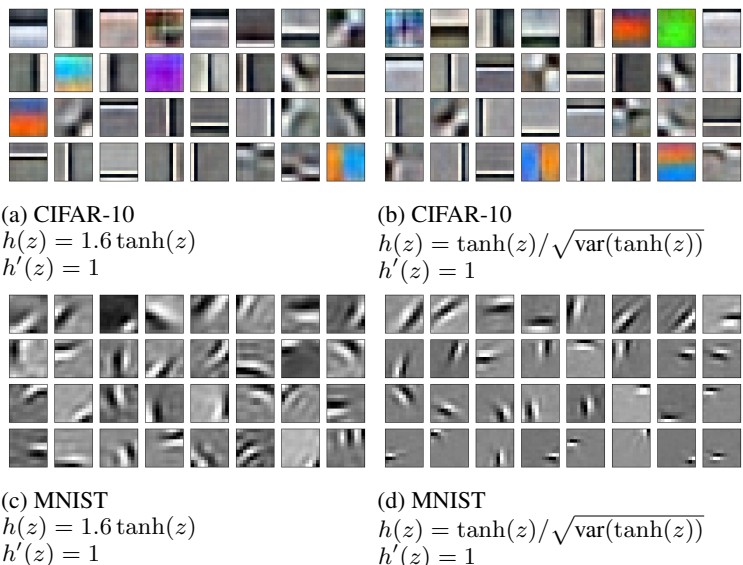

(a) CIFAR-10
$h(z) = 1.6 \tanh(z)$
$h'(z) = 1$

(b) CIFAR-10
$h(z) = \tanh(z)/\sqrt{\mathrm{var}(\tanh(z))}$
$h'(z) = 1$

(c) MNIST
$h(z) = 1.6 \tanh(z)$
$h'(z) = 1$

(d) MNIST
$h(z) = \tanh(z)/\sqrt{\mathrm{var}(\tanh(z))}$
$h'(z) = 1$

Figure 13: Filters obtained on CIFAR-10 and MNIST with an asymmetric activation function where $a$ is either a constant or adaptive.

### F.1.6 FIRST 64 FILTERS

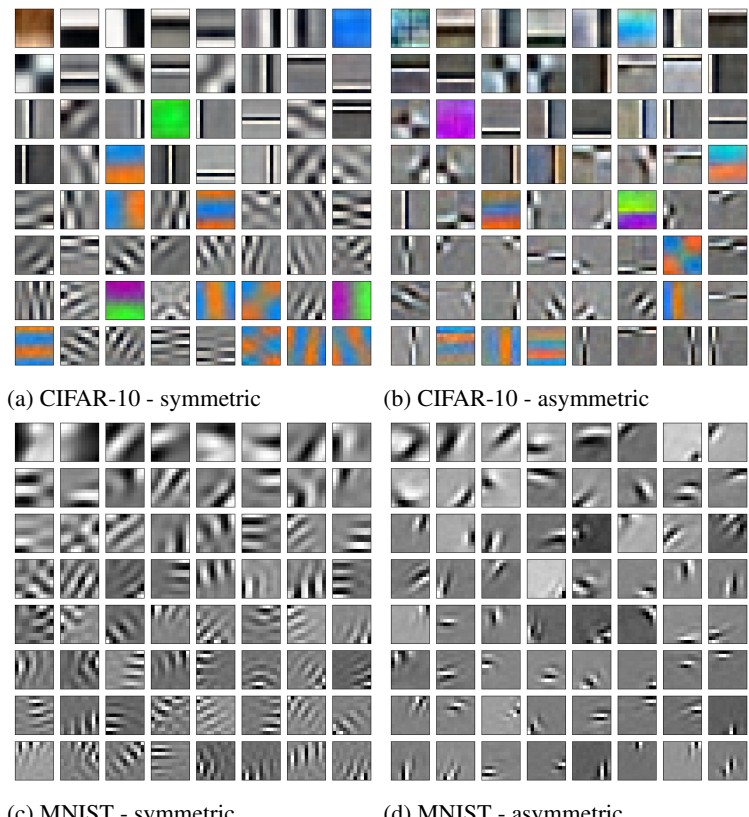

(a) CIFAR-10 - symmetric

(b) CIFAR-10 - asymmetric

(c) MNIST - symmetric

(d) MNIST - asymmetric

Figure 14: The first 64 filters, sorted by variance, obtained on CIFAR-10 and MNIST with either a symmetric activation function ($h(z) = 4\tanh(z/4)$) or an adapative asymmetric activation function ($h(z) = \tanh(z)/\sqrt{\mathrm{var}(\tanh(z))}, h'(z) = 1$). We can note that the obtained filters seem to be more localised in the asymmetric case compared to the symmetric case.

## F.2  $L_1$ SPARSITY - RICA

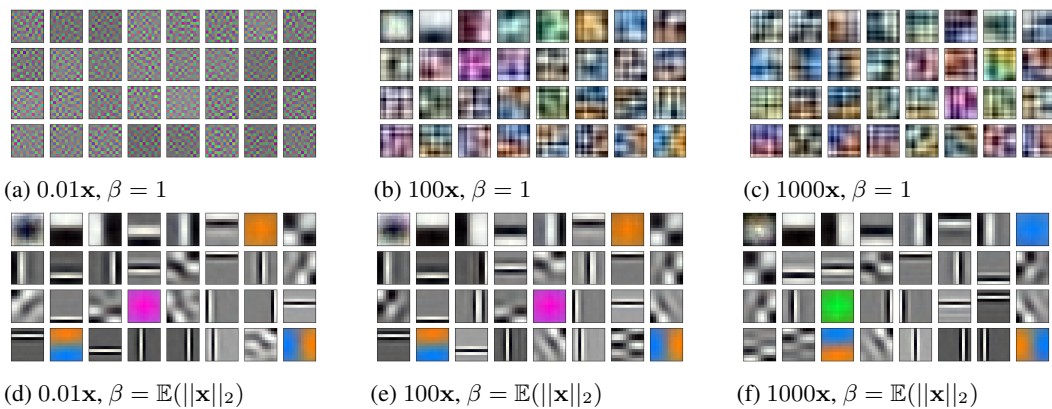

(a) 0.01**x**, $\beta = 1$  (b) 100**x**, $\beta = 1$  (c) 1000**x**, $\beta = 1$

(d) 0.01**x**, $\beta = \mathbb{E}(||\mathbf{x}||_2)$  (e) 100**x**, $\beta = \mathbb{E}(||\mathbf{x}||_2)$  (f) 1000**x**, $\beta = \mathbb{E}(||\mathbf{x}||_2)$

Figure 15: When using RICA E.4, the strength of the sparsity regulariser needs to be adjusted to adapt to the scale of the input. Here we show the effect of the input scale on the obtained filters between adaptive and nonadaptive $\beta$. (a-c) have $\beta = 1$, while (d-f) have $\beta = \mathbb{E}(||\mathbf{x}||_2)$. We see that making $\beta$ proportional to $\mathbb{E}(||\mathbf{x}||_2)$ makes it invariant to the scale of the input.

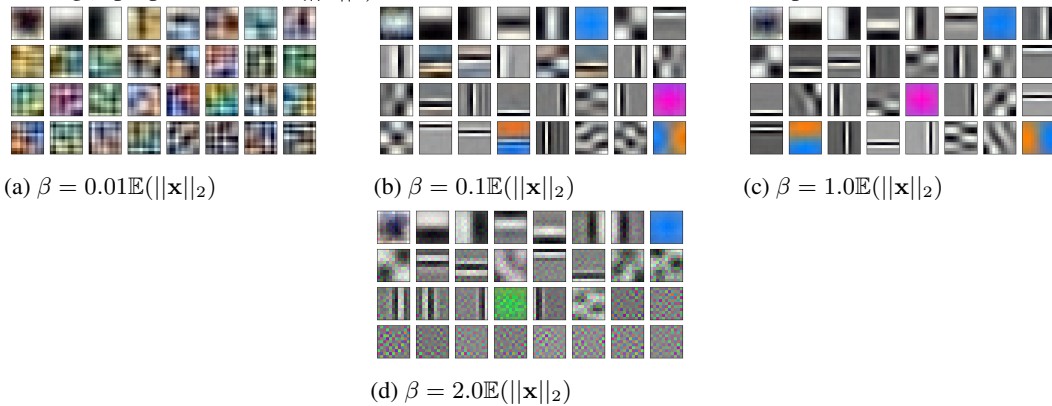

(a) $\beta = 0.01\mathbb{E}(||\mathbf{x}||_2)$  (b) $\beta = 0.1\mathbb{E}(||\mathbf{x}||_2)$  (c) $\beta = 1.0\mathbb{E}(||\mathbf{x}||_2)$

(d) $\beta = 2.0\mathbb{E}(||\mathbf{x}||_2)$

Figure 16: Filters obtained on CIFAR-10 using RICA E.4 with varying $L_1$ sparsity regularisation intensity. Unit weight normalisation was used (see Appendix C).

### F.3  LINEAR PCA

Here we summarise variations of linear PCA methods.

| Method | Filters |
|---|---|
| PCA via SVD (A.1) | |
| Linear autoencoder (A.2.1)

$\mathcal{L}(\mathbf{W}) = \mathbb{E}(||\mathbf{x} - \mathbf{x}\mathbf{W}\mathbf{W}^T||_2^2)$ | |

Subspace learning algorithm (A.2.2)

$$\Delta \mathbf{W} \propto \mathbf{x}^T \mathbf{y} - \mathbf{W}\mathbf{y}^T \mathbf{y}$$

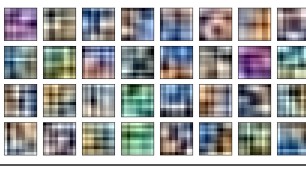

Weighted subspace learning algorithm (WSLA) - variant 1 (A.3.1)

$$\Delta \mathbf{W} \propto \mathbf{x}^T \mathbf{y} - \mathbf{W}\mathbf{y}^T \mathbf{y}\mathbf{\Lambda}^{-1}$$
$$1 \geq \lambda_1 > ... > \lambda_k > 0$$

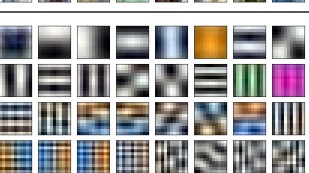

WSLA - variant 2 (A.3.1)

$$\Delta \mathbf{W} \propto \mathbf{x}^T \mathbf{y}\mathbf{\Lambda} - \mathbf{W}\mathbf{y}^T \mathbf{y}$$
$$1 \geq \lambda_1 > ... > \lambda_k > 0$$

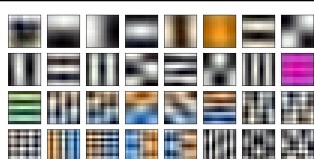

WSLA - variant 3 (A.3.1)

$$\Delta \mathbf{W} \propto \mathbf{x}^T \mathbf{y} - \mathbf{W}(\mathbf{y}\mathbf{\Lambda}^{\frac{1}{2}})^T \mathbf{y}\mathbf{\Lambda}^{-\frac{1}{2}}$$
$$1 \geq \lambda_1 > ... > \lambda_k > 0$$

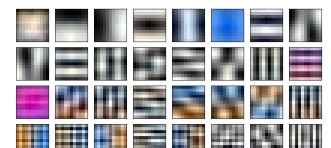

Weighted subspace algorithm with unit norm (A.3.2)

$$\mathcal{L}(\mathbf{W}) = \mathbb{E}(||\mathbf{x}[\mathbf{W}]_{sg}\mathbf{W}^T - \mathbf{x}||_2^2$$
$$+ ||\mathbf{W}\hat{\mathbf{\Sigma}}\mathbf{\Lambda}^{\frac{1}{2}}||_2^2 - ||\mathbf{W}\hat{\mathbf{\Sigma}}||_2^2$$
$$- ||\mathbf{x}\mathbf{W}\mathbf{\Lambda}^{\frac{1}{2}}||_2^2 + ||\mathbf{x}\mathbf{W}||_2^2)$$

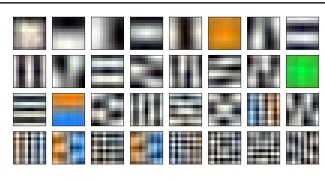

Generalised hebbian algorithm (GHA) (A.3.3)

$$\mathbf{y} = \mathbf{x}\mathbf{W}$$
$$\mathbf{\Delta W} \propto \mathbf{x}^T \mathbf{y} - \mathbf{W}_\triangle(\mathbf{y}^T \mathbf{y})$$

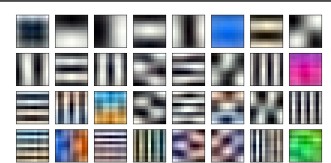

GHA with encoder contribution (A.3.4)

$$\Delta \mathbf{W} \propto \quad \mathbf{x}^T \mathbf{y} - \mathbf{W}_\triangle(\mathbf{y}^T \mathbf{y})$$
$$- \mathbf{x}^T \mathbf{y}(\triangle(\mathbf{W}^T \mathbf{W}) - \mathbf{I})$$

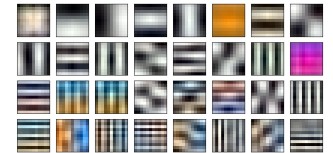

GHA + subspace learning algorithm (A.3.4)

$$\Delta \mathbf{W} \propto \quad \mathbf{x}^T \mathbf{y} - \mathbf{W}_\triangle(\mathbf{y}^T \mathbf{y})$$
$$- \mathbf{W}(\mathbf{y}^T \mathbf{y} - \text{diag}(\mathbf{y}^2))$$

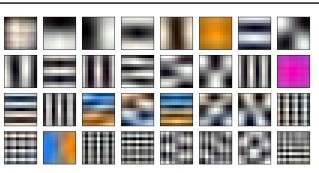

Reconstruction + GHA (A.3.4)

$$\mathcal{L}(\mathbf{W}) = \mathbb{E}(||\mathbf{x} - \mathbf{x}\mathbf{W}\mathbf{W}^T||_2^2$$
$$+ \mathbf{1}\mathbf{W} \odot [\mathbf{W}_\triangle(\mathbf{y}^T \mathbf{y})]_{sg}\mathbf{1}^T)$$

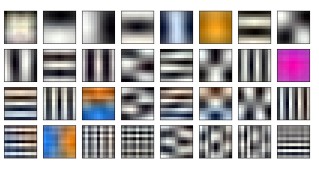

Nested dropout (A.3.5)

$$\mathcal{L}(\mathbf{W}) = \mathbb{E}(||(\mathbf{x}\mathbf{W} \odot \mathbf{m}_{1|j})\mathbf{W}^T - \mathbf{x}||_2^2)$$

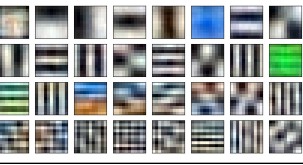

Variance maximiser with weighted regulariser (A.3.6)

$$J(\mathbf{W}) = \mathbb{E}(||\mathbf{x} - \mathbf{x}\mathbf{W}\mathbf{W}^T||_2^2 - ||\mathbf{x}\mathbf{W}\mathbf{\Lambda}^{\frac{1}{2}}||_2^2)$$
$$1 \geq \lambda_1 > ... > \lambda_k > 0$$

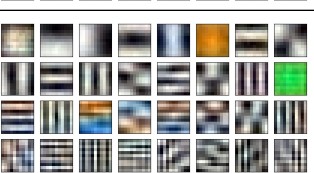

Variance maximiser regulariser with nested dropout (A.3.6)

$$\mathcal{L}(\mathbf{W}) = \mathbb{E}(||\mathbf{x} - \mathbf{x}\mathbf{W}\mathbf{W}^T||_2^2$$
$$- ||\mathbf{x}\mathbf{W} \cdot \mathbf{m}_{1|j}||_2^2)$$

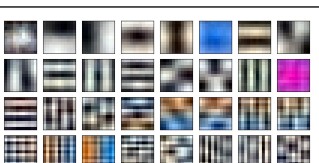

## F.4 NONLINEAR PCA

Here we summarise variations of nonlinear PCA methods.

| Method | Filters |
|---|---|
| Conventional without whitening (2.2) $$\mathcal{L}(\mathbf{W}) = \mathbb{E}(||\mathbf{x} - h(\mathbf{x}\mathbf{W})\mathbf{W}^T||_2^2)$$ | |
| Differentiable $\boldsymbol{\sigma}$ - without stop gradient (3) $$\mathcal{L}(\mathbf{W}, \mathbf{\Sigma}) = \mathbb{E}(||\mathbf{x} - h(\mathbf{x}\mathbf{W}\mathbf{\Sigma}^{-1})\mathbf{\Sigma}\mathbf{W}^T||_2^2)$$ | |
| Differentiable $\boldsymbol{\sigma}$ (Eq. 12) $$\mathcal{L}(\mathbf{W}, \mathbf{\Sigma}) = \mathbb{E}(||\mathbf{x} - h(\mathbf{x}\mathbf{W}\mathbf{\Sigma}^{-1})\mathbf{\Sigma}[\mathbf{W}^T]_{sg}||_2^2)$$ | |
| Differentiable $\boldsymbol{\sigma}$ - latent reconstruction with reconstruction orthogonal regulariser (E.3) $$\mathcal{L}(\mathbf{W}, \mathbf{\Sigma}) = \mathbb{E}(||\mathbf{x}[\mathbf{W}]_{sg} - h(\mathbf{x}\mathbf{W}\mathbf{\Sigma})\mathbf{\Sigma}||_2^2)$$ $$+ \mathbb{E}(||\mathbf{x}\mathbf{W}[\mathbf{W}^T]_{sg} - \mathbf{x}||_2^2)$$ | |

Differentiable $\boldsymbol{\sigma}$ - latent reconstruction with symmetric orthogonal regulariser (E.3)

$$\mathcal{L}(\mathbf{W}, \boldsymbol{\Sigma}) = \mathbb{E}(||\mathbf{x}[\mathbf{W}]_{sg} - h(\mathbf{x}\mathbf{W}\boldsymbol{\Sigma})\boldsymbol{\Sigma}||_2^2)$$
$$+ \beta||\mathbf{I} - \mathbf{W}^T\mathbf{W}||_F^2$$
$$\beta = 10$$

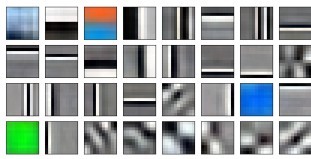

Differentiable $\boldsymbol{\sigma}$ - latent reconstruction with asymmetric orthogonal regulariser (E.3)

$$\mathcal{L}(\mathbf{W}, \boldsymbol{\Sigma}) = \mathbb{E}(||\mathbf{x}[\mathbf{W}]_{sg} - h(\mathbf{x}\mathbf{W}\boldsymbol{\Sigma})\boldsymbol{\Sigma}||_2^2)$$
$$+ \beta||(\triangle(\mathbf{W}^T[\mathbf{W}]_{sg}))||_F^2$$
$$\beta = 10$$

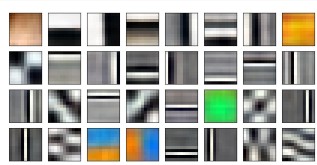

Differentiable $\boldsymbol{\sigma}$ - latent reconstruction with $[\mathbf{W}^T\mathbf{W}]_{sg}$ and symmetric orthogonal regulariser (E.3)

$$\mathcal{L}(\mathbf{W}, \boldsymbol{\Sigma}) = \mathbb{E}(||\mathbf{x}[\mathbf{W}]_{sg} - h(\frac{\mathbf{x}\mathbf{W}}{\boldsymbol{\sigma}})\boldsymbol{\sigma}[\mathbf{W}^T\mathbf{W}]_{sg}||_2^2)$$
$$+ ||\mathbf{I} - \mathbf{W}^T\mathbf{W}||_F^2$$

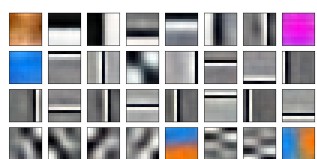

EMA $\boldsymbol{\sigma}$ - tanh (3)

$$\mathcal{L}(\mathbf{W}) = \mathbb{E}(||\mathbf{x} - h(\mathbf{x}\mathbf{W}\boldsymbol{\Sigma}^{-1})\boldsymbol{\Sigma}[\mathbf{W}^T]_{sg}||_2^2)$$
$$h(x) = \tanh(x)$$

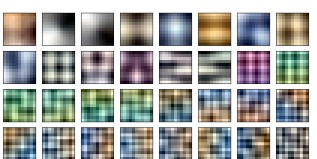

EMA $\boldsymbol{\sigma}$ - scaled tanh (3.3)

$$\mathcal{L}(\mathbf{W}) = \mathbb{E}(||\mathbf{x} - h(\mathbf{x}\mathbf{W}\boldsymbol{\Sigma}^{-1})\boldsymbol{\Sigma}[\mathbf{W}^T]_{sg}||_2^2)$$
$$h(x) = 4\tanh(\frac{x}{4})$$

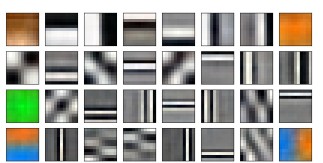

EMA $\boldsymbol{\sigma}$ - scaled tanh with modified derivative (Eq. 117)

$$\mathcal{L}(\mathbf{W}) = \mathbb{E}(||\mathbf{x} - h(\mathbf{x}\mathbf{W}\boldsymbol{\Sigma}^{-1})\boldsymbol{\Sigma}[\mathbf{W}^T]_{sg}||_2^2)$$
$$h(x) = 4\tanh(\frac{x}{4})$$
$$h'(x) = 1$$

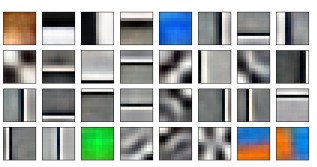

EMA $\boldsymbol{\sigma}$ - with hard tanh approximation (3.3)

$$\mathcal{L}(\mathbf{W}) = \mathbb{E}(||\mathbf{x} - h(\mathbf{x}\mathbf{W}\boldsymbol{\Sigma}^{-1})\boldsymbol{\Sigma}[\mathbf{W}^T]_{sg}||_2^2)$$
$$h(x) = max(-2, min(2, 0))$$
$$h'(x) = 1$$

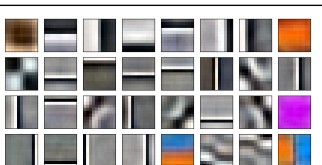

Asymmetric activation - adaptive (E.1.1)

$$\mathcal{L}(\mathbf{W}) = \mathbb{E}(||\mathbf{x} - h(\mathbf{xW\Sigma}^{-1})\mathbf{\Sigma}[\mathbf{W}^T]_{sg}||_2^2)$$
$$h(\mathbf{x}) = \mathbf{a}\tanh(\mathbf{x})$$
$$h'(x) = 1$$
$$\mathbf{a} = 1/[\sqrt{\text{var}(h(\mathbf{z}))}]_{sg}$$

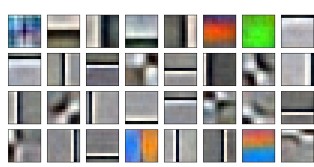

Asymmetric activation - constant (E.1.1)

$$\mathcal{L}(\mathbf{W}) = \mathbb{E}(||\mathbf{x} - h(\mathbf{xW\Sigma}^{-1})\mathbf{\Sigma}[\mathbf{W}^T]_{sg}||_2^2)$$
$$h(\mathbf{x}) = 1.6\tanh(\mathbf{x})$$
$$h'(x) = 1$$

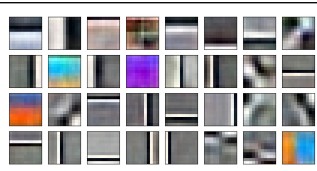

EMA $\boldsymbol{\sigma}$ - latent reconstruction with symmetric orthogonal regulariser - scaled tanh (E.3)

$$\mathcal{L}(\mathbf{W}, \mathbf{\Sigma}) = \mathbb{E}(||\mathbf{x}[\mathbf{W}]_{sg} - h(\mathbf{xW\Sigma})\mathbf{\Sigma}[\mathbf{W}^T\mathbf{W}]_{sg}||_2^2)$$
$$+ ||\mathbf{I} - \mathbf{W}^T\mathbf{W}||_F^2$$
$$h(x) = 4\tanh(\frac{x}{4})$$

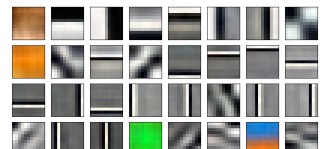

EMA $\boldsymbol{\sigma}$ - differentiable scaled tanh (3.3)

$$\mathcal{L}(\mathbf{W}, a) = \mathbb{E}(||\mathbf{x} - h_a(\mathbf{xW\Sigma}^{-1})\mathbf{\Sigma}[\mathbf{W}^T]_{sg}||_2^2)$$
$$h_a(x) = a\tanh(\frac{x}{a})$$

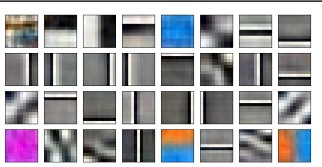

EMA $\boldsymbol{\sigma}$ - baked-in GS. No need to order the components as it is done automatically. (3.5)

$$\mathcal{L}(\mathbf{W}) = \mathbb{E}(||\mathbf{x} - h(\mathbf{xW}P(\mathbf{W})\mathbf{\Sigma}^{-1})\mathbf{\Sigma}[\mathbf{W}^T]_{sg}||_2^2)$$
$$P(\mathbf{W}) = (I - \triangle(\mathbf{W}^T[\mathbf{W}]_{sg})$$

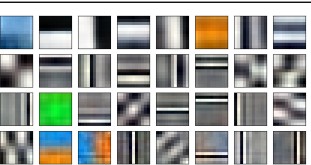

EMA $\boldsymbol{\sigma}$ - scaled tanh with nested dropout (E.7.5)

$$\mathcal{L}(\mathbf{W}) = \mathbb{E}(||\mathbf{x} - h(\mathbf{xW\Sigma}^{-1}) \odot \mathbf{m}_{1|j}\mathbf{\Sigma}[\mathbf{W}^T]_{sg}||_2^2)$$
$$+ ||\mathbf{W}^T\mathbf{W} - \mathbf{I}||_F^2$$
$$h(x) = 4\tanh(\frac{x}{4})$$

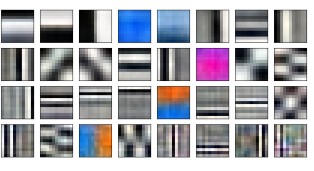

EMA $\boldsymbol{\sigma}$ - modified tanh derivative - with linear reconstruction (E.1.1)

$$\mathcal{L}(\mathbf{W}) = \mathbb{E}(||\mathbf{x} - h(\mathbf{xW\Sigma}^{-1})\mathbf{\Sigma}[\mathbf{W}^T]_{sg}||_2^2)$$
$$+ \alpha\mathbb{E}(||\mathbf{x} - \mathbf{xW}[\mathbf{W}^T]_{sg}||_2^2)$$
$$h(x) = \tanh(x)$$
$$h'(x) = 1$$
$$\alpha \geq 2$$

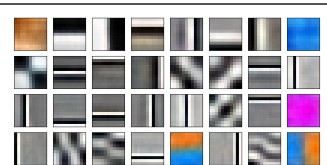

RICA (E.4)

$$\mathcal{L}(\mathbf{W}) = \mathbb{E}(||\mathbf{x} - \mathbf{x}\mathbf{W}\mathbf{W}^T||_2^2 + \beta \sum |\mathbf{x}\mathbf{w_j^T}|)$$
$$\beta = \mathbb{E}(||\mathbf{x}||_2)$$

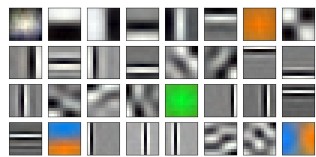

Reconstruction with log cosh regulariser (E.4)

$$\mathcal{L}(\mathbf{W}) = \mathbb{E}(||\mathbf{x} - \mathbf{x}\mathbf{W}\mathbf{W}^T||_2^2 + \beta \sum \log\cosh(\mathbf{x}\mathbf{w_j^T}))$$

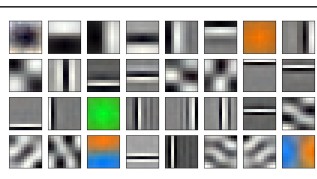

Reconstruction skew symmetric $\mathbf{y}^T h(\mathbf{y})$ (E.5)

$$\mathcal{L}(\mathbf{W}) = \mathbb{E}(||\mathbf{x} - \mathbf{x}\mathbf{W}\mathbf{W}^T||_2^2$$
$$+ \beta \mathbf{1}\mathbf{W} \odot [\mathbf{W}(\mathbf{y}^T h(\mathbf{y}) - h(\mathbf{y})^T\mathbf{y}))]_{sg}\mathbf{1}^T)$$
$$h = \tanh$$
$$\beta = \mathbb{E}(||\mathbf{x}||_2)$$

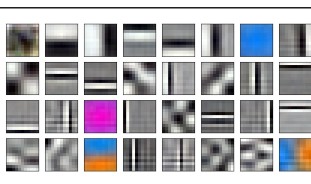

Reconstruction with symmetric $\mathbf{y}^T h(\mathbf{y})$ without diagonal (E.5)

$$\mathcal{L}(\mathbf{W}) = \mathbb{E}(||\mathbf{x} - \mathbf{x}\mathbf{W}\mathbf{W}^T||_2^2$$
$$+ \beta \mathbf{1}\mathbf{W} \odot [\mathbf{W}(\mathbf{y}^T h(\mathbf{y}) - \mathrm{diag}(h(\mathbf{y}) \odot \mathbf{y})))]_{sg}\mathbf{1}^T)$$
$$h = \tanh$$
$$\beta = \mathbb{E}(||\mathbf{x}||_2)$$

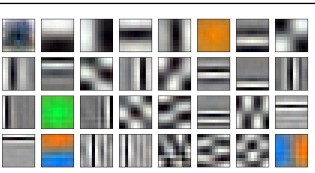

Reconstruction with symmetric $\mathbf{y}^T h(\mathbf{y}/\boldsymbol{\sigma})\boldsymbol{\sigma}$ without diagonal (E.5)

$$\mathcal{L}(\mathbf{W}) = \mathbb{E}(||\mathbf{x} - \mathbf{x}\mathbf{W}\mathbf{W}^T||_2^2$$
$$+ \mathbf{1}\mathbf{W} \odot [\mathbf{W}(\mathbf{y}^T h(\mathbf{y}\boldsymbol{\Sigma}^{-1})\boldsymbol{\Sigma}$$
$$- \mathrm{diag}(h(\mathbf{y}\boldsymbol{\Sigma}^{-1})\boldsymbol{\Sigma} \odot \mathbf{y})))]_{sg}\mathbf{1}^T)$$
$$h = \mathrm{sign}$$

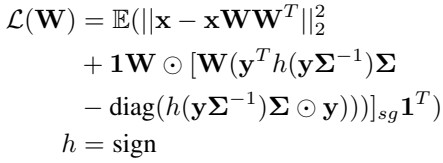
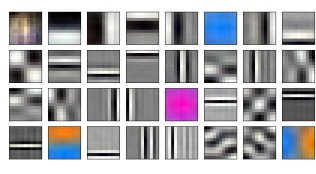

EMA $\boldsymbol{\sigma}$ - scaled tanh with triangular update (E.7.2)

$$\mathcal{L}(\mathbf{W}) = \mathbb{E}(||\mathbf{x} - h(\mathbf{x}\mathbf{W}\boldsymbol{\Sigma}^{-1})\boldsymbol{\Sigma}[\mathbf{W}^T]_{sg}||_2^2$$
$$+ \mathbf{1}\mathbf{W} \odot [\mathbf{W} \triangledown (\mathbf{y}^T\mathbf{y}\boldsymbol{\Sigma}^{-1})]_{sg}\mathbf{1}^T)$$
$$h(x) = 6\tanh(\frac{x}{6})$$

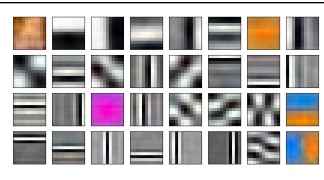

### F.5 TIME SERIES

#### F.5.1 ORTHOGONAL - SAME VARIANCE

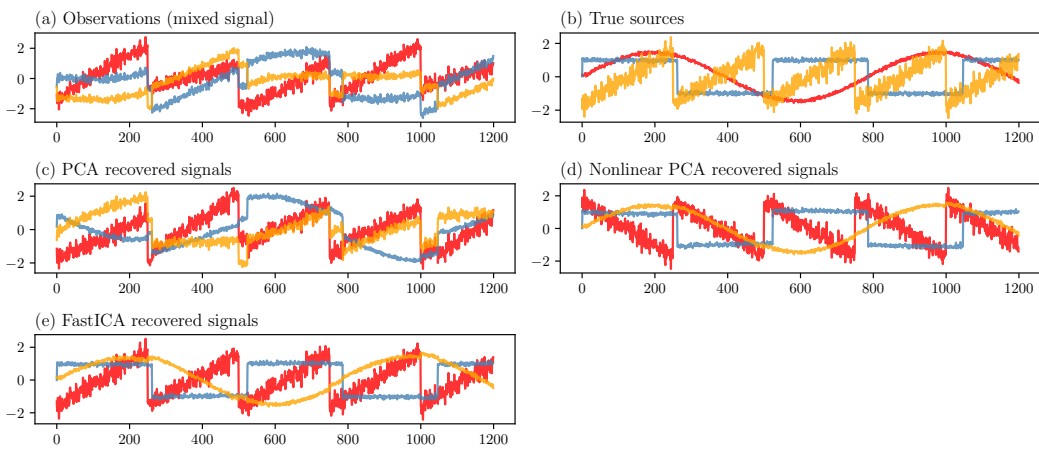

Figure 17: Three signals (sinusoidal, square, and sawtooth) that were mixed with an *orthogonal* mixing matrix. Linear PCA is unable to separate the signals as they all have the same variance.

#### F.5.2 ORTHOGONAL - ALL DISTINCT VARIANCES

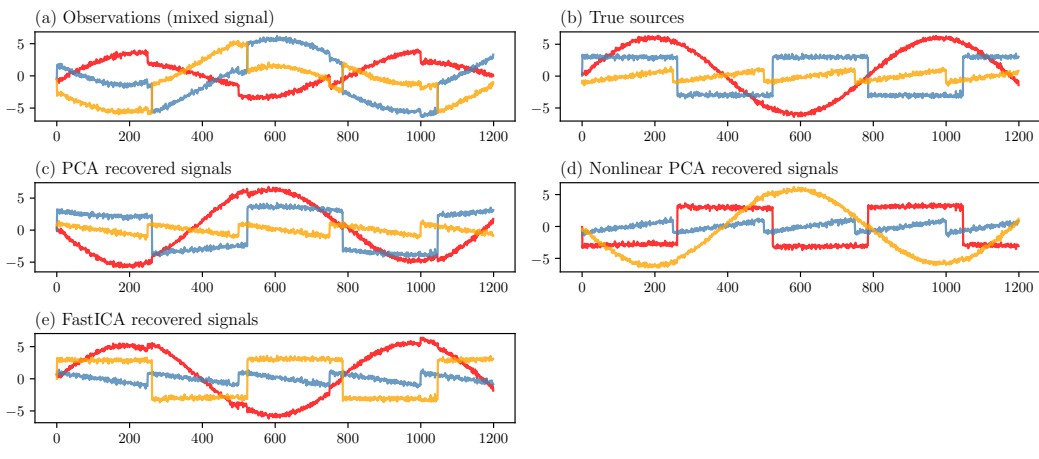

Figure 18: Three signals (sinusoidal, square, and sawtooth) that were mixed with an *orthogonal* mixing matrix. All three manage to separate the signals.

### F.6 SUB- AND SUPER- GAUSSIAN 2D POINTS

Here we look at the effect of applying linear PCA, nonlinear PCA, and ICA on 2D points ($n = 1000$) sampled from either a uniform distribution (sub-Gaussian) or a Laplace distribution (super-Gaussian). Figures 19 and 20 summarise the results.

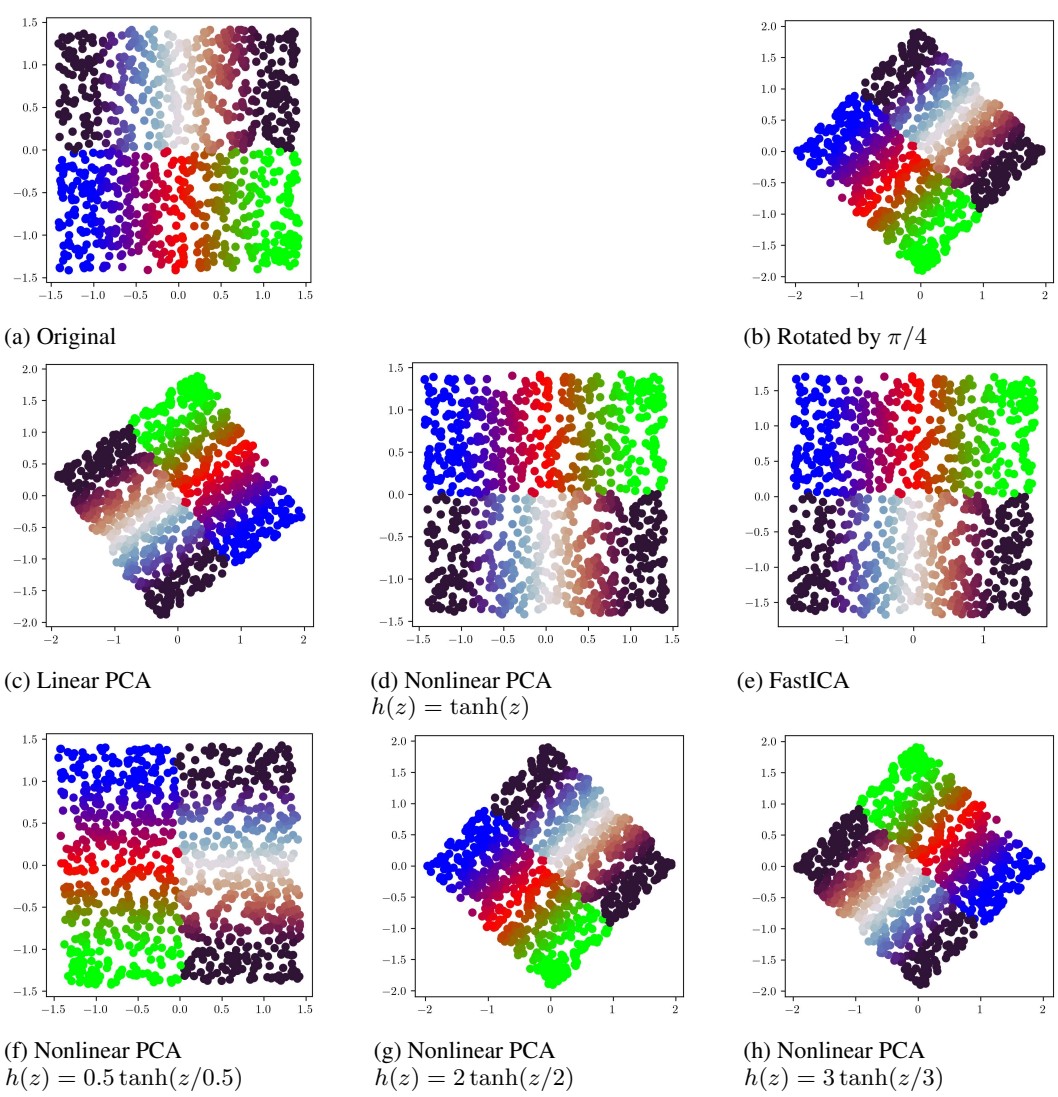

(a) Original

(b) Rotated by $\pi/4$

(c) Linear PCA

(d) Nonlinear PCA
$h(z) = \tanh(z)$

(e) FastICA

(f) Nonlinear PCA
$h(z) = 0.5\tanh(z/0.5)$

(g) Nonlinear PCA
$h(z) = 2\tanh(z/2)$

(h) Nonlinear PCA
$h(z) = 3\tanh(z/3)$

Figure 19: Uniform distribution (sub-Gaussian) with equal variance. The original data (a) is rotated by $\pi/4$ (b), then we attempt to recover the original data from the rotated data. We see that linear PCA (c) fails to recover the original data. Both nonlinear PCA (d) and FastICA (e) manage to recover the original data (up to sign and order permutation, given the equal variance). We also see that $h(z) = a\tanh(z/a)$ with $a \leq 1$ work best for the uniform distribution with nonlinear PCA.

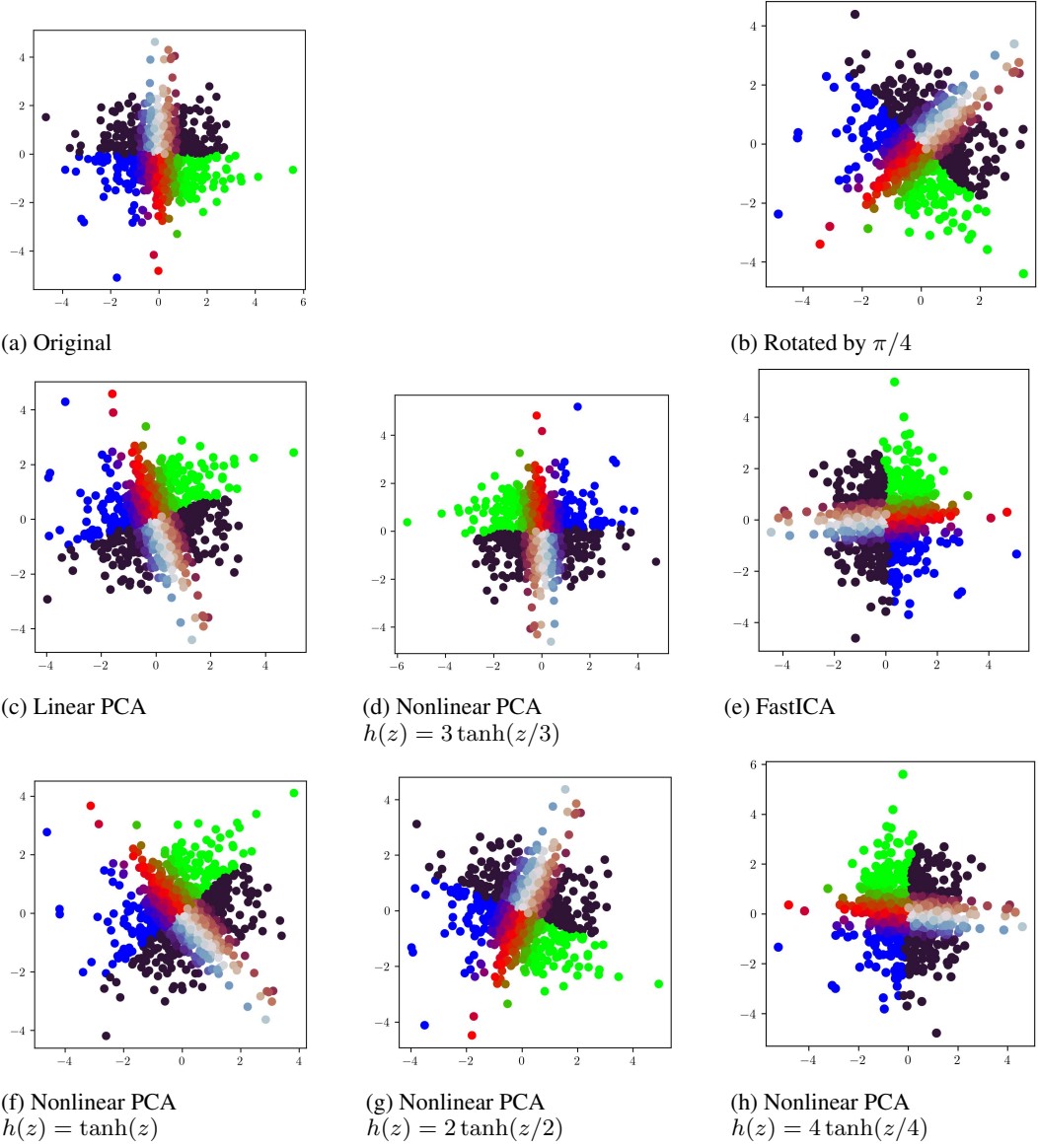

(a) Original

(b) Rotated by $\pi/4$

(c) Linear PCA

(d) Nonlinear PCA
$h(z) = 3\tanh(z/3)$

(e) FastICA

(f) Nonlinear PCA
$h(z) = \tanh(z)$

(g) Nonlinear PCA
$h(z) = 2\tanh(z/2)$

(h) Nonlinear PCA
$h(z) = 4\tanh(z/4)$

Figure 20: Laplace distribution (super-Gaussian) with equal variance. The original data (a) is rotated by $\pi/4$ (b), then we attempt to recover the original data from the rotated data. We see that linear PCA (c) fails to recover the original data. Both nonlinear PCA (d) and FastICA (e) manage to recover the original data (up to sign and order permutation, given the equal variance). We also see that $h(z) = a\tanh(z/a)$ with $a \geq 3$ work best for the Laplace distribution with nonlinear PCA.

## G   TRAINING DETAILS

**patches**   We used the Adam optimiser (Kingma & Ba, 2014) with a learning rate either of 0.01 or 0.001, $\beta_1 = 0.9$ and $\beta_2 = 0.999$. We used a batch size of 128. For the patches, each epoch consisted of 4K iterations, and, unless stated otherwise, we trained for a total of three epochs. With the majority of the nonlinear PCA methods, we used a projective unit norm constraint (see Appendix C).

**Time signals**   We used the vanilla SGD optimiser with a learning rate of 0.01 and momentum 0.9. We used a batch size of 100 and trained for a total of 200 epochs and picked the weights with the lowest reconstruction loss. When using the Adam optimiser, we noted that it seemed better to also use differentiable weight normalisation in addition to the projective unit norm constraint.

**2D points**   We used the vanilla SGD optimiser with a learning rate of 0.01 and momentum 0.9. We used a batch size of 100 and trained for a total of 100 epochs.

## H   BLOCK ROTATION MATRIX RS = SR

$$\mathbf{SR} = \begin{pmatrix} s_1\mathbf{I}_{2\times2} & \mathbf{0} \\ \mathbf{0} & s_2 \end{pmatrix} \begin{pmatrix} \mathbf{R}_{2\times2} & \mathbf{0} \\ \mathbf{0} & 1 \end{pmatrix} = \begin{pmatrix} s_1\mathbf{R}_{2\times2} & \mathbf{0} \\ \mathbf{0} & s_2 \end{pmatrix} = \begin{pmatrix} \mathbf{R}_{2\times2} & \mathbf{0} \\ \mathbf{0} & 1 \end{pmatrix} \begin{pmatrix} s_1\mathbf{I}_{2\times2} & \mathbf{0} \\ \mathbf{0} & s_2 \end{pmatrix} = \mathbf{RS}$$

(211)

## I   EXPONENTIAL MOVING AVERAGE (EMA)

$$\bar{\boldsymbol{\mu}} = \frac{1}{b}\sum \mathbf{y}_i \tag{212}$$

$$\bar{\boldsymbol{\sigma}}^2 = \frac{1}{b}\sum (\mathbf{y}_i - \bar{\boldsymbol{\mu}})^2 \tag{213}$$

$$\hat{\boldsymbol{\mu}} = \alpha\hat{\boldsymbol{\mu}} + (1-\alpha)[\bar{\boldsymbol{\mu}}]_{sg} \tag{214}$$

$$\hat{\boldsymbol{\sigma}}^2 = \alpha\hat{\boldsymbol{\sigma}}^2 + (1-\alpha)[\bar{\boldsymbol{\sigma}}^2]_{sg} \tag{215}$$

$$\text{Batch} \quad n_{\mathrm{B}}(\mathbf{y}) = \frac{\mathbf{y} - \bar{\boldsymbol{\mu}}}{\sqrt{\bar{\boldsymbol{\sigma}}^2 + \epsilon}} \tag{216}$$

$$\text{EMA} \quad n_{\mathrm{E}}(\mathbf{y}) = \frac{\mathbf{y} - \hat{\boldsymbol{\mu}}}{\sqrt{\hat{\boldsymbol{\sigma}}^2 + \epsilon}} \tag{217}$$

where $[\quad]_{sg}$ is the stop gradient operator, $\alpha$ is the momentum, and $b$ is the batch size.

## J   NONLINEAR FUNCTIONS

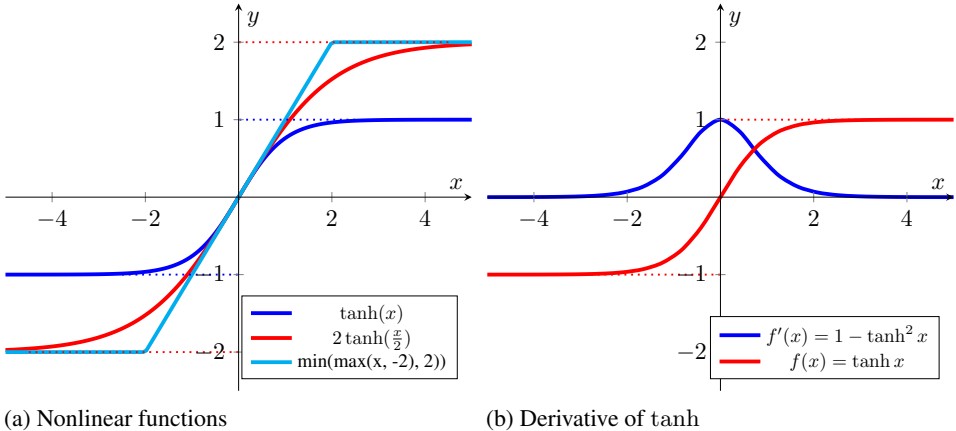

(a) Nonlinear functions

(b) Derivative of $\tanh$

Figure 21: We see in (a) examples of $\tanh$, a scaled $\tanh$, and a crude approximation of $\tanh$

