# OpenReview forum: "$\sigma$-PCA: a unified neural model for linear and nonlinear principal component analysis"
_ICLR.cc/2024/Conference — ICLR 2024 Conference Withdrawn Submission_

### Official Review · Reviewer_cVoi · 2023-10-30

**Soundness:** 1 poor
**Presentation:** 2 fair
**Contribution:** 2 fair
**Rating:** 3
**Confidence:** 3

**Summary:**

A new framework for understanding both linear PCA and nonlinear PCA as single-layer autoencoders is introduced. This allows one to reconcile the fact that linear PCA decomposes data into an orthogonal bases with ordered eigenvalues representing coefficients in that basis, whereas non-linear PCA decomposes data into non-orthogonal bases with coefficients that cannot be estimated. In this framework, both PCA and nonlinear PCA retain orthogonal bases and ordered variances. Independent component analysis is obtained as a special case where the variances have magnitude 1. Nonlinear PCA is discussed as a middle ground in the context of PCA and ICA. Nonlinear PCA is applied to CIFAR10 and some synthetic timeseries.

**Strengths:**

- Understanding connections between the various classical unsupervised methods such as PCA, ICA, autoencoders, nonlinear PCA, etc. is an interesting and important research direction, especially if it helps us design better variants of known existing algorithms.
- The paper cites a large amount of classical literature (I only have two minor additions below from 2001 and 2008), which I think is good because it mitigates against the increasing risk that old known results are forgotten.

**Weaknesses:**

I am excited by this work and think it has potential. I state my concerns below bluntly and directly and do not mean to discourage the authors. If the authors can provide a sound rebuttal or if I have overlooked something, I am happy to raise my score accordingly.

- The biggest weakness, in my opinion, is the imprecision/informality of the presentation. I am not sure whether the derivation is correct/rigorous/precise, because at each step and equation, it is not clear whether the original model is modified to accommodate the step in question, whether an approximation is applied, and whether the approximation or modification is valid. There is no discussion or attempt to quantify the effect of these various simplifications, or even formally state what these simplifications are.
    - As far as I can tell, my confusion begins at equation (5), where the weights of the encoder and decoder are untied. I am not sure why the authors didn't just start with a model that unties the encoder and decoder (which seems to be the model which is analysed) in place of equation (4). (A minor aside, it is not clear why we "drop the expectation").
    - In equation (10), we obtain a loss, which is ostensibly an object that should be minimised. What then does the stop-gradient operator (sg) mean, in terms of a loss that should be minimised? It is clear that procedurally that when applying gradient descent to a loss (10), the gradient step should involve an appropriate modification of the gradient of a modified equation (10). But what does this mean mathematically in terms of the loss?
    - Equation (12). For some reason the authors drop a certain Hadarmard product to obtain (12) from (11). It is not clear what "it does also work without it" (What is the first "it" referring to, what does "work" mean, and what is the second "it"?). Appendix F.4 is not helpful in this respect. I am not sure what the quality of the approximation is here, even intuitively or qualitatively.
    - Equation (13). I suspect the authors mean to say  that $W^\top W \approx I$ (because they mention using a regulariser or penalty rather than a hard constraint). There is no discussion around the approximation error in equation (13).
    - Equation (14). Suddenly the previously mentioned dropped Hadarmard product (or something similar? it is not clear) is reintroduced. I am not sure why, if this is an approximation, or if this modifies the original learning objective. Even more confusing, equation (14) is stated as an equality.
    - Given all the above points, I am not sure whether the paper lives up to the title of a "uniformed neural model for linear/nonlinear PCA", because it seems as though the authors change terms/factors one at a time, here and there until they arrive at the result they want, without justification of the steps involved.

**Questions:**

Questions:
- Top of page 4. " we will untie the weights of the encoder We and the decoder Wd, drop the expectation, and multiply by 1/2 for convenience," Why drop the expectation? This appears to be purely to remove a symbol from the notation, because under extremely mild conditions the gradient will pass through the expectation anyway.
- Also relating to the above, what is the reason for untying the weights of the encoder and decoder?
- Under equation (10) there is a discussion around $\Sigma$, which I do not understand. In particular, it is mentioned that $\Sigma$ need not be differentiable, however $\Sigma$ is a *parameter*, not a *function*. I am not sure what differentiability of $\Sigma$ means (differentiability with respect to what?). Is it possible that the authors meant to say that $\mathcal{L}$ need not be differentiable with respect to $\Sigma$?
- What should be the key takeaway message from the two experiments conducted in section 4? I see that you tried PCA, nonlinear PCA, and ICA, but how should one assemble these qualitative figures into a concrete message for the reader?

Related work: Could the authors comment on the relevance of (probabilistic) exponential family PCA?
- A Generalization of Principal Component Analysis to the Exponential Family, NeurIPS 2001
- Bayesian Exponential Family PCA, NeurIPS 2008

---

> ### Author Response · Authors · 2023-11-13
>
> We thank you for your time and helpful comments. We have added a general comment for everyone that addresses certain points that overlap with all the reviewers. Below we answer point by point, grouping the common weaknesses and questions.
>
> >The biggest weakness, in my opinion, is the imprecision/informality of the presentation. I am not sure whether the derivation is correct/rigorous/precise, because at each step and equation, it is not clear whether the original model is modified to accommodate the step in question, whether an approximation is applied, and whether the approximation or modification is valid. There is no discussion or attempt to quantify the effect of these various simplifications, or even formally state what these simplifications are.
> > As far as I can tell, my confusion begins at equation (5), where the weights of the encoder and decoder are untied. I am not sure why the authors didn't just start with a model that unties the encoder and decoder (which seems to be the model which is analysed) in place of equation (4). (A minor aside, it is not clear why we "drop the expectation").
>
>
> > Question: Top of page 4. " we will untie the weights of the encoder We and the decoder Wd, drop the expectation, and multiply by 1/2 for convenience," Why drop the expectation? This appears to be purely to remove a symbol from the notation, because under extremely mild conditions the gradient will pass through the expectation anyway.
> > Question: Also relating to the above, what is the reason for untying the weights of the encoder and decoder?
>
> The weights are untied only to make it easier to follow how the gradients are computed. The main property with the gradient for a given weight is that it is additive.
>
> For instance, suppose you have the loss $L = ||h(h(h(W)W)W||^2$. $W$ appears three times, and each one of those appearances contributes to the gradient. The total gradient of $W$ is the sum of the gradients of each one of those contributions. To make it easier to compute those gradients and refer to them explicitly, we can imagine that we have annotated them as follows  $||h(h(h(W_1)W_2)W_3||^2$ , but this can also be seen as untying the weights, and the act of summing up the gradients can be seen as tying back the weights.
>
> We can thus now write $dW = dW_1 + dW_2 + dW_3$, and have the knowledge where each term originated from.
>
> The reason we drop the expectation is purely to simplify the notation when writing the gradients, i.e. to avoid having to write $E(\frac{\partial L}{\partial W})$ everywhere. There is no loss in generality by dropping the expectation.
>
> We have amended the paper in the hopes of clarifying that is done only for this purpose.
>
>
>
> >In equation (10), we obtain a loss, which is ostensibly an object that should be minimised. What then does the stop-gradient operator (sg) mean, in terms of a loss that should be minimised? It is clear that procedurally that when applying gradient descent to a loss (10), the gradient step should involve an appropriate modification of the gradient of a modified equation (10). But what does this mean mathematically in terms of the loss?
>
> Now suppose we want to eliminate $dW_2$ from the above $dW$. The stop gradient operator makes it possible to do so. It simply means not to accumulate the gradient from that specific variable. Adding the stop gradient to $||h(h(h(W)[W]_{sg})W||^2$, mathematically means that the resulting gradient is $dW = dW_1 + dW_3$
>
> In tensorflow,this is a direct correspondence with this operator, and we can simply write  `tf.stop_gradient(W_2)`. https://www.tensorflow.org/api_docs/python/tf/stop_gradient
>
> In our case, the stop gradient serves to eliminate the gradient term that would originate from the decoder contribution.
>
> Another example that we mention, is the subspace learning algorithm which has the weight update $\Delta W = x^Ty - Wy^Ty$. This is simply the decoder contribution from the linear PCA autoencoder. To write it as a loss, we can use the stop gradient operator: $||x-x[W]_{sg}W^T||^2$. Without the stop gradient, then  the weight update from $||x-xWW^T||^2$  would be $\Delta W = x^Ty - Wy^Ty + x^Ty(I-W^TW)$, so the stop gradient serves to eliminate $x^Ty(I-W^TW)$.

---

> ### Author Response · Authors · 2023-11-13
>
> > Equation (12). For some reason the authors drop a certain Hadarmard product to obtain (12) from (11). It is not clear what "it does also work without it" (What is the first "it" referring to, what does "work" mean, and what is the second "it"?).Appendix F.4 is not helpful in this respect. I am not sure what the quality of the approximation is here, even intuitively or qualitatively.
>
>
> The dropping of h' is not part of the main result: it is simply an interesting observation. In hindsight, it does detract significantly from the main result. We simply noticed that its absence did not affect the results, in particular for the image patches. We have moved the mention of it to appendix E.1.1 where it is needed for the stationary point approximation. Apart from Appendix F.4 where we explicitly state that it is dropped, it is not used anywhere else. Initially we had placed it there in attempt to highlight that the key driver for nonlinear is the latent reconstruction while for linear PCA it is the input reconstruction
>
>
>
> > Equation (13). I suspect the authors mean to say that (because they mention using a regulariser or penalty rather than a hard constraint). There is no discussion around the approximation error in equation (13).
>
>
> To avoid this giving the impression that we are also making another approximation, we have moved this part to the Appendix E.3, as the main reason for it was simply to show that it can be put in relation to blind deconvolution. And it is this connection to blind convolution that connects nonlinear PCA with kurtosis maximisation and maximum likelihood to show that it maximises independence.
>
>
> [1] Juha Karhunen, Petteri Pajunen, and Erkki Oja. The nonlinear pca criterion in blind source separation:
> Relations with other approaches. Neurocomputing, 22(1-3):5–20, 1998
>
> [2] Aapo Hyvärinen and Erkki Oja. Independent component analysis: algorithms and applications.
> Neural networks, 13(4-5):411–430, 2000. Chapter 12.7, p 251
>
> >Equation (14). Suddenly the previously mentioned dropped Hadarmard product (or something similar? it is not clear) is reintroduced. I am not sure why, if this is an approximation, or if this modifies the original learning objective. Even more confusing, equation (14) is stated as an equality.
>
> Our removal of the mention of any dropping of h' should hopefully clear up the confusion.

---

> ### Author Response · Authors · 2023-11-13
>
> > Given all the above points, I am not sure whether the paper lives up to the title of a "uniformed neural model for linear/nonlinear PCA", because it seems as though the authors change terms/factors one at a time, here and there until they arrive at the result they want, without justification of the steps involved.
>
> The only modification that we have made is the introduction of sigma and the use of the stop gradient to omit the decoder contribution.
> In hindsight, the mention of dropping h' at that point in the analysis was a mistake. The intention was to make the observation that it's not the main driver for nonlinear PCA and in fact it is the difference in the latent reconstruction. But we failed to phrase that in a clear way.
>
> > Under equation (10) there is a discussion around , which I do not understand. In particular, it is mentioned that need not be differentiable, however is a parameter, not a function. I am not sure what differentiability of means (differentiability with respect to what?). Is it possible that the authors meant to say that need not be differentiable with respect to ?
>
>
> That is a lapsus on our part, and indeed we meant differentiable with respect to the loss. We have updated the manuscript to replace it with "trainable" as a single word to best describe this.
>
> > What should be the key takeaway message from the two experiments conducted in section 4? I see that you tried PCA, nonlinear PCA, and ICA, but how should one assemble these qualitative figures into a concrete message for the reader?
>
> We hope that the above explanation that we have added in terms of the SVD of a linear transformation helps in clarifying the takeaway.
> The main point is that our formulation allows nonlinear PCA to be applied directly on the data to learn an orthogonal linear transformation that can separate components that have similar variances. This is something that linear PCA is unable to do. For the time series this is evident where linear PCA failed to separate the signals that had the same variance, while nonlinear was able to separate them.
> The difference between linear ICA and nonlinear PCA is that linear ICA is more general in that it learns a linear transformation of the form $B=W\Sigma^{-1}V$, while nonlinear PCA learns $B=U\Sigma^{-1}$. This evident when in Fig.5 where nonlinear PCA fails to separate the components when the overall transformation is not orthogonal, i.e. of the form $B=U\Sigma^{-1}V$, but nonlinear PCA can be used to learn both $U$ and $V$ to recover $B$.
>
>
> > Related work: Could the authors comment on the relevance of (probabilistic) exponential family PCA?
> > A Generalization of Principal Component Analysis to the Exponential Family, NeurIPS 2001
> > Bayesian Exponential Family PCA, NeurIPS 2008
>
> They relate to generalising PCA to non-real-valued data. Both papers approach PCA from the probabilistic interpretation. As we've mentioned in our introduction, PCA has in fact three interpretation: neural, manifold, and probabilistic. In context to our work, we've only focused on the neural interpretation. It would, however, be interesting to explore, as future work, the modification that we have made to nonlinear PCA and how that relates to the probabilistic interpretation.

---

> > ### Comment · Reviewer_cVoi · 2023-11-21
> >
> > Thanks very much for your response to my questions, as well as your general response and response to other reviewers. I have read the other reviews and your responses and see that other reviewers had different concerns and also shared some of my concerns. I appreciate your clarifications for some of the points. I don't think this work is ready to be published, even with the changes after the review process. I will keep my score as is. I would encourage the authors to rethink the way they organise the paper for their next submission.
> >
> > - Respect the meaning of the symbol $=$. For example, instead of writing somewhere $\mathcal{L}(\ldots) = \mathbb{E}[\ldots]$ and then later removing the $\mathbb{E}$ and writing $\mathcal{L}(\ldots) = \ldots$, just define a loss $\ell(\ldots)$ and an expected loss $\mathcal{L} = \mathbb{E}[\ell(\ldots)]$ and then describe the properties of $\ell$. Another example is redefining a $4$ argument version of $\mathcal{L}$ after defining a 2 argument version of $\mathcal{L}$. Just define the one that is required. If the proof technique for the gradient requires introducing new terms, these are things that can be safely sent to the appendix to avoid load on the reader. Use the symbol $\approx$ with caution.
> > - I am still not clear on the meaning of the stop gradient operator. As I initially mentioned, I understand that it procedurally stops gradients in a backpropagation algorithm and is easy to implement in deep learning frameworks. I do not understand what this loss, as an objective function itself, actually means in terms of mathematical optimsiation. How does stop gradient translate to a mathematical objective?
> > - "The only modification that we have made is the introduction of sigma and the use of the stop gradient to omit the decoder contribution". These are still modifications that change the meaning of the claimed equivalence. If I we change an equation $1 = 1$ to $1 = 1.00000001$, we can prove any equivalence. Is there any way you can present a formal, exact equivalence, and then later informally talk about the other modifications you make?

---

### Official Review · Reviewer_v4Cb · 2023-11-08

**Soundness:** 2 fair
**Presentation:** 3 good
**Contribution:** 2 fair
**Rating:** 5
**Confidence:** 3

**Summary:**

This work proposes $\sigma$-PCA, which is a variant of non-linear PCA that can distinguish the principle components with the same variances, and maintain orthogonality without whitening process. The author claims that $\sigma$-PCA can maximise both variance and statistical independence.

**Strengths:**

1. The paper is clearly written and well-structured.

2. The paper proposes a new variant of non-linear PCA, which emphasizes on latent reconstruction, and can seperate the components with the same variances, which cannot be achived with linear PCA.

3. The experiments shows that $\sigma$-PCA can lead to more recognizable filters and can also recover the signals with the same variances.

**Weaknesses:**

1. I feel some claims are not well supported, for instance, the author claims that $\sigma$-PCA can maximise the statiscal independence, which is the purpose of ICA. Does it mean we can also use it to replace FastICA? I also can't find why $\sigma$-PCA can maximise statiscal independence in the paper, although it is claimed to be.

2. For Eq. (9), why the stop-gradient operator is introduced, I found it is a little bit arbitrary, my understanding is that is is aimed to solve the issue that the gradient of encoder is zero when $W_{e}=W_{d}$. But why this is rational to introduce stop-gradient operator here? please clarify it.

3. The experiment only contains qualitative results, adding some quantitative results would be appreciated.

**Questions:**

1. How is $\sigma$-PCA connect with kernel PCA. What is the advantages and disadvantages?

2. In the time-signal experiment, why non-linear PCA can't recover the signals but 2-layer non-linear PCA can? What is the advantages and disadvantages of $\sigma$-PCA compared with fastICA?

3. How the statistical independece is maximized in $\sigma$-PCA, does that mean it also guarantees identifiability like ICA?

---

> ### Author Response · Authors · 2023-11-13
>
> We thank you for your time and helpful comments. We have added a general comment for everyone that addresses certain points that overlap with all the reviewers. Below we answer point by point, grouping the common weaknesses and questions.
>
> > I feel some claims are not well supported, for instance, the author claims that -PCA can maximise the statiscal independence, which is the purpose of ICA. Does it mean we can also use it to replace FastICA? I also can't find why -PCA can maximise statiscal independence in the paper, although it is claimed to be.
>
> > Question: How the statistical independece is maximized in  $\sigma$-PCA, does that mean it also guarantees identifiability like ICA?
>
> Proof of how nonlinear PCA maximises independence has been shown two decades ago [1], with a review in context in [2]. Why it maximises independence in particular can be seen when the loss is put in relation to blind deconvolution, which would then connect it to kurtosis maximisation and maximum likelihood [2]. Previously we had mentioned the relationship to blind deconvolution in the weight update analysis, but we have moved it now to Appendix E.3 to avoid detracting from the main result. Nonlinear PCA guarantees identifiability when the transformation is orthogonal. It does not replace FastICA, because FastICA learns the linear transformation $B=W\Sigma^{-1}V$, where $W$ and and $V$ are orthogonal. What we show is that with our proposed formulation, nonlinear PCA can learn not just $V$ but also $W$.
>
> We hope that the interpretation from the point of view of the SVD of a linear transformation helps explain the connection between linear PCA, nonlinear PCA, and linear ICA. We have updated the abstract and the end of the introduction to reflect this interpretation.
>
>
> [1] Juha Karhunen, Petteri Pajunen, and Erkki Oja. The nonlinear pca criterion in blind source separation:
> Relations with other approaches. Neurocomputing, 22(1-3):5–20, 1998
>
> [2] Aapo Hyvärinen and Erkki Oja. Independent component analysis: algorithms and applications.
> Neural networks, 13(4-5):411–430, 2000. Chapter 12.7, p 251

---

> ### Author Response · Authors · 2023-11-13
>
> > For Eq. (9), why the stop-gradient operator is introduced, I found it is a little bit arbitrary, my understanding is that is is aimed to solve the issue that the gradient of encoder is zero when. But why this is rational to introduce stop-gradient operator here? please clarify it.
>
> The stop gradient is a straightforward way to omit the contribution of certain weights to the weight update.
>
> For instance, suppose you have the loss $L = ||h(h(h(W)W)W||^2$. $W$ appears three times, and each one of those appearances contributes to the gradient. The total gradient of $W$ is the sum of the gradients of each one of those contributions. To make it easier to compute those gradients and refer to them explicitly, we can imagine that we have annotated them as follows  $||h(h(h(W_1)W_2)W_3||^2$ , but this can also be seen as untying the weights, and the act of summing up the gradients can be seen as tying back the weights.
>
> We can thus now write $dW = dW_1 + dW_2 + dW_3$, and have the knowledge where each term originated from.
>
> Now suppose we want to eliminate $dW_2$ from  $dW$. The stop gradient operator makes it possible to do so. It simply means not to accumulate the gradient from that specific variable. Adding the stop gradient to $||h(h(h(W)[W]_{sg})W||^2$, mathematically means that the resulting gradient is $dW = dW_1 + dW_3$
>
> In tensorflow, there is a direct correspondence with this operator, and we can simply write  `tf.stop_gradient(W_2)`. https://www.tensorflow.org/api_docs/python/tf/stop_gradient
>
> In our case, the stop gradient serves to eliminate the gradient term that would originate from the decoder contribution. Another example that we mention, is the subspace learning algorithm which has the weight update $\Delta W = x^Ty - Wy^Ty$. This is simply the decoder contribution from the linear PCA autoencoder. To write it as a loss, we can use the stop gradient operator $||x-[W]_{sg}W^T||^2$
>
> The reason why we need to remove it is because when $W$ is not a square orthogonal matrix,  the decoder contribution overpowers the encoder contribution. We gave an explanation why it overpowers in appendix E6. We also discuss E6, how instead of removing it completely, they could be put on the same scale by normalising the decoder contribution, but it is simpler to omit it as the decoder contribution is detrimental for nonlinear PCA, while it is relevant for linear PCA.
>
>
> > The experiment only contains qualitative results, adding some quantitative results would be appreciated.
>
> Our results includes three types of data: two synthetic and one based on images. For the synthetic data we have
>
> - The time series of which part of the results we've included in the main paper, and the rest in Appendix F.5.
> - 2D points that explore the behaviour based on sub- and super- Gaussian distribution. Included in the Appendix F6.
>
> The main goal is to see if we recover the original data up to sign indeterminacy. This can be immediately seen from the figures. Quantitatively, we can compute the difference between the original transformation used to generate the data and its recovered version, taking into account the sign indeterminacy, but then that provides a less intuitive way to understand what happened that from looking at figures. The use of times series, 2D points, and image filters is something that has been used many times to demonstrate how linear PCA and linear ICA work.
>
> [1] Aapo Hyvärinen and Erkki Oja. Independent component analysis: algorithms and applications.
> Neural networks, 13(4-5):411–430, 2000.
>
> Could you suggest a quantitative result that you think would be appropriate in this case?
>
>
> >How is $\sigma$-PCA connect with kernel PCA. What is the advantages and disadvantages?
>
> Kernel PCA is for learning nonlinear transformations. The scope of this paper is linear transformations, so kernel PCA solves a different problem.
>
> > In the time-signal experiment, why non-linear PCA can't recover the signals but 2-layer non-linear PCA can? What is the advantages and disadvantages of -PCA compared with fastICA?
>
>
> We hope that the above explanation that we have added in terms of the SVD of a linear transformation helps to clarify this point
> The difference between linear ICA and nonlinear PCA is that linear ICA is more general in that it learns a linear transformation of the form $B=W\Sigma^{-1}V$, while nonlinear PCA learns $B=W\Sigma^{-1}$. This means that if the linear transformation is orthogonal, then $B$ reduces to  $B=W\Sigma^{-1}I = W\Sigma^{-1}$. so both nonlinear PCA and fastICA would coincide. But if the transformation is non-orthogonal $B=W\Sigma^{-1}V$, fastICA first learns $W'\Sigma^{-1}$ with linear PCA, then $V'$ with the fast-fixed point algorithm. Both $W$ and $V$ can be learnt with our formulation of nonlinear PCA. So if we use nonlinear PCA to learn both $W$ and $V$, this is similar to a 2-layer linear autoencoder, but can be shown it reduces to a single-layer because it is linear.

---

### Official Review · Reviewer_dby8 · 2023-11-09

**Soundness:** 1 poor
**Presentation:** 1 poor
**Contribution:** 1 poor
**Rating:** 3
**Confidence:** 4

**Summary:**

This paper tries to unify the linear and nonlinear PCA as single-layer autoencoders, in order to keep the orthogonality and ordering of variances that Linear principal component analysis (PCA) maintains.

**Strengths:**

Trying to use single-layer autoencoders to unify the linear and nonlinear PCA may be a good attempt, so that even the nonlinear PCA can keep the orthogonality and ordering of variances that Linear principal component analysis (PCA) maintains.

**Weaknesses:**

1. The paper consistently makes incorrect assumptions, particularly in the context of non-linear PCA. In traditional non-linear PCA, the specific non-linear mapping functions, such as tanh, are typically unknown and considered part of the modeling process. The paper, however, specifies these mappings explicitly, and deviates their new method based on this.


2. While linear PCA maintains orthogonality and orders variances to derive principal components, the insistence on preserving these characteristics in non-linear PCA might seem counterintuitive. Understanding the advantages of maintaining orthogonality and ordered variances in the non-linear context is crucial. Clarification on why this approach is beneficial by  keeping the orthogonality and ordering of variances?

3. The paper appears to leverage complex neural networks, such as single-layer autoencoders, to unveil linear and non-linear patterns in observed data. This approach raises concerns about the departure from the traditionally interpretable nature of principal component analysis (PCA). Even with the effectiveness of neural networks in capturing underlying patterns, explaining the significance of each component becomes challenging, potentially compromising the simplicity and interpretability associated with PCA.

**Questions:**

1.  In the non-linear PCA, the author specifies the non-linear mapping, such as tanh. Isn't the non-linear mapping some function that we should not know in the non-linear PCA? Is the non-linear mapping some function that we should know in the non-linear PCA?

2. We all know that the Linear principal component analysis (PCA) maintains the orthogonality and ordering of variances, and this is how we get the PCs. But for non-linear PCA, why do the authors insist on keeping the orthogonality and ordering of variances? What advantages will it bring to keep the orthogonality and ordering of variances?

3. It seems that the authors use complex neuron networks, even just single-layer autoencoders, to discover the linear/non-linear patterns of the observed data, if I understand correctly. Isn't a bit far away from the attractive easily explanatory property of of the principal component analysis (PCA)?
even single-layer autoencoders can perfectly discover the underlying patterns of the observed data, is it easy to explain which component is more important by using neural networks?

---

> ### Author Response · Authors · 2023-11-13
>
> We thank you for your time and comments. We have added a general comment for everyone that addresses certain points that overlap with all the reviewers. Below we answer point by point, grouping the weaknesses and questions that overlap.
>
>
> > 1. The paper consistently makes incorrect assumptions, particularly in the context of non-linear PCA. In traditional non-linear PCA, the specific non-linear mapping functions, such as tanh, are typically unknown and considered part of the modeling process. The paper, however, specifies these mappings explicitly, and deviates their new method based on this.
>
> > Question: In the non-linear PCA, the author specifies the non-linear mapping, such as tanh. Isn't the non-linear mapping some function that we should not know in the non-linear PCA? Is the non-linear mapping some function that we should know in the non-linear PCA?
>
> Which traditional nonlinear PCA are you referring to? As we have mentioned in the related work, the term "nonlinear PCA" has been used to refer to many different variants. In the background section, we have explicitly mentioned that it is the single-layer autoencoder variant that has been studied in [1,2]. In particular, it is the nonlinear PCA from the point of view of ICA, and it is the same as the one defined in the comprehensive book [3].
>
> Nonlinear PCA is a special case of ICA. tanh is a function that is at the core of many linear ICA methods, in particular being the derivative of logcosh. We have, therefore, not deviated from this.
>
> [1] Juha Karhunen and Jyrki Joutsensalo. Representation and separation of signals using nonlinear pca
> type learning. Neural networks, 7(1):113–127, 1994
>
> [2] Juha Karhunen, Petteri Pajunen, and Erkki Oja. The nonlinear pca criterion in blind source separation:
> Relations with other approaches. Neurocomputing, 22(1-3):5–20, 1998
>
> [3] Aapo Hyvärinen and Erkki Oja. Independent component analysis: algorithms and applications.
> Neural networks, 13(4-5):411–430, 2000 Chapter 12.7, p 251
>
>
>
> > 1. While linear PCA maintains orthogonality and orders variances to derive principal components, the insistence on preserving these characteristics in non-linear PCA might seem counterintuitive. Understanding the advantages of maintaining orthogonality and ordered variances in the non-linear context is crucial. Clarification on why this approach is beneficial by keeping the orthogonality and ordering of variances?
>
> >Question: We all know that the Linear principal component analysis (PCA) maintains the orthogonality and ordering of variances, and this is how we get the PCs. But for non-linear PCA, why do the authors insist on keeping the orthogonality and ordering of variances? What advantages will it bring to keep the orthogonality and ordering of variances?
>
>
> We hope that the general comment addressed to all reviewers helps clarify why. The gist is that if you consider any linear transformation $B$, it has the decomposition $W\Sigma^{-1}V$. Both $W$ and $V$ are orthogonal. Linear PCA can learn the first orthogonal transformation, but conventional nonlinear PCA cannot -- which is why it currently only works with whitening. This means that nonlinear PCA can only learn the second orthogonal transformation, $V$.  Although $W$ and $V$ are orthogonal, $B= W\Sigma^{-1}V$ is not necessarily orthogonal, and so the overall transformation when we apply nonlinear PCA is not orthogonal. In this paper we propose a way to make it work to learn the first orthogonal transformation $W$. With $W$ we can order the variances because the data has not been normalised yet by $\Sigma^{-1}$.

---

> ### Author Response · Authors · 2023-11-13
>
> > 3. The paper appears to leverage complex neural networks, such as single-layer autoencoders, to unveil linear and non-linear patterns in observed data. This approach raises concerns about the departure from the traditionally interpretable nature of principal component analysis (PCA). Even with the effectiveness of neural networks in capturing underlying patterns, explaining the significance of each component becomes challenging, potentially compromising the simplicity and interpretability associated with PCA.
>
> > Question: It seems that the authors use complex neuron networks, even just single-layer autoencoders, to discover the linear/non-linear patterns of the observed data, if I understand correctly. Isn't a bit far away from the attractive easily explanatory property of of the principal component analysis (PCA)? even single-layer autoencoders can perfectly discover the underlying patterns of the observed data, is it easy to explain which component is more important by using neural networks?
>
>
> To clarify, linear PCA, nonlinear PCA, and linear ICA all apply in the context of learning a linear transformation. This paper is solely about linear transformations.
> Despite the presence of the term nonlinear, with a single-layer autoencoder, the resulting transformation is still linear. We have not used a complex neural network, and we are not using it to unveil nonlinear patterns. We have updated the paper to emphasise these points.
>
> A key insight about PCA, discovered a few decades ago, is that a single-layer autoencoder is PCA [1,2,3]. Many variant such as the generalised hebbian algorithm [4], the weighted subspace algorithm [5] -- which have been proven to converge to the linear PCA solution -- can be shown to have a relationship to a linear autoencoder. In the Appendix A, we have listed these variants and many more.
> The problem with linear PCA is that it fails to separate components that share the same variance -- by definition from the SVD. There is a rotational indeterminacy. Nonlinear PCA does not suffer from rotational indeterminacy because it is an ICA method. Without the rotational indeterminacy there is even better interpretability as it disentangles the components that share the same variance.
>
>
> [1] Pierre Baldi and Kurt Hornik. Neural networks and principal component analysis: Learning from
> examples without local minima. Neural networks, 2(1):53–58, 1989.
>
> [2] Hervé Bourlard and Yves Kamp. Auto-association by multilayer perceptrons and singular value
> decomposition. Biological cybernetics, 59(4):291–294, 1988
>
> [3] Erkki Oja. Principal components, minor components, and linear neural networks. Neural networks, 5
> (6):927–935, 1992b
>
> [4] Terence D Sanger. Optimal unsupervised learning in a single-layer linear feedforward neural network.
> Neural networks, 2(6):459–473, 1989
>
> [5] E Oja. Principal component analysis by homogeneous neural networks, part i: The weighted subspace
> criterion. IEICE Trans. on Inf. and Syst., 75(3):366–375, 1992a

---

### Official Review · Reviewer_xK5D · 2023-11-10

**Soundness:** 3 good
**Presentation:** 3 good
**Contribution:** 2 fair
**Rating:** 5
**Confidence:** 2

**Summary:**

This paper presented $\sigma$-PCA, which is a unified neural model for both linear PCA and nonlinear PCA.

$\sigma$-PCA dropped the requirement of whitening and both the orthogonality and the variance order are preserved in the nonlinear version.

The nonlinear version can handle the orthogonal case with similar variance which linear PCA cannot handle. It can also perform ICA to deal with the non-orthogonal case.

Experiments on image patches and time signals show some advantage compared with linear and nonlinear PCA.

**Strengths:**

1. One of the key contribution for this work is that it draw connection among a wide range of literature for nonlinear PCA. The proposed objective can be easily adapted to those cases.
2. In particular for linear PCA, nonlinear PCA, and ICA, comprehensive discussion was provided on which case which models will coincide.
3. Compared with ICA, since $\sigma$-PCA dosen't require whitening, then the orghogonality is preserved.
4. Compared with conventional PCA, $\sigma$-PCA unified the linear and nonlinear case which can handle the case when components have similar variance.

**Weaknesses:**

1. The key idea that this work claim is to explicitly model the varience, which conventional nonlinear PCA methods utilize whitening to bypass. However, the idea of modeling th variance is already explored in the literature (the author also mentioned this in the relative work section).
2. The necessity for introducing $\mathbf{\Sigma}$ is unclear.
2. The nonlinearity is another major concern. The nonlinear function discussed in this literature is rather simple, especially compared with nonlinear ICA literature. Also in the experiments together with the additional experiment results in the appendix, all those $h$ are fairly simple and may not suitable for more complex data.
3. Some experiment result is not very straightforward for showing the advantage of the proposed method. Especially for the image patches dataset, it's quite hard to tell the quality for the method purely from the figures showed in the paper. Also detailed analysis and discussion on the experiment result is missing.
4. Anothe concern is that the ICA mentioned in this work is linear ICA, however, for the nonlinearity, there are nonlinear ICA methods, which also emphasis the latent reconstruction.

**Questions:**

1. For the derivation of eq 5 to 8, it's a little bit confusing if the optimizating process doesn't distinguish encoder and decoder, it use the tied weight, why the analysis which seperates the $\mathbf{W}_e$ and $\mathbf{W}_d$ still holds valid?
2. As mentioned in Weaknesses, what is the necessity for introducing $\mathbf{\Sigma}$?
3. For the nonlinearity, if $h$ is not an element wise function like tanh, but a complex nonlinear function (which is usually the case in nonlinear ICA literature), what will happen? Seems that the analysis for dropping $h'$ may not hold anymore.
4. How those filters picture in Figure 2 are connected with the quality for the method. The time series experiment is intuitive but it is too simple to judge the effectiveness. Some quantitive result is expected to show $\sigma$-PCA is a better solution. Some experiments can be synthetic with assumptions like varient scales for data varience.
5. Figure 2 seems have a lot of information by comparing the performance among those methods. A detailed discussion about this result is expected.

---

> ### Author Response · Authors · 2023-11-13
>
> We thank you for your time and helpful comments. We have added a general comment for everyone that addresses certain points that overlap with all the reviewers. Below we answer point by point, grouping the weaknesses and questions that overlap.
>
> > 1. The key idea that this work claim is to explicitly model the varience, which conventional nonlinear PCA methods utilize whitening to bypass. However, the idea of modeling th variance is already explored in the literature (the author also mentioned this in the relative work section).
>
> The idea of modelling the variance was explored in the literature from the probabilistic view, specifically for general linear ICA, and the variances were integrated out. They have not been explored from the neural point of view, i.e. in the form of an autoencoder. We also explicitly compute the variance.
>
>
> > 2. The necessity for introducing is unclear.
> > Question: As mentioned in Weaknesses, what is the necessity for introducing ?
>
> Intuitively, the reason why $\Sigma$ is necessary is because it standardises $y$ to be on the same scale, namely to have unit variance, before applying $h$. If for instance we use a squashing function like $\tanh$, any value $|y| > 2$ would be squashed close to $1$, having no gradient. This would be a problem if the variance of $y$ is large.
>
> A secondary reason is that it naturally appears in the SVD. Any matrix can be decomposed into $A = W\Sigma^{-1}V$, with W and V orthogonal and $\Sigma$ diagonal. So instead of doing the grouping $(W)\Sigma^{-1}V$, we can do $(W\Sigma^{-1})V$. This grouping $W\Sigma^{-1}$ now allows us to have unit variance.
>
> > 3. The nonlinearity is another major concern. The nonlinear function discussed in this literature is rather simple, especially compared with nonlinear ICA literature. Also in the experiments together with the additional experiment results in the appendix, all those are fairly simple and may not suitable for more complex data.
> >
>
> The scope of this paper is learning linear transformations, and so nonlinear ICA is outside the scope of this paper, given that it is for learning nonlinear transformations. Unfortunately, in the ICA literature, the term "nonlinear" refers to the transformation; nonetheless, linear ICA uses a nonlinear function -- otherwise, it cannot work. FastICA, for instance, relies on using the $\tanh$ function, being the derivative of logcosh. On the other hand, for PCA, and in particular, single-layer autoencoder, the term "nonlinear" refers to the function [1,2], and so nonlinear PCA uses a nonlinear function, just like linear ICA (in fact, it is a linear ICA method), but it learns a linear transformation.
>
>
> We are not concerned in this paper with complex data.
>
> [1] Juha Karhunen, Petteri Pajunen, and Erkki Oja. The nonlinear pca criterion in blind source separation:
> Relations with other approaches. Neurocomputing, 22(1-3):5–20, 1998
> [2] Juha Karhunen and Jyrki Joutsensalo. Representation and separation of signals using nonlinear pca
> type learning. Neural networks, 7(1):113–127, 1994.
>
> > 4. Some experiment result is not very straightforward for showing the advantage of the proposed method. Especially for the image patches dataset, it's quite hard to tell the quality for the method purely from the figures showed in the paper. Also detailed analysis and discussion on the experiment result is missing.
>
> The appendix F contains additional experiments. In particular, the main advantage of nonlinear PCA over linear PCA is that it can separate components that have the same variance. This is clearly demonstrated with the time series as well as with 2D points for sub- and super-Gaussian distribution in the Appendix F.6. For the image patches, this can be seen in how some of the linear PCA filters appear as blurry superpositions of crisp nonlinear PCA filters. The main discussion points we currently have are about highlighting those observations.

---

> ### Author Response · Authors · 2023-11-13
>
> > 5. Anothe concern is that the ICA mentioned in this work is linear ICA, however, for the nonlinearity, there are nonlinear ICA methods, which also emphasis the latent reconstruction.
>
>
> As mentioned above, the context of this paper is linear transformation. Nonlinear ICA is thus outside the scope of this paper.
>
> Questions:
> > 1. For the derivation of eq 5 to 8, it's a little bit confusing if the optimizating process doesn't distinguish encoder and decoder, it use the tied weight, why the analysis which seperates the $W_e$ and $W_d$  still holds valid?
>
>
> The weights are untied only to make it easier to follow how the gradients are computed. The main property with the gradient for a given weight is that it is additive.
>
> For instance, suppose you have the loss $L = ||h(h(h(W)W)W||^2$. $W$ appears three times, and each one of those appearances contributes to the gradient. The total gradient of $W$ is the sum of the gradients of each one of those contributions. To make it easier to compute those gradients and refer to them explicitly, we can imagine that we have annotated them as follows  $||h(h(h(W_1)W_2)W_3||^2$ , but this can also be seen as untying the weights, and the act of summing up the gradients can be seen as tying back the weights.
>
> We can thus now write $dW = dW_1 + dW_2 + dW_3$, and have the knowledge where each term originated from.
>
>
> We have amended the paper in the hopes of clarifying that is done only for this purpose.
>
>
> > For the nonlinearity, if is not an element wise function like tanh, but a complex nonlinear function (which is usually the case in nonlinear ICA literature), what will happen? Seems that the analysis for dropping h' may not hold anymore.
>
> The dropping of h' is not part of the main result: it is simply an interesting observation. We simply noticed that its absence did not affect the obtained filter. In hindsight, and as you and another reviewer point out, it does detract from the main result, and so we have moved this to the appendix E.1.1, where it is needed for a simplified stationary point analysis. In the summary figure, listing the variations of nonlinear PCA, we have a result that shows the obtained filters are not affected by the removal of h', highlighting that the main driver of the separation is the difference between the latents.
>
>
> > How those filters picture in Figure 2 are connected with the quality for the method. The time series experiment is intuitive but it is too simple to judge the effectiveness. Some quantitive result is expected to show -PCA is a better solution. Some experiments can be synthetic with assumptions like varient scales for data varience.
>
> Appendix F6 presents additional experiments with synthetic data. In particular, it looks at super- and sub- gaussian distributions. A well-know problem with PCA is that it cannot separate components with the same variances -- this is by definition from SVD. The experiments attempt to highlight this in particular and how nonlinear PCA does not have the same problem.
>
>
> > Figure 2 seems have a lot of information by comparing the performance among those methods. A detailed discussion about this result is expected.
>
> The main thing that Figure 2 wants to show is
>
> - conventional nonlinear PCA fails to generate any useful filters.
> - nonlinear PCA fails to generate any useful filters when the decoder contribution is included.
> - nonlinear PCA generates crisp filters compared to linear PCA, mainly because it can separate filters that have the similar variances.
>
> Would perhaps including an additional figure in the appendix that explicitly shows what the variances are for each filter help emphasise our last point?

---

### Author Response · Authors · 2023-11-13

We thank all the reviewers for their time and helpful comments. They have helped to highlight the points that we need to emphasise and the ones that have perhaps caused a bit of confusion.

The main things that we want to emphasise are:

1. This paper is concerned with linear transformations only -- we have updated the paper so that this is explicit rather than implicit from context.
2. Linear PCA, nonlinear PCA, and linear ICA are methods for learning linear transformations from data -- we have updated the paper to explicitly state this.
3. For ICA, the terms linear and nonlinear refer to the transformation. On the other hand, for PCA, they refer to the function. Both linear ICA and nonlinear PCA use nonlinear functions, but their resulting transformations are still linear  -- this is based on what's currently in the literature; we have added this as a footnote.


We have updated the paper to focus on an explanation from the point of view of the singular value decomposition, which we hope would make it easier to clear up certain points of confusion. We have thus made a revision to the abstract and the end of the introduction to emphasise this part, but it is equivalent to what we had before, except that it is stated with different terminology. The update abstract is the following:



Linear principal component analysis (PCA), nonlinear PCA, and linear independent component analysis (ICA) -- those are three methods with single-layer autoencoder formulations for learning linear transformations with certain characteristics from data. Linear PCA learns rotations that orient axes to maximise variance, but it suffers from a subspace rotational indeterminacy: it fails to find a unique rotation for axes that share the same variance. Nonlinear PCA learns rotations that orient axes to maximise statistical independence -- this reduces the subspace indeterminacy from rotational to permutational. Linear ICA learns any linear transformations -- not just rotations -- that maximise statistical independence. The relationship between all three can be understood by the singular value decomposition of the linear ICA transformation into a sequence of rotation, scale, rotation. Linear PCA learns the first rotation; nonlinear PCA learns the second. The scale is simply the inverse of the standard deviations. The problem is that, in contrast to linear PCA, conventional nonlinear PCA cannot be used directly on the data to learn the first rotation, the first being special as it reduces dimensionality and orders by variances. In this paper, we have identified the reason it fails, and as a solution we propose $\sigma$-PCA:  a unified neural model for linear and nonlinear PCA as single-layer autoencoders.
Its key ingredient: modelling not just the rotation but also the scale -- the variances. This model bridges the disparity between linear and nonlinear PCA. With our formulation, nonlinear PCA can learn not just the second, but also the first rotation. And so, like linear PCA, it can learn an orthogonal transformation (a rotation) that reduces dimensionality and orders by variances, but, unlike linear PCA, it does not suffer from rotational indeterminacy.


Equivalent statement for after vs before:

- nonlinear PCA can only learn the 2nd rotation = nonlinear PCA requires a whitening steps
- Nonlinear PCA can learn the 1st rotation = nonlinear PCA retains the orthogonality of the overall transformation and can order by variances.
- unable to separate components that have the same variance = subspace rotational indeterminancy.
- two isolated layers can perform conventional ICA = nonlinear PCA can be used to learn both the first and second rotation